# MMLONGBENCH-DOC: Benchmarking Long-context Document Understanding with Visualizations

**Yubo Ma**[1], **Yuhang Zang**[2*], **Liangyu Chen**[1], **Meiqi Chen**[3], **Yizhu Jiao**[4]
**Xinze Li**[1], **Xinyuan Lu**[5], **Ziyu Liu**[6], **Yan Ma**[7], **Xiaoyi Dong**[2], **Pan Zhang**[2]
**Liangming Pan**[8], **Yu-Gang Jiang**[9], **Jiaqi Wang**[2], **Yixin Cao**[9*], **Aixin Sun**[1]
[1] S-Lab, Nanyang Technological University, [2] Shanghai AI Laboratory, [3] Peking University
[4] University of Illinois Urbana-Champaign, [5] National University of Singapore, [6] Wuhan University
[7] Singapore Management University, [8] University of Arizona, [9] Fudan University

## Abstract

Understanding documents with rich layouts and multi-modal components is a long-standing and practical task. Recent Large Vision-Language Models (LVLMs) have made remarkable strides in various tasks, particularly in single-page document understanding (DU). However, their abilities on long-context DU remain an open problem. This work presents **MMLONGBENCH-DOC**, a long-context, multi-modal benchmark comprising 1,082 expert-annotated questions. Distinct from previous datasets, it is constructed upon 135 lengthy PDF-formatted documents with an average of 47.5 pages and 21,214 textual tokens. Towards comprehensive evaluation, answers to these questions rely on pieces of evidence from (1) different sources (text, image, chart, table, and layout structure) and (2) various locations (*i.e.,* page number). Moreover, 33.7% of the questions are *cross-page questions* requiring evidence across multiple pages. 20.6% of the questions are designed to be *unanswerable* for detecting potential hallucinations. Experiments on 14 LVLMs demonstrate that long-context DU greatly challenges current models. Notably, the best-performing model, GPT-4o, achieves an F1 score of only 44.9%, while the second-best, GPT-4V, scores 30.5%. Furthermore, 12 LVLMs (all except GPT-4o and GPT-4V) even present worse performance than their LLM counterparts which are fed with lossy-parsed OCR documents. These results validate the necessity of future research toward more capable long-context LVLMs.

## 1   Introduction

Documents are one of the fundamental forms of information preservation and exchange. In each year, tens of millions of documents are created, read, saved, and dispatched [1]. Beyond unstructured pure-text, documents feature both complicated layout structures and information across distinct modalities such as text, table, chart, image, *etc.* Accordingly, the automatic understanding of documents (Document Understanding; DU) stands as a long-standing task in urgent and practical needs.

Recently, a number of LVLMs, both closed-source ones (GPT-4o [2], Gemini-1.5 [3], Claude-3 [4], *etc.*) and open-source ones (InternLM-XC2-4KHD [5], InternVL-Chat [6], Otter [7], LLaVA-NeXT [8], CogVLM [9], mPLUG-DocOwl 1.5 [10], TextMonkey [11], *etc.*) have been developed and presented the great potential to handle documents. Most of them have achieved promising performance on single-page DU datasets like DocVQA [12], ChartQA [13], InfoVQA [14], TAT-DQA [15], *etc.* However, considerable amounts of documents in the real world are long-context

---

*Corresponding Authors.
Project Page: `https://mayubo2333.github.io/MMLongBench-Doc`

38th Conference on Neural Information Processing Systems (NeurIPS 2024) Track on Datasets and Benchmarks.

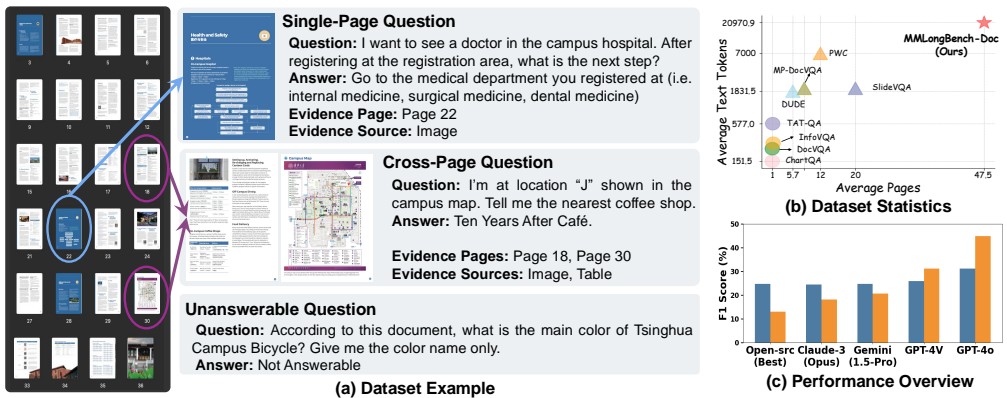

**Figure 1:** MMLONGBENCH-DOC evaluates understanding abilities of LVLMs on lengthy documents that span tens of pages and incorporate multi-modal elements. Experiments (bottom-right) indicate that most LVLMs struggle, even falling behind LLMs that are fed with only OCR-parsed documents.

documents with tens or even hundreds of pages. The understanding of these lengthy documents brings new challenges for LVLMs from at least two aspects: (1) **Localization**: identify and retrieve information from massive, heterogeneous information (similar to the *needle in a haystack* task); (2) **Cross-page comprehension**: collect and reason over multi-source information across different pages. These two kinds of abilities are beyond the evaluation scopes of the aforementioned single-page DU datasets. Some recent DU datasets [16; 17; 18] feature multiple-page DU, but almost all their documents are either as short of only several pages or of low information density, making the localization-related questions over-simple. Additionally, few (if any) questions in these datasets necessitate cross-page comprehension. See more detailed related work in Section 2. In summary, there lacks a unified and high-quality benchmark on lengthy documents, leaving the evaluation of long-context DU largely unexplored.

In this paper, we present MMLONGBENCH-DOC, a benchmark designed to evaluate the **M**ulti-**M**odality **Long**-context **Doc**ument understanding abilities of LVLMs. Towards a comprehensive benchmark, it incorporates lengthy documents from both four existing datasets [13; 17; 18; 19] and other various papers, brochures, *etc.* Consequently, our benchmark includes 135 PDF-formatted documents spanning across 7 diverse domains, with each document averaging 47.5 pages and 21,214.1 textual tokens. Regarding the questions, we employ ten expert-level annotators to (1) edit questions associated with documents from existing datasets to meet our benchmark's standard and (2) create new questions for all collected documents to expand the scale of the benchmark. Then a three-round, semi-automatic reviewing process ensures the benchmark's annotation quality. As a result, MMLONGBENCH-DOC comprises 1,082 human-annotated questions, with 184 sourced from four existing datasets and 898 newly annotated. Being a multi-modal benchmark, the answer to each question requires evidence from one or more of these five in-document sources: *text*, *layout*, *chart*, *table*, and *image*. Questions are categorized into three types based on the number of evidence pages [1], with examples illustrated in Figure 1(a): (1) 494 *single-page* questions (with one evidence page) mainly to evaluate localization abilities, (2) 365 *cross-page* questions (with multiple evidence pages) to assess cross-page comprehension, and (3) 223 *unanswerable* questions (no evidence for answering it, *i.e.,* no evidence pages) to reduce shortcuts and measure LVLMs' potential hallucinations. Meta-information including evidence pages, sources, and answer formats, is preserved for fine-grained evaluation and analysis. Detailed descriptions of the annotation pipeline and statistics can be found in Section 3.

We conduct extensive experiments on MMLONGBENCH-DOC to evaluate the long-context DU abilities of 14 LVLMs, including 4 proprietary and 10 open-source ones. Given a document, we screenshot each page and feed all of these PNG-formatted images to LVLMs in an end-to-end approach. For comparison, we also convert the documents to textual format by optical character recognition (OCR) and evaluate another 6 proprietary and 4 open-source 10 LLMs (6 proprietary and

---

[1]Given a document $D$ and a question $q$ upon $D$, We call page $P$ (in document $D$) an *evidence page* of $q$ if the answer of $q$ necessitates one or more pieces of evidence in page $P$.

Table 1: Comparison between our benchmark and previous DU datasets. **Unans.**: unanswerable question. **TXT/L/C/TAB/I**: pure text/generalized layout/chart/table/image. **Doc. Rel.**: document relevance. Whether document information is indispensable for the answer. **Avg. Position**: the average page index on which the answer evidence is located. *:Statistics from [20].

| Benchmarks | Document | | Question type | | | Answer Evidence | |
| --- | --- | --- | --- | --- | --- | --- | --- |
| | # Pages | # Tokens | Cross-page (%) | Unans. (%) | Doc. Rel. | Source | Avg. Position |
| DocVQA [12] | 1.0 | 151.5 | ✗ | ✗ | ↯ | TXT/L/C/TAB/I | - |
| ChartQA [13][2] | 1.0 | 236.9 | ✗ | ✗ | ✓ | C | - |
| InfoVQA [14][2] | 1.2 | 288.0 | ✗ | ✗ | ↯ | L/C/TAB/I | - |
| TAT-DQA [15] | 1.1 | 577.0 | ✗ | ✗ | ↯ | TXT/TAB | - |
| VisualWebBench [21][2] | 1.0 | 452.4 | ✗ | ✗ | ✓ | LAY/I | - |
| PWC [22] | ~12* | ~7000* | ✗ | ✗ | ↯ | TAB | - |
| MP-DocVQA [16] | 8.3 | 2026.6 | ✗ | ✗ | ↯ | TXT/L/C/TAB/I | 6.0 |
| DUDE [17] | 5.7 | 1831.5 | ✓(2.1%) | ✓(12.7%) | ↯ | TXT/L/C/TAB/I | 2.5 |
| SlideVQA [18] | 20.0 | 2030.5 | ✓(13.9%) | ✗ | ↯ | TXT/L/C/TAB/I | 9.1 |
| MMLONGBENCH-DOC | 47.5 | 21214.1 | ✓(33.0%) | ✓(22.5%) | ✓ | TXT/L/C/TAB/I | 23.6 |

4 open-source ones). The results in Figure 1(c) highlight the challenges that current LVLMs face with long-context DU. The best-performing LVLM, GPT-4o, achieves an overall F1 score of only 44.9%, while the second-best LVLM, GPT-4V, scores 30.5%. Moreover, all the remaining LVLMs tested with multi-modal documents performed worse than single-modal LLMs handling lossy, OCR-parsed texts. Specifically, the Gemini-1.5-Pro and Claude-3-Opus present 4.2% and 6.4% absolute decrease when the inputs change from document screenshots to OCR-parsed texts. Regarding open-source models, the best-performing LVLM lags behind the best-performing LLM by 11.7%. These results reveal that long-context DU is a far-from-resolved task for current LVLMs.

## 2    Related Work

**Benchmarks for Document Understanding.** A great amount of datasets have emerged to evaluate the DU capabilities of LVLMs. Many datasets focus exclusively on either a single component (*e.g.,* table, chart) [13; 15; 21; 22] or a single page [12; 14] from the full documents. Some recent DU datasets [16; 17; 18; 23; 19] attempt to assess multi-page documents, but still exhibit shortcomings in terms of document length (page number), information density (token number) and the construction approaches. Specifically, MP-DocVQA [16] is an extension of DocVQA [12] and inherently absent of both cross-page and unanswerable questions. Annotating from scratch, DUDE [17] includes a small percentage of cross-page questions (2.1%) and unanswerable questions (12.7%). However, due to the relatively short context length (5.3 pages on average) and the use of crowd-sourced annotations, questions in DUDE tend to be less challenging and somewhat less rigorous. SlideVQA features 20-page documents and cross-page questions (12.9%). Nevertheless, the documents in SlideVQA are in slide-deck format and of relatively low information density. Moreover, these cross-page questions are HotpotQA-style [24] created by instantiating entity graphs and co-referencing in-graph entities across multiple pages. The entity graph from a closed document tends to be sparse and has significant shortcuts (see examples in Appendix A.4). These shortcuts sometimes lead to false cross-page questions that actually do not require answer evidence across different pages. The recent FinanceBench [19] features both extremely long-context documents and practical, scalable cross-page questions. However, its documents are exclusively financial reports. Additionally, the reference answers are in open-ended formats, making the expert-level manual evaluation indispensable. The above reasons limit the broader applicability of FinanceBench. To our best knowledge, MMLONGBENCH-DOC is the first comprehensive, qualified, and easy-to-use benchmark on the long-context DU task. More detailed descriptions and comparisons are presented in Table 1.

**Models for Document Understanding.** There are two main branches of models for automatic DU tasks. The first approach employs two-stream, OCR-dependent architectures to separately encode textual information (parsed via OCR) and visual information (images and/or layout structures) [25; 26; 27]. In contrast, the second approach develops OCR-free models that understand documents

---

[2]We view website screenshots and posters as generalized documents and define *equivalent page number* (EPN) to measure their context lengths: $\texttt{EPN(D)} = \texttt{ceil}(\frac{\texttt{Pixel(D)}}{P})$. Here $\texttt{Pixel(D)}$ is the pixel number of generalized document D, and P is the average pixel numbers of each page (converting from .pdf to .png format with resolution 240) in MMLONGBENCH-DOC.

in an end-to-end manner [28; 29]. With the rapid advancement of LVLMs, the latter approach has dominated the current DU solutions. As mentioned above, a range of LVLMs demonstrate promising performance on single-page DU datasets. However, as shown in Section 4, even the most advanced LVLMs fall significantly short of achieving satisfactory performance on our benchmark. It reveals that understanding lengthy documents still poses great challenges to current LVLMs.

**Long-context LVLMs and LLMs.**   Lengthy documents necessitate the use of LVLMs or LLMs with extended context sizes. Several benchmarks [30; 31; 32; 33] and solutions [34; 35; 36; 37] have been proposed to evaluate and develop long-context LLMs. However, there exists limited related work for long-context LVLMs, leaving this area largely unexplored. Until very recently, contemporary studies [38; 39; 40] assess and/or improve LVLMs' multi-image understanding capabilities. Evaluations on both MMLONGBENCH-DOC and these works indicate that current LVLMs are still not fully equipped to handle long-context DU and many other practical tasks that require extensive contextual comprehension.

## 3   MMLONGBENCH-DOC

We design a three-stage annotation pipeline for the construction of our benchmark. The three stages will be introduced in Section 3.1, Section 3.2, and Section 3.3, respectively. We also provide key statistics of our benchmark in Section 3.4.

### 3.1   Document Collection

As a long-context DU benchmark, the documents shall be of diverse topics and lengthy enough. To this end, we crawl a great amount of documents from various sources. Then we select the lengthy ones from these documents. Specifically, we encompass a diverse array of documents from two approaches. (1) **Existing documents** from four previous datasets: DUDE [17], SlideVQA [18], ChartQA [13], and FinanceBench [19]. (2) **Newly-collected documents** from Arxiv [3], ManualsLib [4] and Google Search [5]. Then we (1) filter out the documents with fewer than 15 pages or license restrictions and (2) down-sample documents from DUDE, SlideVQA, and FinanceBench for a more balanced distribution. Detailed descriptions of our selection and processing procedure can be found in Appendix A.1 and Appendix A.2.

In summary, we collect a total of 135 documents. Among them, 76 documents are from existing datasets and incorporate previously annotated questions (represented as triangles). The remaining 59 documents are newly collected and incorporate no existing questions. We manually categorize them into 7 types: *Research Report*, *Financial Report*, *Academic Paper*, *Brochure*, *Guideline*, *Administration & Industry File*, *Tutorial / Workshop*. We showcase some instances of these documents in Appendix A.3.

### 3.2   Question and Answer Collection

To serve as a high-quality and comprehensive benchmark, the question annotation of our benchmark adheres to the following standards: (1) All questions shall be neither over-easy nor over-difficult. (2) Questions are not repetitively derived from the same page or the same pattern. (3) The distribution of evidence numbers, evidence sources, and evidence locations for the questions shall be balanced. (4) No questions shall be answered correctly without accessing the relevant documents.

Ten authors serve as expert-level annotators for the question-and-answer collection. All of them are doctors or Ph.D. students proficient in English reading and writing. Before formal annotation, they undergo a training session and pre-annotate three documents for practice. We iteratively review their annotation results and provide personalized feedback until their annotations meet the standards mentioned above. Regarding the formal annotation, we divide 135 documents into 54 batches (each having 2-4 documents) and dispatch these batches to annotators. We then ask the annotators to submit their results in units of batches and set reasonable time intervals for each batch's submission. We

---

[3]`https://arxiv.org`
[4]`https://www.manualslib.com`
[5]`https://www.google.com.sg`

timely evaluate their annotations after each submission and remind the annotators if their questions in this turn diverge from the standards. It avoids the annotators rushing all assignments in a short time and benefits the annotation quality. We recommend the annotators take 60-90 minutes on each document. Specifically, the annotators shall rapidly read through the whole document in the first 15-30 minutes. For the remaining time, they shall dive deep into specific components to modify existing annotations and/or add new annotations as detailed below.

**Modify Existing Questions.** Documents collected from existing datasets had been annotated with some questions and answers from previous work. However, their crowd-sourcing annotations inevitably make some questions, answers, and other meta information unqualified. Therefore, we edit their annotations before including them as a component of our benchmark.

Specifically, we classify six potential problems in original annotations: *Wrong Answers or Evidence Pages*, *Repetitive Question*, *Ambiguous Question*, *Decontextualization-required Question*, *Low Document-relevant Question* and *Potential Shortcut*. See detailed explanations and examples about these problems in Appendix A.4. Given an existing document, the annotators are tasked to evaluate each existing question's quality according to whether they have one or more above problems and assign a label from {Retain, Revise, Remove} for each question. Then the annotators would revise the Revise questions to meet our quality criteria and remove the Remove questions. Among all 425 original questions from 76 existing documents, 32.2% of them are revised and 46.1% are removed. We finally collect 211 questions in this procedure. The corresponding GUI is shown in Appendix A.7.

**Add New Questions.** We newly annotate questions on both existing and newly collected documents to expand the questions in our benchmark. Specifically, we ask annotators to add about 3 questions on existing documents, and 6 questions on newly-collected documents. Given most existing questions (even after editing) are single-page ones and sourced from texts, we put more focus on (1) cross-page and unanswerable questions and (2) questions sourced from tables, charts, and images for newly added questions to balance the distribution. We detail the quantitative requirements in Appendix A.5. Associated with questions, annotators also provide reference answers and meta-information (*i.e.,* evidence sources, answer format, evidence locations) for all samples. We finalized a collection of 965 samples in this procedure. The corresponding GUI is shown in Appendix A.7.

## 3.3 Quality Control

Combining the merits of humans and LVLMs, we adopt a three-round, semi-automatic quality control procedure to improve the annotation quality of our benchmark. We detail each round in the following components and leave the discussion of potential bias in Appendix A.6.

**Document-relevant Detection.** Our benchmark is designed to evaluate LVLMs' long-context document understanding abilities. All questions are expected to be unanswerable without access to corresponding documents. To remove low document-relevant questions (*i.e.,* questions not relying on documents), we feed each annotated question **WITHOUT** documents to GPT-4o. A question will be identified as *low document-relevant* question if GPT-4o correctly predicts under this case. Ultimately, 94 samples are identified as low document-relevant questions and removed in this round.

**Self-reflection.** We draw inspirations from MMBench [41] and leverage LVLMs to reduce the wrongly-annotated samples. Specifically, we feed the remaining questions from the last round **WITH** their documents to GPT-4o. Samples whose model predictions are inconsistent with the reference answers are sent back to corresponding annotators. The annotators are asked to check each question and identify whether the inconsistency is caused by *problematic annotation* or not. As a result, 13.8% of the samples are identified as problematic annotations. The annotators revise them accordingly.

**Cross-checking.** In parallel, annotators cross-check the annotated samples from other annotators and determine the inconsistency reasons the same as described above. We calculate Cohen's kappa value of their identifications as 0.42 (17.5% inconsistent samples), showing a moderate agreement. Regarding the 17.5% inconsistent samples, two primary authors serve as meta-annotators and make final decisions on them (and if necessary, revise accordingly).

## 3.4 Dataset Overview and Analysis

The main statistics of MMLONGBENCH-DOC are presented in Table 2. Overall, our benchmark consists of 1,082 questions. These questions are constructed upon 135 lengthy documents across 7

| Statistic | Number |
|---|---|
| **Documents** | 135 |
| - Type | 7 |
| - Average/Medium pages | 47.5 / 28 |
| - Average/Medium length | 21,214.1 / 12,179 |
| **Total questions** | 1,082 |
| - Single-page question | 494 (45.7%) |
| - Cross-page questions | 365 (33.7%) |
| - Unanswerable questions | 223 (20.6%) |
| - Derived questions | 184 (17.0%) |
| - Newly-annotated questions | 898 (83.0%) |
| (Evidence source) | |
| - Pure-text | 305 (35.5%) |
| - Layout | 119 (13.9%) |
| - Table | 218 (25.4%) |
| - Chart | 178 (20.7%) |
| - Image | 304 (35.4%) |
| (Answer Format) | |
| - String | 250 (29.1%) |
| - Integer | 299 (34.8%) |
| - Float | 159 (18.5%) |
| - List | 151 (17.6%) |
| Avg./Max. question length | 16.4 / 60 |
| Avg./Max. answer length | 2.8 / 54 |

Table 2: Dataset Statistics

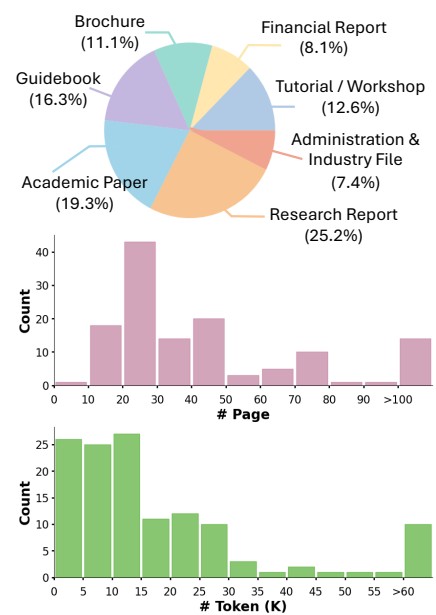

Figure 2: Detailed distribution of documents. **Top**: Document type. **Middle**: Page Number. **Bottom**: Token Number.

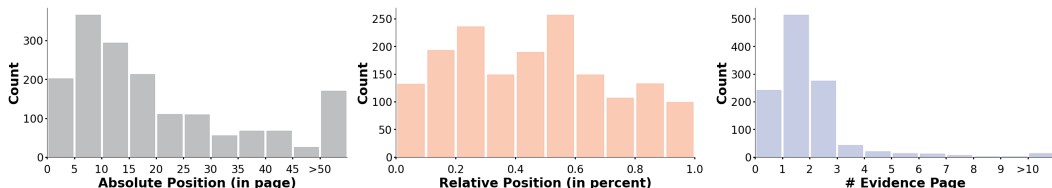

Figure 3: Detailed distribution of questions & answers. **Left**: Absolute position of answer evidences (the page index). **Middle**: Relative position (the page index/document page number). **Right**: Evidence page number of each question. (0: unanswerable question; >2: cross-page question).

document types, with an average of 47.5 pages and 21,214.1 tokens. Please see detailed distributions of these documents in Figure 2. Regarding the questions, there are 494 single-page questions (1 evidence page), 365 cross-page questions (2+ evidence pages), and 223 unanswerable questions (no evidence page). These three types of questions evaluate the LVLMs's long-context DU capabilities from complementary aspects: the localization ability, the cross-page comprehension ability, and the hallucination severity, respectively. For single-page and cross-page questions, their answer evidence is scattered among different context sources (*i.e.,* text, layout, table, chart, image) and evenly distributed across different locations of the documents (see Table 2, Figure 3 Left and Middle). Also notably, 28.6% of cross-page questions have more than two evidence pages, which further enhances the challenge of our benchmark.

# 4 Evaluation

## 4.1 Evaluation Protocol

We follow MATHVISTA [56] to conduct a three-step evaluation protocol: *response generation*, *answer extraction*, and *score calculation*. We adopt such a protocol out of three considerations: (1) Current LVLMs are instructed to generate long responses, rather than short-form answers, in conventional settings. (2) The evaluation of long responses, however, remains an open and challenging problem. (3) We focus on the document understanding (not instruction following) abilities of LVLMs.

Table 3: **Evaluation of various models on MMLONGBENCH-DOC.** We report the generalized accuracy of five types of evidence sources including pure text (TXT), layout (LAY), chart (CHA), table (TAB), and image (IMG). We also present the generalized accuracy of questions categorized by the number of evidence pages: single-page (SIN), cross-page (MUL), and unanswerable (UNA) questions. The best and second-best performance in each section are highlighted.

| Model | #Param | Context Window | Evidence Source | | | | | Evidence Page | | | ACC | F1 |
|---|---|---|---|---|---|---|---|---|---|---|---|---|
| | | | TXT | LAY | CHA | TAB | FIG | SIN | MUL | UNA | | |
| *OCR (Tesseract [42]) + Large Language Models (LLMs)* | | | | | | | | | | | | |
| *Open-source Models* | | | | | | | | | | | | |
| ChatGLM-128k [37] | 6B | 128k | 23.4 | 12.7 | 9.7 | 10.2 | 12.2 | 18.8 | 11.5 | 18.1 | 16.3 | 14.9 |
| Mistral-Instruct-v0.2 [43] | 7B | 32k | 19.9 | 13.4 | 10.2 | 10.1 | 11.0 | 16.9 | 11.3 | 24.1 | 16.4 | 13.8 |
| Mixtral-Instruct-v0.1 [44] | 8x7B | 32k | 24.2 | 14.8 | 12.5 | 15.0 | 13.7 | 21.3 | 14.1 | 13.1 | 17.0 | 16.9 |
| Mixtral-Instruct-v0.1 [44] | 8x22B | 64k | 34.2 | 21.3 | 19.5 | 21.3 | 19.2 | 27.7 | 21.9 | 32.4 | 26.9 | 24.7 |
| *Proprietary Models* | | | | | | | | | | | | |
| QWen-Plus [45] | - | 32k | 17.4 | 15.6 | 7.4 | 7.9 | 8.8 | 14.2 | 10.6 | 42.2 | 18.9 | 13.4 |
| DeepSeek-V2 [46] | - | 32k | 27.8 | 19.6 | 8.8 | 17.0 | 9.4 | 20.2 | 15.4 | 48.1 | 24.9 | 19.6 |
| Claude-3 Opus [4] | - | 32k | 30.8 | 30.1 | 16.4 | 24.4 | 16.3 | 32.0 | 18.6 | 30.9 | 26.9 | 24.5 |
| Gemini-1.5-Pro [3] | - | 32k | 29.3 | 15.9 | 12.5 | 17.7 | 11.5 | 21.2 | 16.4 | 73.4 | 31.2 | 24.8 |
| GPT-4-turbo [47] | - | 128k | 36.5 | 21.0 | 20.7 | 24.3 | 17.3 | 28.7 | 23.8 | 31.2 | 27.6 | 25.9 |
| GPT-4o [2] | - | 128k | 41.1 | 23.4 | 28.5 | 38.1 | 22.4 | 35.4 | 29.3 | 18.6 | 30.1 | 30.5 |
| *Large Visual Language Models (LVLMs)* | | | | | | | | | | | | |
| *Open-source, 7-14B Models* | | | | | | | | | | | | |
| DeepSeek-VL-Chat [48] | 7.3B | 4k | 7.2 | 6.5 | 1.6 | 5.2 | 7.6 | 5.2 | 7.0 | 12.8 | 7.4 | 5.4 |
| Idefics2 [49] | 8B | 8k | 9.0 | 10.6 | 4.8 | 4.1 | 8.7 | 7.7 | 7.2 | 5.0 | 7.0 | 6.8 |
| MiniCPM-Llama3-V2.5 [50; 51] | 8B | 2k | 11.9 | 10.8 | 5.1 | 5.9 | 12.2 | 9.5 | 9.5 | 4.5 | 8.5 | 8.6 |
| InternLM-XC2-4KHD [5] | 8B | 16k | 9.9 | 14.3 | 7.7 | 6.3 | 13.0 | 12.6 | 7.6 | 9.6 | 10.3 | 9.8 |
| mPLUG-DocOwl 1.5 [52] | 8.1B | 4k | 8.2 | 8.4 | 2.0 | 3.4 | 9.9 | 7.4 | 6.4 | 6.2 | 6.9 | 6.3 |
| Qwen-VL-Chat [53] | 9.6B | 6k | 5.5 | 9.0 | 5.4 | 2.2 | 6.9 | 5.2 | 7.1 | 6.2 | 6.1 | 5.4 |
| Monkey-Chat [54] | 9.8B | 2k | 6.8 | 7.2 | 3.6 | 6.7 | 9.4 | 6.6 | 6.2 | 6.2 | 6.2 | 5.6 |
| *Open-source, >14B Models* | | | | | | | | | | | | |
| CogVLM2-LLaMA3-Chat [9] | 19B | 8k | 3.7 | 2.7 | 6.0 | 3.2 | 6.9 | 3.9 | 5.3 | 3.7 | 4.4 | 4.0 |
| InternVL-Chat-v1.5 [6] | 26B | 4k | 14.0 | 16.2 | 7.1 | 10.1 | 16.6 | 14.9 | 12.2 | 17.5 | 14.6 | 13.0 |
| EMU2-Chat [55] | 37B | 2k | 6.1 | 9.7 | 2.6 | 3.8 | 7.7 | 5.7 | 6.1 | 16.5 | 8.3 | 5.5 |
| *Proprietary Models* | | | | | | | | | | | | |
| Claude-3 Opus [4] | - | 200k | 24.9 | 24.7 | 14.8 | 13.0 | 17.1 | 25.6 | 13.8 | 7.6 | 17.4 | 18.1 |
| Gemini-1.5-Pro [3] | - | 128k | 21.0 | 17.6 | 6.9 | 14.5 | 15.2 | 21.1 | 11.1 | 69.2 | 28.2 | 20.6 |
| GPT-4V(ision) [47] | - | 128k | 34.4 | 28.3 | 28.2 | 32.4 | 26.8 | 36.4 | 27.0 | 31.2 | 32.4 | 31.2 |
| GPT-4o [2] | - | 128k | 46.3 | 46.0 | 45.3 | 50.0 | 44.1 | 54.5 | 41.5 | 20.2 | 42.8 | 44.9 |
| *Human Baseline* | | | | | | | | | | | | |
| Human Experts | - | - | - | - | - | - | - | - | - | - | 65.8 | 66.0 |

Specifically, we impose no limitations on *response generation* stage to encourage LVLMs to answer the questions in a freestyle. Then we propose a unified LLM-based *answer extractor* (GPT-4o under our setting) to convert their long responses to short-form answers. Finally, we use a rule-based *score calculator* to evaluate the converted short answers. We report both generalized accuracy and generalized F1 score to balance the answerable (positive) and unanswerable (negative) questions. The used prompt, the high correlation between our automatic *answer extractor* and human evaluation, and the detailed rules of our *score calculation* are described in Appendix B.

## 4.2 Experimental Setup

We evaluate 14 LVLMs on MMLONGBENCH-DOC, including 4 proprietary LVLMs and 10 open-source LVLMs. To purely evaluate LVLMs' long-context DU abilities, we screenshot each page of the PDF-formatted document with 144 DPI and feed all these PNG-formatted images to LVLMs in an end-to-end approach. Notably, all evaluated open-source LVLMs do not support multi-image inputs or present significant performance drops when fed with excessive images (*e.g.,* more than 10 or 20 images). Therefore, we employ a concatenation strategy that combines all screenshot pages into 1 or 5 images and feeds these concatenated images to open-source LVLMs. Regarding proprietary LVLMs, we adopt the same concatenation strategy and reduce the image number to 20 for Claude-3-Opus to fit its maximum image threshold. For GPT-4o, GPT-4V, and Gemini-1.5-Pro, we directly send all original screenshots to them (*i.e.,* the image number equals the page number).

For comparison, we also use the Tesseract [42] OCR model to recognize and extract texts from the documents and feed the parsed documents to 10 LLMs, including 6 proprietary and 4 open-source

ones. Texts exceeding their context lengths are truncated. Notably, as a key component of the classical solution for the DU task, the OCR model can handle most flattened texts and some structured tables in the document. However, it cannot perceive the information from the charts or images. Thus the TXT-formatted, OCR-parsed documents are lossy documents in which the information is not fully preserved. More detailed hyperparameters are introduced in Appendix B.5. Additionally, we also conduct manual evaluation on a subset of our datasets (238 questions from 29 documents) to indicate the difficulty of this task for humans.

## 4.3 Main Results

We compare the performance of different LVLMs and LLMs in Table 3, reporting their generalized accuracy and F1 scores (shown in the last two columns). Regarding LVLMs, we draw several conclusions as below: (1) The performance demonstrates that long-context DU is still a challenging and unsolved task for current LVLMs. The best-performing LVLM, GPT-4o, merely achieves a 44.9% F1 score. The second best-performing LVLM, GPT-4V, lags behind by over 10% percent and presents a 31.4% F1 score. All other LVLMs only achieve about 20% or even lower F1 scores. (2) Though far from satisfactory, GPT-4o performs much better than all other models (including GPT-4V). Thus we speculate that the multi-modal pre-training paradigm significantly benefits LVLMs' cross-modality understanding capabilities. (3) Proprietary LVLMs perform better than open-source LVLMs by a large margin. We attribute it to the difference of acceptable image numbers: open-source LVLMs only support single-image or several-image inputs, while proprietary LVLMs can be fed with at least 20 images or even more. Given that lengthy documents have tens of even hundreds of pages, it is impractical for open-source LVLMs to accurately perceive the information in the documents from the excessively concatenated images. (4) The performances of different models are highly correlated with their acceptable image numbers and maximum image resolutions. Notably, open-source LVLMs that support high-resolution images (*i.e.,* InternLM-XC2-4KHD and InternVL-Chat-v1.5) exhibit superior performance compared to those with lower resolution limits.

Surprisingly, LVLMs even demonstrate overall worse performance than LLMs, even LLMs are fed with lossy OCR-parsed documents. Specifically, Gemini-1.5-Pro and Claude-3 Opus have 4.2% and 6.4% absolute F1-score degradations on vision versions. And the best-performing LLM (Mixtral) also surpasses the best-performing LVLM (InternVL-v1.5) by 11.7%. The above results clearly reveal that most current LVLMs are still not proficient in cross-modality, long-context document understandings. It is promising that GPT-4o and GPT-4-turbo achieve better performance when seeing multi-modality PDF documents than parsed text by 14.4% and 5.3% F1-score, respectively. Their performances validate the feasibility, benefit, and necessity of understanding documents in an end-to-end, cross-modality approach. We speculate that the scarce related pre-training corpus (*i.e.,* extremely multi-image or lengthy documents) hinders the long-context DU capabilities of other LVLMs. We will leave related explorations for future work.

Regarding the human evaluation, we observe 66.0% F1-score from our annotators and a significant performance gap (exceeding 20% in absolute) between the current LVLMs and humans. This gap highlights the challenges of document understanding for LVLMs and the necessity of our benchmark.

## 4.4 Fine-grained Results.

**Document Type.** As illustrated in Figure 4, LVLMs and LLMs exhibit distinct performance patterns across various document types. Our findings include: (1) All evaluated models demonstrate decent performance on industrial documents, which tend to have more standardized formats and less non-textual information. (2) The GPT series and Mixtral (*i.e.,* the SoTA open-source LLM) show relatively balanced performance across different document types. In contrast, other models perform significantly worse in specialized domains such as academic papers and financial reports. (3) When equipped with OCR, LLM-based models like GPT-4 and Mixtral achieve comparable or even superior performance on industrial documents, academic papers, and brochures. Conversely, end-to-end LVLMs outperform OCR+LLMs in areas such as tutorials, research reports, and guidelines. We speculate that comprehending these latter document types requires more extensive multi-modal information, from which LVLMs significantly benefit.

**Evidence Source.** We categorize questions based on their evidence sources and present fine-grained results in Figure 4 and Table 3. Our observations reveal that only GPT-4o exhibits relatively balanced

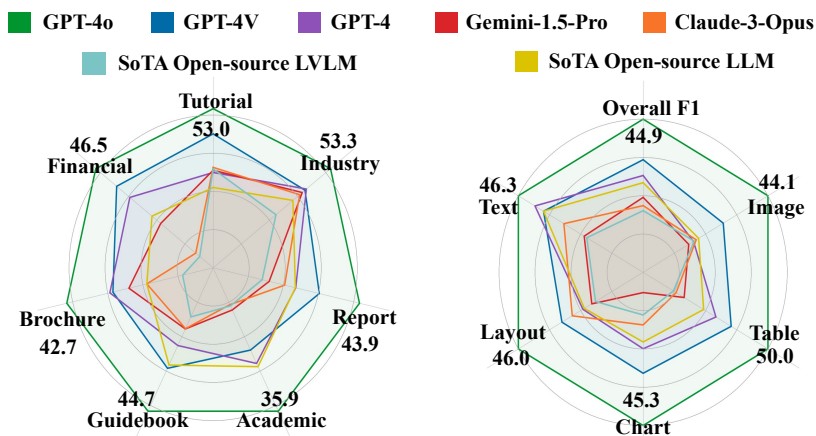

Figure 4: Fine-grained results on various document types and evidence sources.

performance across the different sources. Other LVLMs, however, show inferior performance on questions related to charts and/or images compared to those related to text and/or layout. Additionally, LLMs generally demonstrate better or comparable performance to LVLMs on text- and table-related questions but show worse performance on questions involving other elements. This highlights the limitations of OCR (and other PDF parsers) when dealing with charts and images, as well as the gap in OCR capabilities between LVLMs and pure-text LLMs.

**Evidence Position.** We also examine how the evidence locations (*i.e.,* the page indexes where the answer evidence is found) affect model performance. The results shown in Figure 5 reinforce that MMLONGBENCH-DOC poses significant challenges for current models, at least partially due to the extended length of the documents. Almost all models (except InternVL-v1.5) exhibit their best performance on questions derived from the initial pages, while their performance declines progressively as the page index increases. Interestingly, two proprietary models, Gemini-Pro-1.5 and Claude-3-Opus, experience particularly sharp declines in performance.

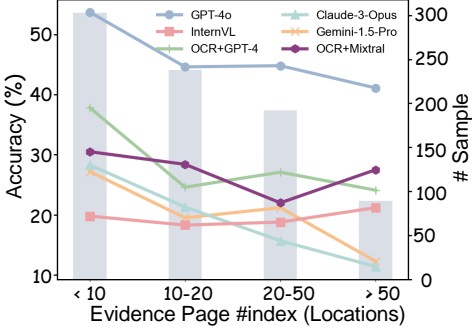

Figure 5: Relationships between evidence positions and model performances.

**Number of Evidence Page.** We observe a consistent trend that all models achieve higher scores on single-page questions than cross-page questions. It reveals that gathering and reasoning over all necessary information across different pages is not trivial for current LVLMs and LLMs. More interestingly, evaluated LVLMs behave differently on unanswerable questions. GPT-4o and Claude-3 Opus adopt more aggressive strategies and usually tend to provide some answers. It makes their answers more likely helpful, but also increases the risk of hallucination and unfaithfulness (see their scores on unanswerable questions are much lower than answerable questions). On the contrary, Gemini-1.5-Pro, DeepSeek-VL-Chat, and EMU2-Chat are much more cautious and tend to refuse to answer questions about which they are uncertain. It makes their answers safer but less helpful (with large amounts of responses like *I don't know*).

## 5  Analysis & Discussion

### 5.1  Oracle Setting

We conduct additional experiments to explore to what extent the challenges of MMLONGBENCH-DOC are caused by the long-context lengths of documents. Specifically, we feed 820 answerable questions along with their oracle evidence pages (instead of the whole documents) to three representative LVLMs and show results in Figure 6. On one hand, it indicates that long-context length is a

significantly challenging factor for document understanding. Compared with the oracle-page setting, lengthy documents lead to more than 20% absolute performance degradation on Gemini-1.5-Pro and InternLM-XC2-4KHD. Regarding the single-page questions, the performance difference even achieves up to 30%. On the other hand, the overall performance achieves only about 40% and 30% for Gemini-1.5-Pro and InternLM-XC2-4KHD even under oracle-page setting. And the improvement for GPT-4o is much less (about 10%). It demonstrates that the development of long-context LVLMs can largely facilitate, though still can not fully solve, the long-context DU task.

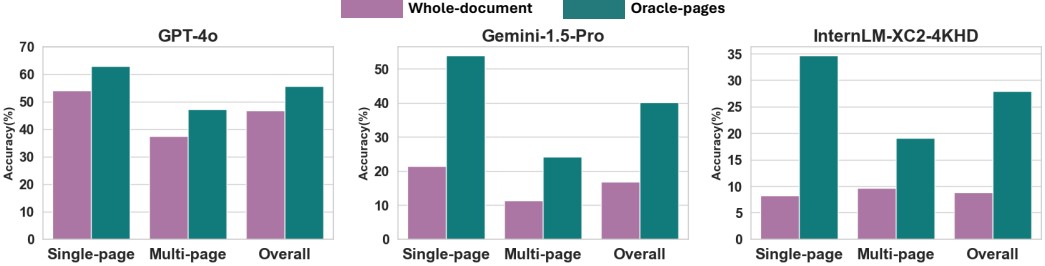

Figure 6: Performance comparisons between normal setting (feeding models with the whole documents) and oracle setting (feeding models only with the evidence pages) among three LVLMs.

## 5.2 Error Analysis

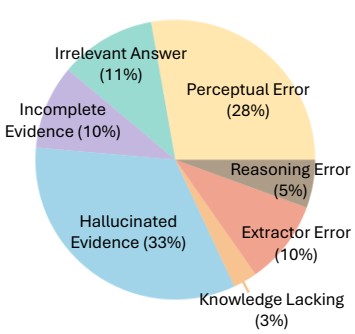

Figure 7: Error distribution

We further conduct error analysis to understand the bottleneck of current LVLMs in a qualitative approach. Specifically, we randomly select 72 error predictions from GPT-4o's responses and manually check their error reasons. These errors are categorized into seven types: *Perceptual Error*, *Irrelevant Answer*, *Incomplete Evidence*, *Hallucinated Evidence*, *Extractor Error*, *Reasoning Error* and *Knowledge Lacking*. The distribution of these errors is illustrated in Figure 7. It indicates that most errors come from the model's hallucination (*i.e.,* wrong explanations and answers to unanswerable questions) and perceptual errors (mainly in visual contexts). Additionally, GPT-4o sometimes misunderstands the intent of questions and provides irrelevant responses. The errors caused by collecting incomplete evidence (for cross-page questions) are also unignorable. The descriptions and examples of these error types are detailed in Appendix C.1.

## 6 Conclusion

In this work, we present MMLONGBENCH-DOC to evaluate the long-context DU capabilities of LVLMs. Extensive experiments on 14 LVLMs (and 10 LLMs for comparison) reveal that the understanding of lengthy documents poses great challenges to current LVLMs. Even though the performance of GPT-4o proves the benefit of end-to-end, multi-modality perception for DU tasks, most LVLMs struggle on long visual contexts (*i.e.,* extremely multiple images) and show inferior performance compared to OCR+LLM pipelines. We hope that the construction of our benchmark could push forward the development of more powerful LVLMs on lengthy document understanding.

## Acknowledgements

This study is supported under the RIE2020 Industry Alignment Fund – Industry Collaboration Projects (IAF-ICP) Funding Initiative, as well as cash and in-kind contribution from the industry partner(s). This work is also supported by Shanghai Artificial Intelligence Laboratory, the National Key R&D Program of China (2022ZD0160201).

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

# A  Benchmark Construction Details

## A.1  Existing Document Collection

Although previous datasets contain a relatively small proportion of lengthy documents, their absolute quantity should not be disregarded. Therefore, we compile lengthy documents from various datasets to include them as part of the documents in this benchmark. Specifically, we review and consider 21 previous document understanding (DU) datasets, and ultimately select 4 of them for further document selection. The selection reasons are shown in Table 4. All of these four datasets are licensed under the Creative Commons license (CC-BY) or other open-source licenses. Regarding the 4 selected datasets: DUDE [17], SlideVQA [18], ChartQA [13] and FinanceBench [19], we collect a total of 76 documents and detail our collection procedures as below.

Table 4: Comparison of selected and considered datasets for our benchmark.

| Dataset | Selected | Comment |
|---|---|---|
| DUDE [17] | ✓ | - |
| SlideVQA [18] | ✓ | - |
| ChartQA [13] | ✓ | - |
| FinanceBench [19] | ✓ | - |
| DocVQA [12] | ✗ | Repetitive with some documents/questions in DUDE; Single-page documents only |
| MP-DocVQA [16] | ✗ | Repetitive with some documents/questions in DUDE; Single-page questions only |
| Kleister Charity [57] | ✗ | Repetitive with some documents/questions in DUDE; Over-simple |
| Kleister NDA [57] | ✗ | Repetitive with some documents/questions in DUDE; Over-simple |
| DeepForm [58] | ✗ | Repetitive with some documents/questions in DUDE; Over-simple |
| FUNSD [59] | ✗ | Repetitive with some documents/questions in DUDE; Over-simple |
| SROIE [60] | ✗ | Repetitive with some documents/questions in DUDE; Over-simple |
| Infograohics VQA [14] | ✗ | Infographs are not long-context documents |
| TAT-QA [15] | ✗ | Repetitive with some documents/questions in FinanceBench |
| PWC [22] | ✗ | Repetitive with our self-annotated questions from academic papers |
| PaperQA [56] | ✗ | Repetitive with our self-annotated questions from academic papers |
| TextbookQA [61] | ✗ | Low document-relevance; Over-simple |
| PlotQA [62] | ✗ | Repetitive with our self-annotated questions from academic papers and research reports |
| VisualMRC [63] | ✗ | Human performance reached; Website screenshots are not long-context documents |
| WebSRC [64] | ✗ | Human performance reached; Website screenshots are not long-context documents |
| VisualWebBench [21] | ✗ | Human performance reached; Website screenshots are not long-context documents |
| PDFTriage [23] | ✗ | Not publicly available |

**DUDE:** We first filter all documents over 15 pages in the validation set of the original dataset, resulting in 87 documents. From these, we randomly sample 23 to include as a component of our benchmark documents.

**SlideVQA**: We download slide decks in the test set by following the instructions in the original repository [6]. Pursuing lengthy documents, we slightly modified the code to remove the 20-page truncation procedure. Then we randomly select 27 slide decks for our benchmark documents.

**FinanceBench**: We randomly sample 5 financial reports from the test set.

**ChartQA**: Different from the above three datasets, ChartQA only contains chart screenshots cropped from documents. We take the following steps to recover these original documents: (1) We use the Tesseract OCR model [42] to recognize the text within the charts. (2) We use these texts as keywords to search for related documents on Google Search. (3) We manually identify these documents and remove all those that are less than 15 pages. From the ChartQA test set, we finalize a collection of 53 research reports from the Pew Research Center. We randomly sample 18 of these documents to include as a component of our benchmark documents.

---

[6]https://github.com/nttmdlab-nlp/SlideVQA

## A.2 Newly-annotated Document Collection

Most documents collected from previous datasets are *Industrial Files*, *Tutorial & Workshop*, *Finance Report* and *Research Report*. To diversify our benchmark, we additionally collect 59 documents including *Academic Paper*, *Brochure*, and *Guideline*. We detail the collection procedures as below.

**Academic Paper** We collect 24 academic papers from Arxiv. All selected papers are over 15 pages (including references and appendix). To ensure annotation quality, each paper is either written or thoroughly read by at least one of the annotators.

**Guideline and Brochure** We collect 21 guidelines and 14 brochures from either ManualsLib or Google Search, covering diverse topics such as school, company, institution, products, service *etc.*. Each document is manually reviewed by one corresponding annotator and other primary authors to ensure its availability for academic use [7].

## A.3 Document Examples

As stated in Section 2.1, the documents in MMLONGBENCH-DOC can be categorized into seven types. We show the examples of each type as below.

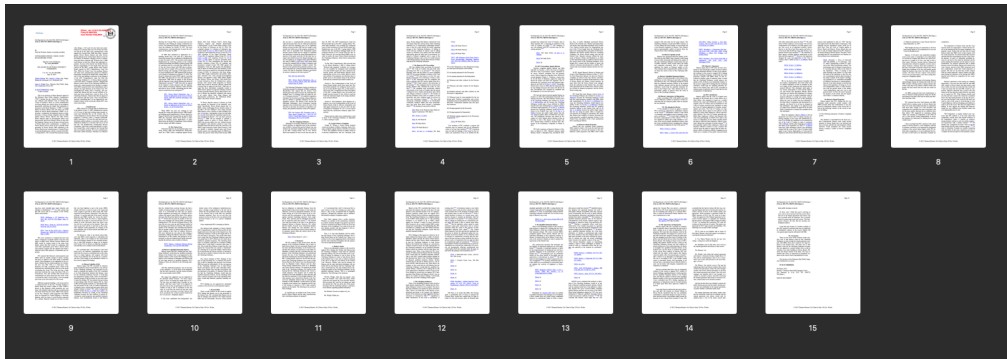

Figure 8: Document example about **Administration & Industrial File**

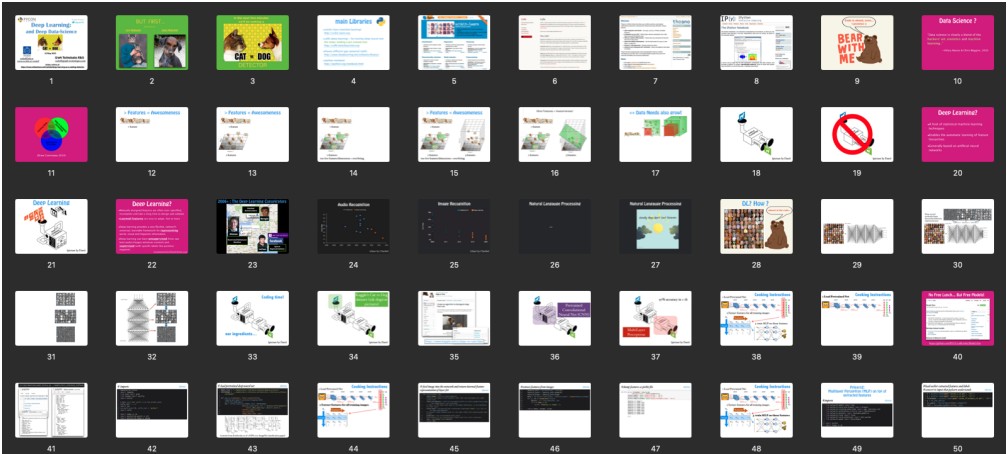

Figure 9: Document example about **Tutorial & Workshop** (only show first 50 pages)

---

[7]Should any authors request the removal of their documents, we will promptly comply.

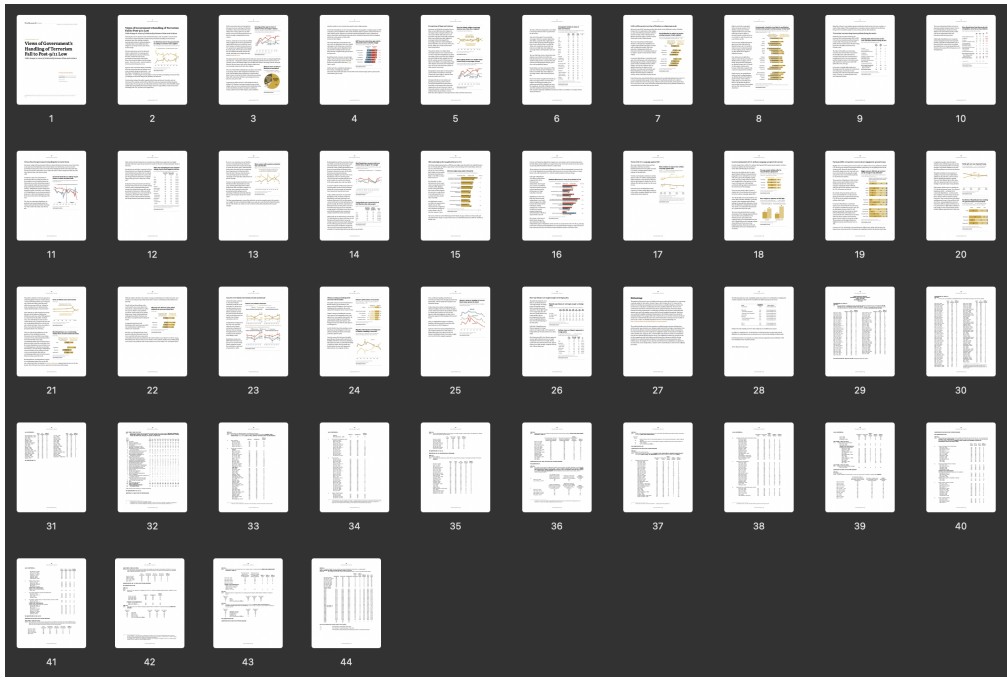

Figure 10: Document example about **Research Report**

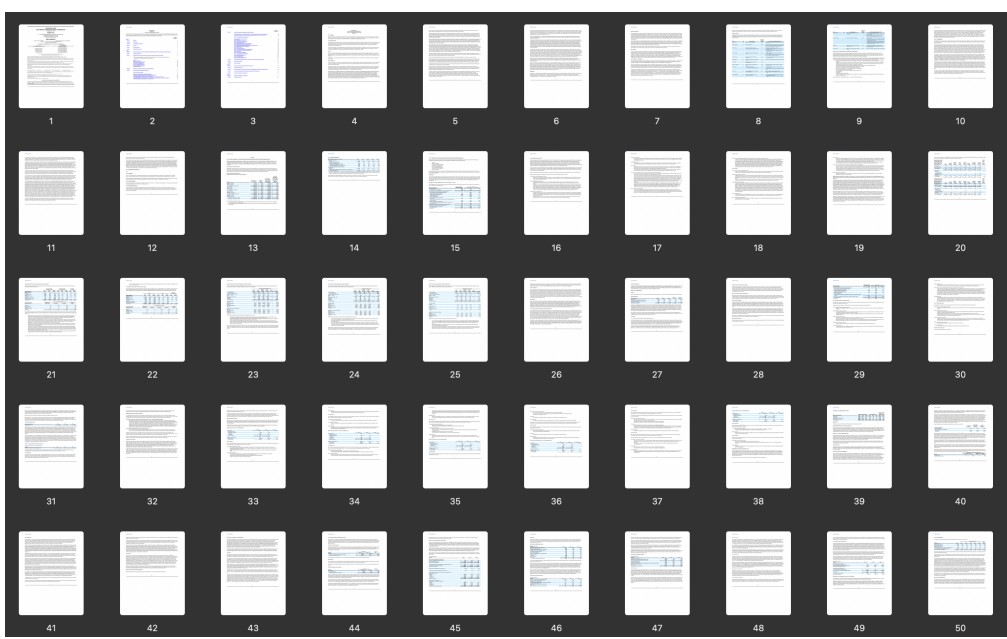

Figure 11: Document example about **Financial Report** (only show first 50 pages)

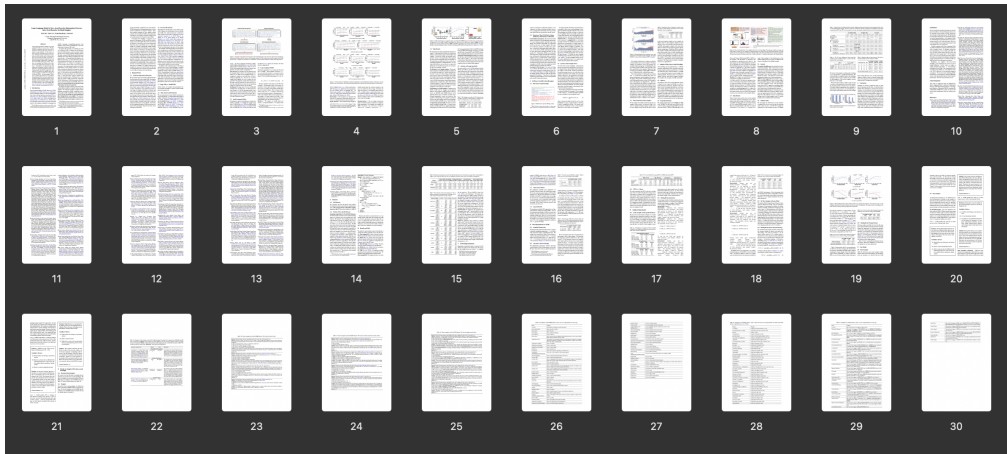

Figure 12: Document example about **Academic Paper**

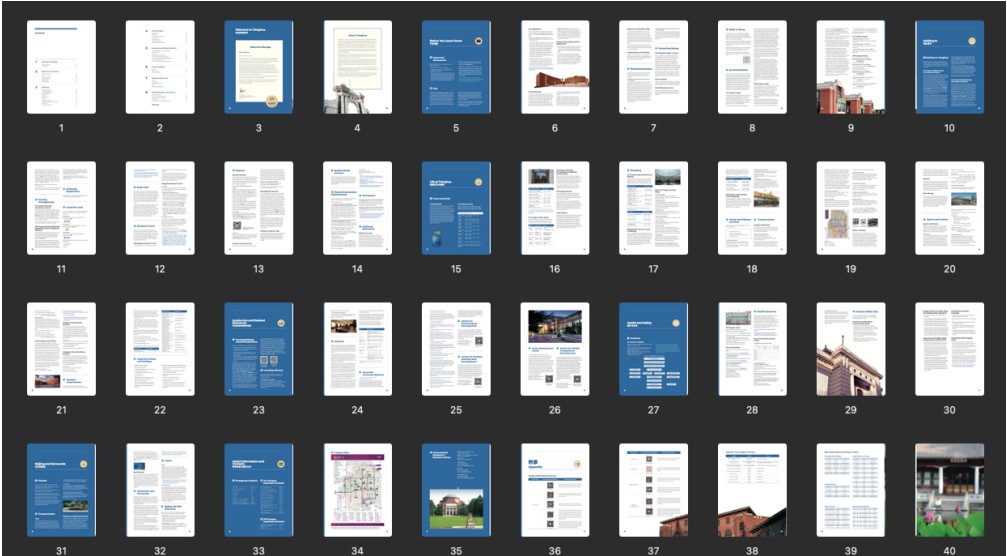

Figure 13: Document example about **Guidebook**

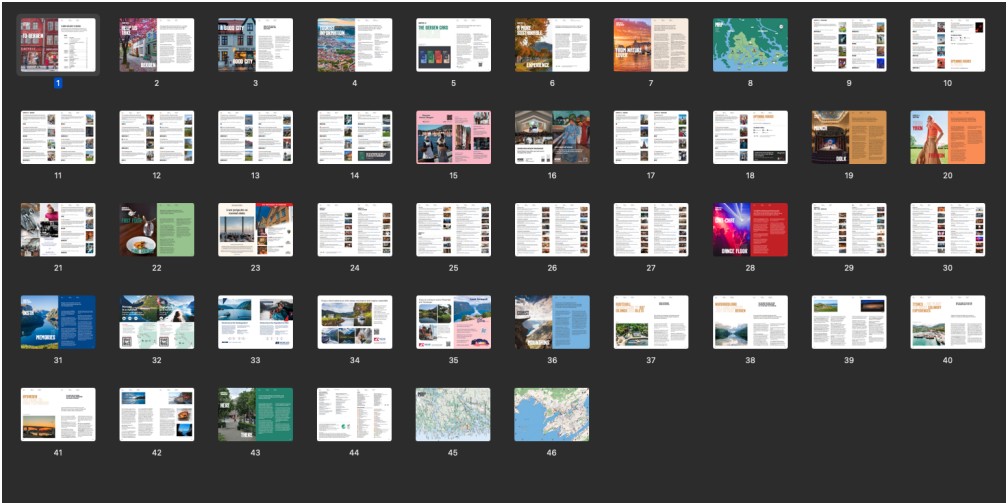

Figure 14: Document example about **Brochure**

## A.4  Existing Question Editing

Documents collected from existing datasets had been annotated with some questions and answers. However, their crowd-sourcing annotations inevitably make some questions, answers, and other meta information unqualified. So we conduct a systematic and manual pipeline to edit their annotations. Specifically, we classify six potential problems in original annotations. The definitions and examples of these problems are shown below.

**1. Wrong Answer or Evidence Pages:** The reference answers and/or evidence pages in original datasets are wrongly annotated.

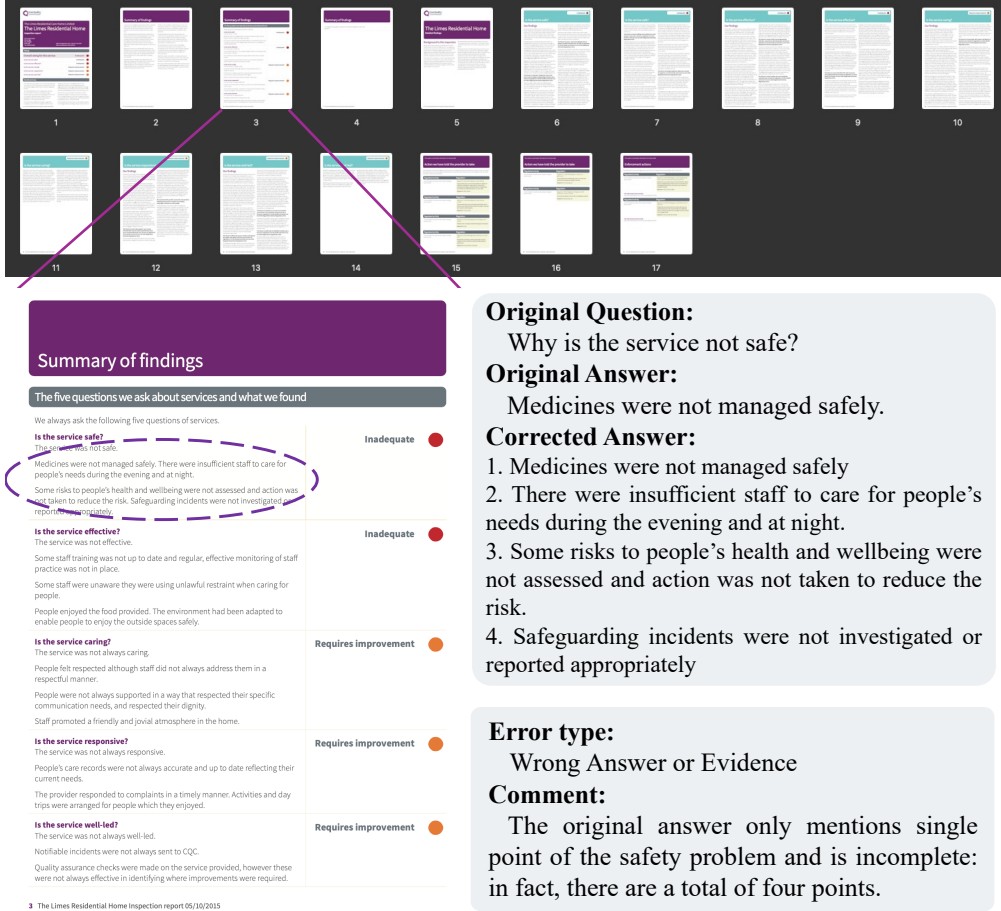

Figure 15: Example of the original annotation with *Wrong Answer or Evidence Pages*.

**2. Repetitive Question:** Too many questions with the same types (*e.g.,* key information extraction) occur in a single document (or even on the same page or point).

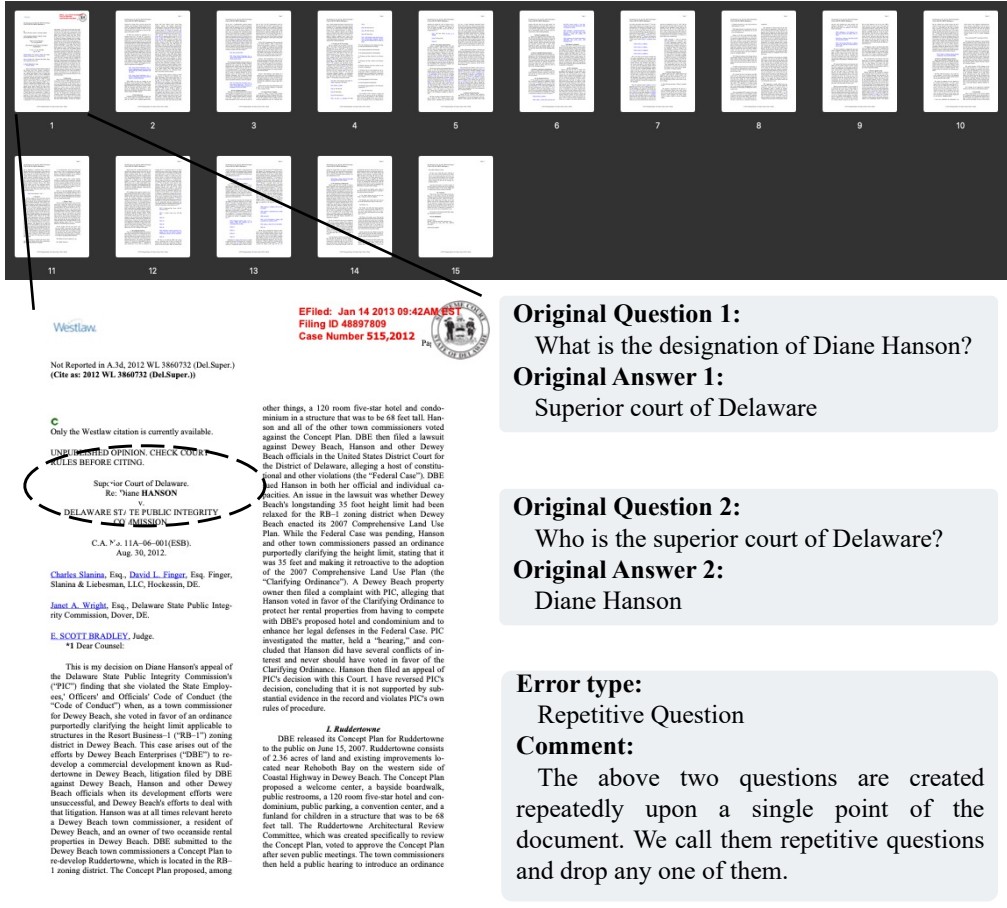

**Original Question 1:**
What is the designation of Diane Hanson?
**Original Answer 1:**
Superior court of Delaware

**Original Question 2:**
Who is the superior court of Delaware?
**Original Answer 2:**
Diane Hanson

**Error type:**
Repetitive Question
**Comment:**
The above two questions are created repeatedly upon a single point of the document. We call them repetitive questions and drop any one of them.

Figure 16: Example of the original annotation with *Repetitive Question*.

**3. Ambiguous Question:** The question is ambiguous at the document level (*e.g.,* the absence of entity, period, exact section or page, *etc.*), or too broad to exactly answer.

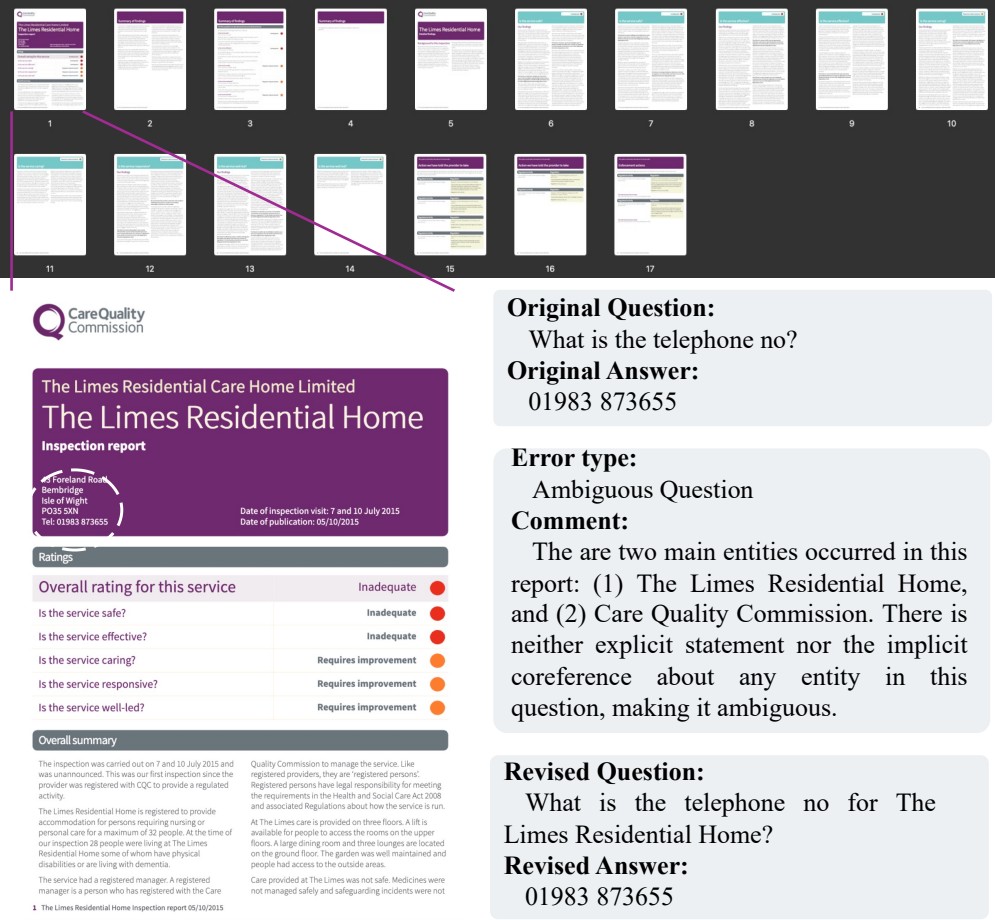

**Original Question:**
   What is the telephone no?
**Original Answer:**
   01983 873655

**Error type:**
   Ambiguous Question
**Comment:**
   The are two main entities occurred in this report: (1) The Limes Residential Home, and (2) Care Quality Commission. There is neither explicit statement nor the implicit coreference about any entity in this question, making it ambiguous.

**Revised Question:**
   What is the telephone no for The Limes Residential Home?
**Revised Answer:**
   01983 873655

Figure 17: Example of the original annotation with *Ambiguous Question*.

**4. Potential Shortcut:** The resolution of the question does not rely on two entities (across different pages) but only one of them, *i.e.,* there exists a shortcut for this question.

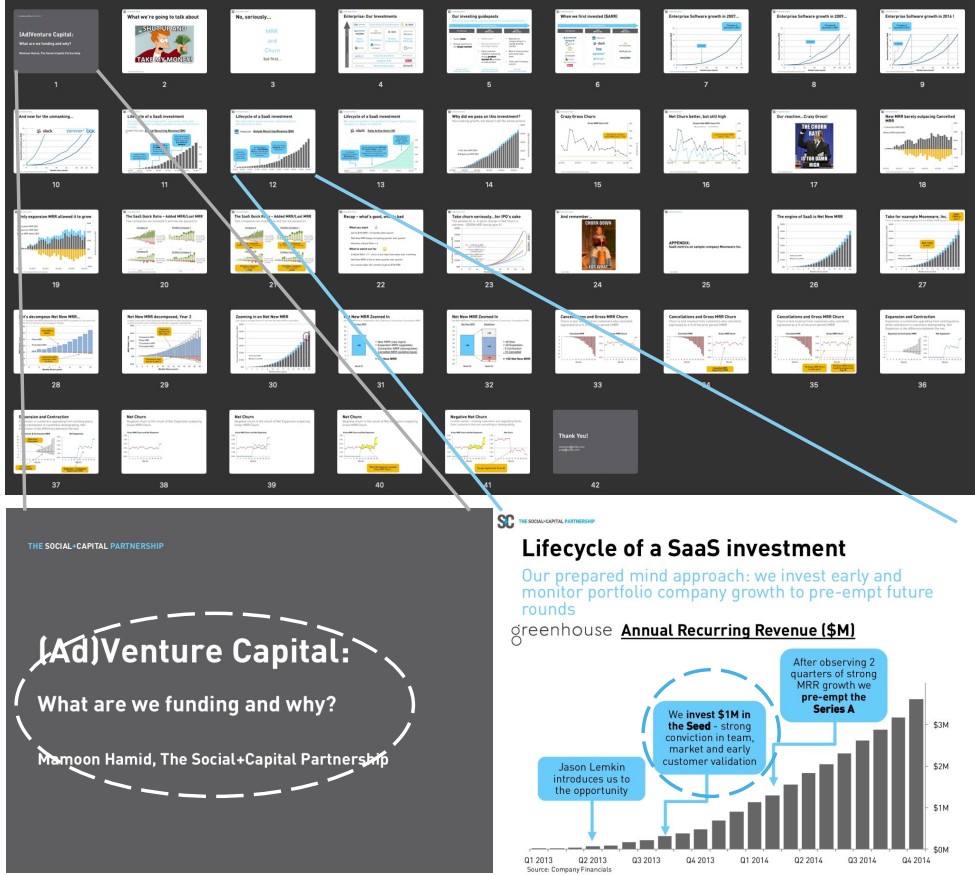

**Original Question:**
   Why did the company which Mamoon Hamid is affiliated with invest $1M in the seed for greenhouse?
**Original Answer:**
   Strong conviction in team, market and early customer validation.

**Error Type:**
   Potential Shortcut
**Comment:**
   The coreference of Adventure Capital circled in white in the left slide, *i.e., the company which Mammon Hamid is affiliated with*, makes no sense for answering this question. It is because that the words circled in blue in the right slide, *i.e., invest $1M in the seed*, is a potentially a strong shortcut for answering this question. Though seemingly relying on the information across two pages, it is still likely a single-page question.

**Revised Question:**
   Why did greenhouse invest $1M in the seed for greenhouse?
**Revised Answer:**
   Strong conviction in team, market and early customer validation.

Figure 18: Example of the original annotation with *Potential Shortcut*.

**5. Low Document-relevant Question:** The resolution of the question does not rely on the information from the document. It can be solved by the parametric knowledge in the LVLMs.

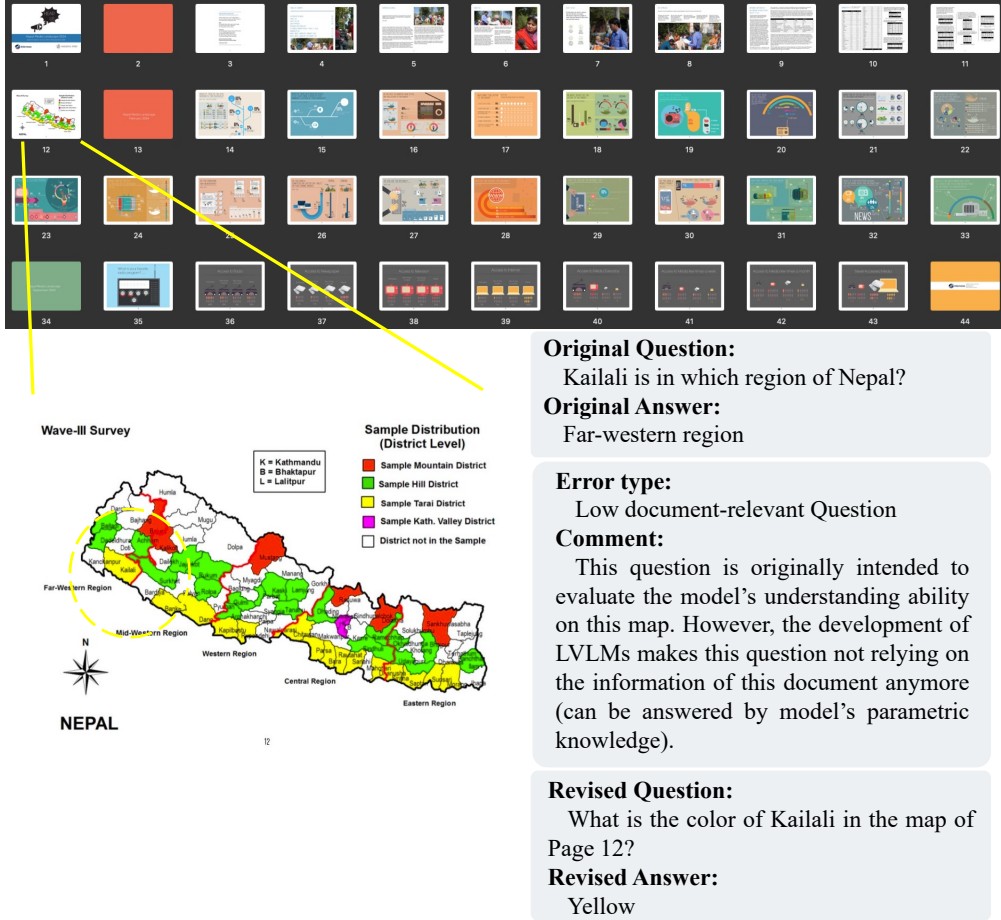

**Original Question:**
Kailali is in which region of Nepal?
**Original Answer:**
Far-western region

**Error type:**
Low document-relevant Question
**Comment:**
This question is originally intended to evaluate the model's understanding ability on this map. However, the development of LVLMs makes this question not relying on the information of this document anymore (can be answered by model's parametric knowledge).

**Revised Question:**
What is the color of Kailali in the map of Page 12?
**Revised Answer:**
Yellow

Figure 19: Example of the original annotation with *Low Document-relevant Question*.

**6. Decontextulization-required Question:** The understanding of the question is conditioned on a single page or even a single component of the document.

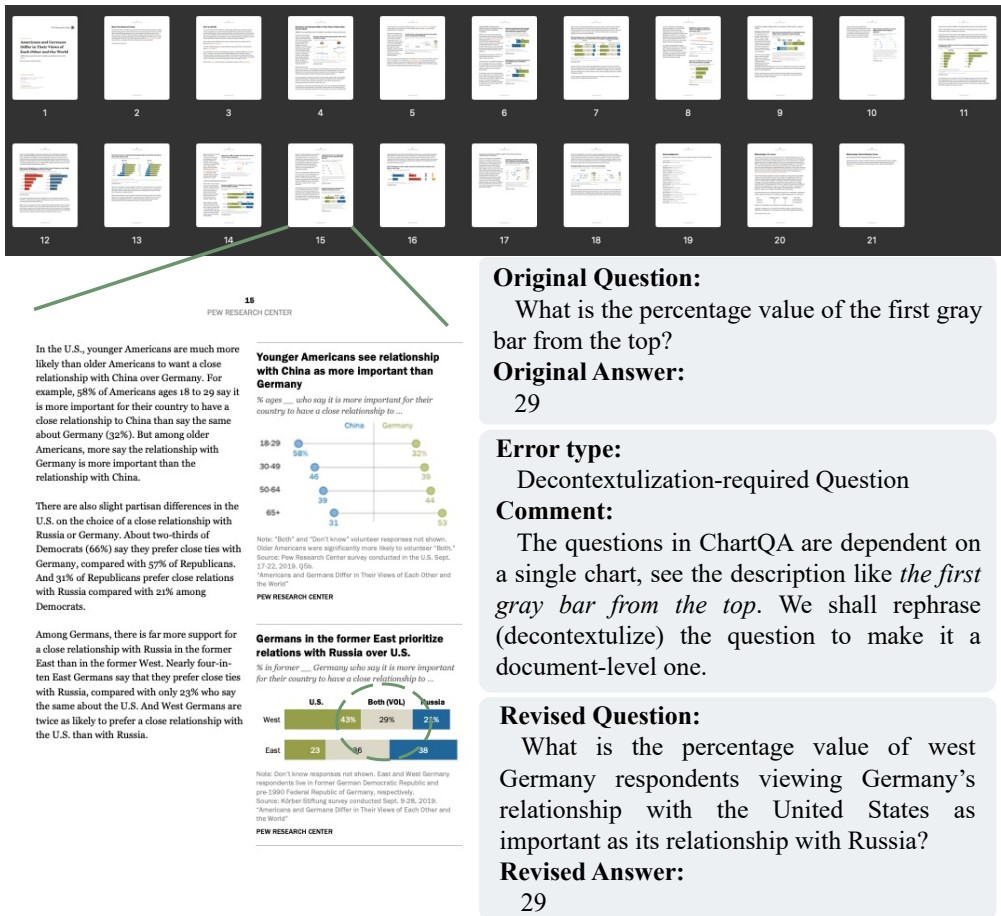

Figure 20: Example of the original annotation with *Decontextulization-required Question*.

When dealing with questions categorized under any of these six problem types, annotators are instructed to either revise or remove them. Typically, repetitive questions and those with potential shortcuts are removed. In contrast, wrongly-annotated or decontextualization-required questions are generally revised. For ambiguous and low document-relevant questions, the course of action depends more on the annotators' discretion.

## A.5    New Question Annotation

We annotate new questions on both existing and newly-collected documents. To ensure a diverse range of questions, we impose limitations on the question distributions categorized by their types (*i.e.,* single-page, cross-page or unanswerable) and evidence sources (*i.e.,* table, chart, image). To balance existing questions which are mostly single-page and text-based, we place greater emphasis on cross-page, unanswerable, table-related, chart-related, and image-related questions. The detailed standards are as follows:

| Document Type | Evidence Page | | Evidence Source | | | All |
|---|---|---|---|---|---|---|
| | Cross-page | Unanswerable | Table | Chart | Image | |
| Industrial File | ≥ 2 | - | | - | | ≥ 3 |
| Workshop & Tutorial | ≥ 2 | ≥ 1 | —— ≥ 3 —— | | | ≥ 6 |
| Research Report | ≥ 3 | ≥ 1 | ≥ 2 | ≥ 2 | - | ≥ 5 |
| Financial Report | ≥ 5 | ≥ 2 | ≥ 7 | - | - | ≥ 10 |
| Academic Paper | ≥ 3 | ≥ 1 | ≥ 2 | —- ≥ 3 —- | | ≥ 6 |
| Guidebook | ≥ 3 | ≥ 1 | - | - | ≥ 4 | ≥ 7 |
| Brochure | ≥ 2 | ≥ 1 | - | - | ≥ 3 | ≥ 7 |

Table 5: The **minimum** requirements for the number and distribution of questions, categorized by the evidence page numbers and evidence sources. We have set varying requirements for different document types based on their specific characteristics.

## A.6    Potential Bias for LVLM-based Quality Checking

As described in Section 3.3, we employ GPT-4o to remove document-agnostic (*i.e.,* can be correctly answer without documents) samples and review potential wrongly-labeled samples. A reasonable speculation raises that our final benchmark can be biased toward GPT-4o's answers, especially when GPT-4o outperforms others by a large margin. We discuss this potential bias as follows.

We check the effect of GPT-4o's involvement in the quality control step-by-step. Specifically, we compare the performance of samples remained after each step across GPT-4o and two other competitive models (GPT-4V and Gemini-1.5-Pro). We show their results in the table below.

| | GPT-4o | GPT-4V | Gemini-1.5-Pro |
|---|---|---|---|
| No quality control | 43.1% | 35.2% | 23.3% |
| + document-relevance detection | 41.2% | 31.0% | 20.5% |
| + document-relevance detection + self-reflection / cross-checking | 42.7% | 31.4% | 20.9% |

Table 6: Step-wise performance comparison with and without LVLM-based quality checking

The results illustrate that the potential bias in step 1 (document-relevance detection) actually reduce, rather than increase, the performance gap between GPT4o and other models. It is because that we filter out all samples correctly answered by GPT4o without the access to documents. Under this case, the more significant performance drop of GPT-4V and Gemini-1.5-Pro can only be attributed to their limited document understanding and over-reliance on their internal knowledge. Regarding the step 2 and 3 (self-reflection and cross-checking), we provide inconsistent answers between human annotations and GPT4o's predictions to annotators and ask them to check and revise accordingly. The potential bias of this step does lead to a slight performance bias (1.1% absolute difference at maximum). We believe that such bias is NOT the main cause of GPT4o's significantly best performance. Without the involvement of GPT-4o in the quality control process, GPT-4o still significantly outperforms GPT-4V by 7.9% (43.1% - 35.2%) and Gemini-1.5-Pro by 19.8% (43.1% - 23.3%). Accordingly, all primary conclusions in our paper still hold.

## A.7 GUI Screenshots

We present the screenshots for editing existing questions and annotating new questions (along with their reference answers and meta-data) in Figure 21 and Figure 22 respectively.

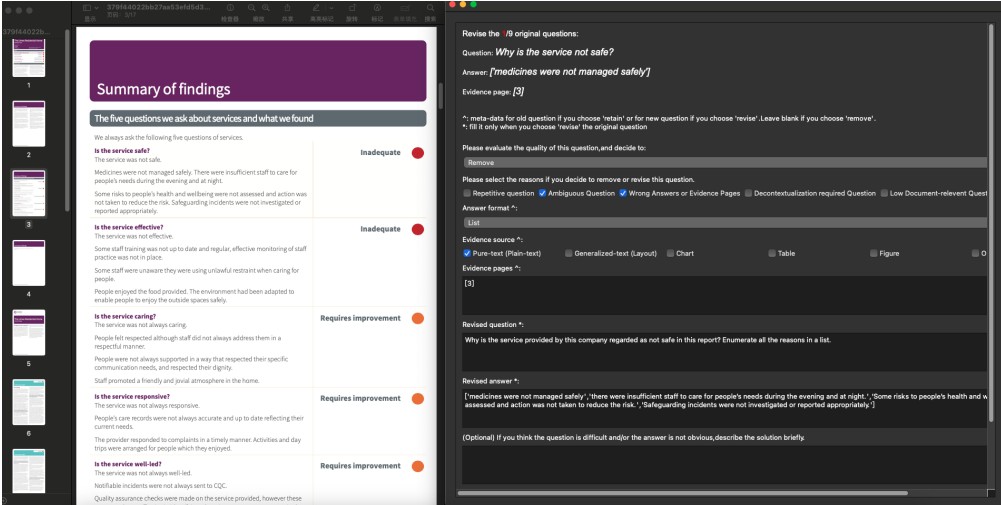

Figure 21: GUI screenshot for editing existing questions (along with reference answers and meta-data)

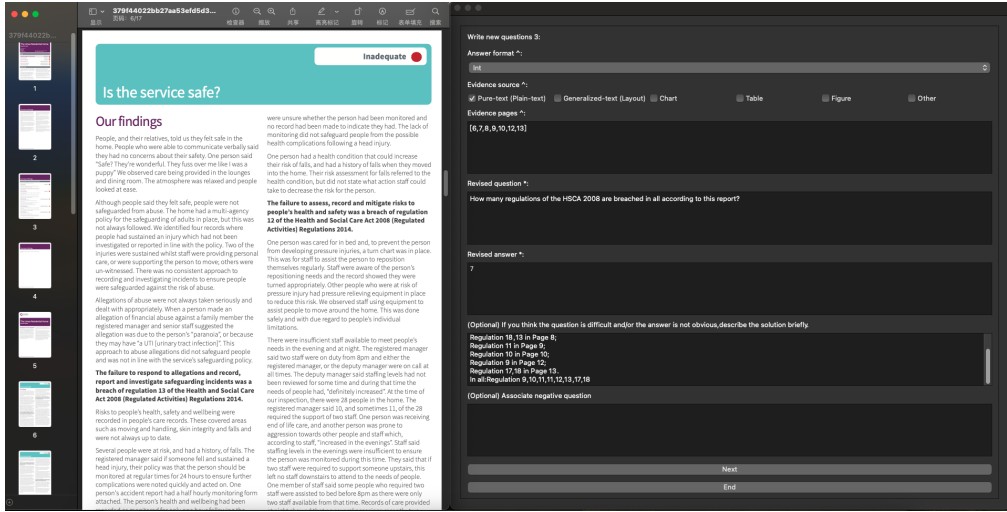

Figure 22: GUI screenshot for annotating new questions (along with reference answers and meta-data)

## A.8 Annotation Cost

This benchmark is annotated by the authors of this paper. Therefore, the data collection does not need compensation. And we count the time cost of our benchmark as below.

**Pre-annotation** (about 45h): the development of annotation interface (10h), the writing of annotation guideline (5h), training session (10h), preliminary annotation and personalized feedback (20h).

**Annotation** (about 150h): It takes about 60-90 minutes for the annotation of each document. And all of the 130 documents take about 150 hours.

**Post-annotation** (about 45h): quality checking (30h), data processing and release preparation (15h).

In summary, our benchmark annotation approximately takes a total of 45+150+45=240 hours (1.36 man months).

# B  Experimental Details

## B.1  Prompt for Response Generation

Listing 1: Prompt used for response generation. The [Document] is in PNG format (page screenshots) for LVLMs, and TXT format for LLMs.

```
[Document]

Read the above documents and answer this question:
[question]

Please make your answer as concise as possible.
```

## B.2  Prompt for Answer Extraction

Listing 2: Prompt used for answer extraction.

```
Given the question and analysis, you are tasked to extract answers with required
    formats from the free-form analysis.
- Your extracted answers should be one of the following formats: (1) Integer, (2)
    Float, (3) String and (4) List. If you find the analysis the question can not
    be answered from the given documents, type "Not answerable". Exception: If the
    analysis only tells you that it can not read/understand the images or documents,
     type "Fail to answer".
- Please make your response as concise as possible. Also note that your response
    should be formatted as below:
'''
Extracted answer: [answer]
Answer format: [answer format]
'''

Please read the following example, then extract the answer from the model response
    and type it at the end of the prompt.

---
Question: List the primary questions asked about the services in this report.
Analysis: The primary questions asked about the services in the report for The Limes
     Residential Home are:
1. Is the service safe?
2. Is the service effective?
3. Is the service caring?
4. Is the service responsive?
5. Is the service well-led?
Extracted answer: ['Is the servife safe?', 'Is the service effective', 'Is the serve
     caring?', 'Is the service responsive?', 'Is the service well-led?']
Answer format: List

---
Question: How many regulations of the HSCA 2008 are breached in all according to
    this report?
Analysis: According to the report, the provider breached 10 Health and Social Care
    Act 2008 (Regulated Activities) Regulations in total. Here are the specifics:

1. Regulation 13: Safeguarding service users from abuse and improper treatment
2. Regulation 12: Safe care and treatment
3. Regulation 18: Staffing
4. Regulation 11: Need for consent
5. Regulation 10: Dignity and respect
6. Regulation 9: Person-centred care
7. Regulation 17: Good governance
8. Regulation 18 (CQC Registration Regulations 2009): Notification of other
    incidents
9. Regulation 18: Failure to maintain an accurate and up-to-date care plan
```

10. Regulation 11: Failure to implement the Mental Capacity Act 2005 code of
    practice effectively

These breaches involve issues concerning staffing, safeguarding, medicines
    management, dignity and respect, consent, care planning, governance, and
    failure to notify the CQC of incidents.
Extracted answer: 10
Answer format: Integer

---
Question: According to the survey that is the percentage of Chinese who are paying
    more or about the same attention to politics after Trump's election?
Analysis: The survey provided does not specify the percentage of Chinese individuals
     specifically who are paying more or about the same attention to politics after
     Trump's election. The report focuses primarily on American demographics and
    does not include specific details about the Chinese population in relation to
    this question. If you need information about a different demographic or a
    summary of the findings from the American demographic, I can certainly help
    with that!
Extracted answer: Not answerable
Answer format: None

---
Question: How many quotations from male respondent over 50 years old are included in
     this report?
Analysis: The image you've provided appears to be a screenshot of a document with
    multiple charts. However, the text is too small and blurry to read accurately.
    If you can provide a clearer image or more context, I might be able to help you
    with your question.
Extracted answer: Fail to answer
Answer format: None

---

---

## B.3 Rules for Score Calculation

We evaluate the model's responses by scoring the extracted answers against the reference answers.
The scorer is rule-based and employs different strategies according to the format of the reference
answer. We detail its rules as below:

**String:** We firstly use a series of regular expressions to determine whether the answers require exact
matching (*e.g.,* telephone numbers, email addresses, website addresses, file names, times, dates, *etc.*).
If an exact match is needed, we perform a straightforward string comparison and score the answer
either 0 or 1. Otherwise, we follow previous work [17] and calculate the ANLS (Average Normalized
Levenshtein Similarity) with a pre-defined threshold ($\tau = 0.5$).

**Integer:** We perform an exact match comparison and score the answer either 0 or 1.

**Float:** We view the prediction and reference answers as equal if they fall within a 1% relative
tolerance.

**List:** We adopt a relatively strict rule for scoring answers in list format: predictions that do not have
the same number of elements as the reference receive a score of 0. For the remaining predictions, as
Eq. 1 indicates, we score each element in order and use the minimum element-wise score as the score
for the entire list. The element-wise scoring strategies is determined by the formats of elements (*i.e.,*
string, integer or float).

$$
\begin{aligned}
&\texttt{pred\_list}, \texttt{ref\_list} = \texttt{sorted(pred\_list)}, \texttt{sorted(ref\_list)} \\
&\texttt{Score(pred\_list, ref\_list)} = \texttt{min(} \\
&\qquad \texttt{[Score(pred, ref) for pred, ref in zip(pred\_list, ref\_list)]} \\
&\texttt{)}
\end{aligned} \tag{1}
$$

Evaluation detailed in the Appendix B.4 shows that while this scorer is not perfect, it aligns well with human judgment. We will continue refining these rules to cover more corner cases and enhance their accuracy.

## B.4 Human Evaluation on the Automatic Evaluation Pipeline

We conduct human evaluations to assess the performance of our automatic evaluation pipeline, which includes the answer extractor and the score calculator. Specifically, we randomly select 100 questions and review their responses from two representative LVLMs: GPT-4o and Gemini-1.5-Pro. We manually evaluate the correctness of each response and compare the results between human evaluation and automatic evaluation. The performance, as shown in Table 7, indicates a high correlation between human judgment and our automatic pipeline.

| Model | Inconsistent Evaluation | | |
|---|---|---|---|
| | Ans. Extractor | Scorer | Overall |
| GPT-4o | 4 | 2 | 6 |
| Gemini-1.5-Pro | 2 | 2 | 4 |

Table 7: We manually check 100 responses from GPT-4o and Gemni-1.5-Pro, and compare the evaluation results between humans and our automatic pipeline.

## B.5 Model Hyperparameters

The hyperparameters of used LVLMs and LLMs in Section 3.3 are detailed in Table 8. The temperature is set as $0.0$, and the max_new_tokens is set as $1024$ for all the models. The 'concatenated_images' parameter determines the maximum number of images that can be combined into a single input for LVLMs. By concatenating multiple images, we can meet the minimum context window requirements. The 'max_pages' parameter specifies the maximum number of images that can be directly input into the LVLMS without concatenation.

| Model | Hyperparameters |
|---|---|
| *LLM* | |
| ChatGLM-128k | max_input_words=60000 |
| Mistral-Instruct-v0.2-7B | max_input_words=20000 |
| Mixtral-Instruct-v0.1-8x7B | max_input_words=20000 |
| Mixtral-Instruct-v0.1-8x22B | max_input_words=40000 |
| QWen-Plus | max_input_words=16000 |
| DeepSeek-V2 | max_input_words=20000 |
| *LVLM* | |
| DeepSeek-VL-Chat | concatenated_images=5 |
| Qwen-VL-Chat | concatenated_images=5 |
| Idefics2 | concatenated_images=5 |
| MiniCPM-Llama3-V2.5 | concatenated_images=2 |
| InternLM-XC2-4KHD | concatenated_images=2 |
| Monkey-Chat | concatenated_images=1 |
| CogVLM2-Llama3-Chat | concatenated_images=1 |
| InternVL-Chat-v1.5 | concatenated_images=5 |
| EMU2-Chat | concatenated_images=5 |
| *LLM & LVLM* | |
| Claude-3 Opus | version=`claude-3-opus-20240229`, concatenated_images=20 |
| Gemini-1.5-Pro | max_pages=120, version=`gemini-1.5-pro-latest` |
| GPT-4-turbo | max_pages=120, version=`gpt-4-turbo-2024-04-09` |
| GPT-4o | max_pages=120, version=`gpt-4o-2024-05-13` |

Table 8: Model Hyperparameters

## C  Qualitative Study

### C.1  Error Analysis

We delve into the analysis of error by GPT-4o to further understand its bottlenecks and potentials on long-context document understanding. We manually check 72 incorrect responses and categorized their error reasons into 7 types. Except for the *Extraction Error* caused by our automatic evaluation pipeline (see Appendix B.4), we detail and showcase another six reasons as below:

**Perceptual Error:** GPT-4o sometimes struggles to extract or understand visual information from document screenshots. For instance, it misinterprets the axes and colored circles in the charts shown in Figure 23. Additionally, it inaccurately counts the number of green bars in Figure 24. They demonstrate that even the cutting-edge LVLMs still fall short in fundamental perceptual capabilities.

**Incomplete Evidence:** Though GPT-4o has achieved significantly better *global searching abilities* compared to other models when dealing with lengthy, multi-modal documents, it sometimes still omits certain information. For example, GPT-4o misses one chapter author from Columbia University in the full list (Figure 25). Additionally, it overlooks an app that appears across two pages (Figure 26).

**Hallucinated Evidence:** As stated in Section 3.4, GPT-4o adopts more aggressive strategies and tends to provide more false-positive answers. It sometimes even fabricates non-existent evidence in documents to support its incorrect responses. For example, it references a non-existent page in Figure 27, and fabricates the content of a page in Figure 28. The above examples clearly reveal the importance of further research on LVLMs' hallucination and safety.

**Knowledge Lacking:** Resolving certain questions requires both information from the documents and the parametric knowledge within LVLMs. We have observed error cases stemming from the absence of specific knowledge. For example, GPT-4o overlooks details about the *fixed asset turnover ratio* and uses the single-point value instead of the average value to calculate this metric (Figure 29). Additionally, it misidentifies buildings at Tsinghua University in Figure 30.

**Reasoning Error:** Though not a primary cause, flawed reasoning based on correctly collected evidence and information from documents can sometimes lead to wrong answers. For example, GPT-4o correctly gathers all data but calculates a relative percentage instead of an absolute percentage in Figure 31. Additionally, as shown in Figure 32, it correctly lists all quizzes but inaccurately counts them in the final step.

**Irrelevant Answer:** GPT-4o sometimes misunderstands the intent of questions and provides irrelevant responses. For instance, in Figure 33, GPT-4o answers about button operations when the question asks about button functions. Similarly, in Figure 34, where the question asks for the MOST discrimination type, GPT-4o summarizes all types instead.

**Perceptual Error: Case 1**

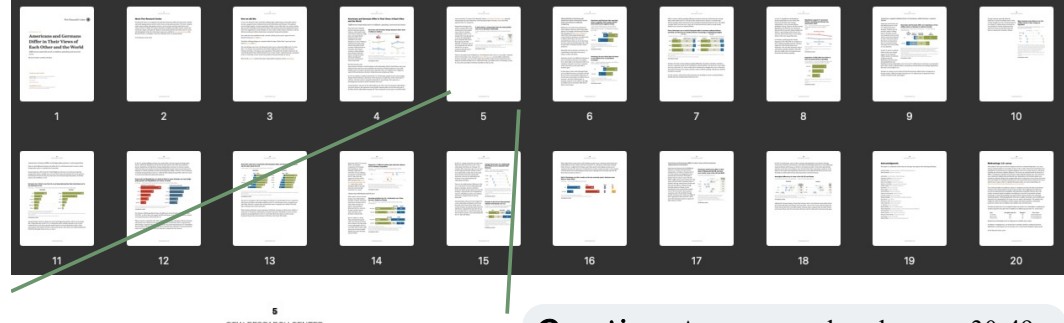

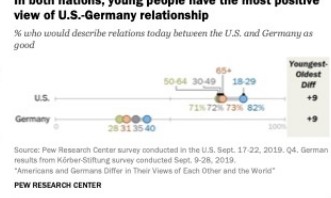

**In both nations, young people have the most positive view of U.S.-Germany relationship**

*% who would describe relations today between the U.S. and Germany as good*

views toward the U.S. found in Pew Research Center's 2019 Global Attitudes survey, especially among people who place themselves on the ideological right in Germany, even as favorable opinions of the U.S. remain low.

Despite these divergences in opinion, young people in both countries have more positive views of the U.S.-German relationship. In the U.S., for example, 82% of people ages 18 to 29 say the relationship is good, compared with 73% of those ages 65 and older. Similarly, in Germany, four-in-ten young people say relations with the U.S. are good, compared with only 31% of those ages 65 and older.

Source: Pew Research Center survey conducted in the U.S. Sept. 17-22, 2019. Q4. German results from Körber-Stiftung survey conducted Sept. 9-28, 2019.

"Americans and Germans Differ in Their Views of Each Other and the World"

PEW RESEARCH CENTER

These are among the major findings from a Pew Research Center survey of 1,004 adults conducted in the U.S. from Sept. 17-22, 2019, and a Körber-Stiftung survey of 1,000 adults conducted in Germany from Sept. 9-28, 2019. This analysis also includes results from Pew Research Center's Spring 2019 Global Attitudes survey, conducted among 1,503 adults in the U.S. from May 13-June 18, 2019, and 2,015 adults in Germany from May 31-July 25, 2019.

---

**Question:** Among people who age 30-49, what is the difference of percentage value between Americans and Germen having positive view on their bilateral relationship?

**Response:** The difference in the percentage of people aged 30-49 who have a positive view of the U.S.-Germany relationship between Americans and Germans is 9% points. According to data on page 5, 40% of Americans aged 30-49 view the relationship positively, while only 31% of Germans in the same age group share this view.

---

**Error Type:** Perceptual Error
**Analysis:** The model wrongly identifies the percentages of people aged 30-49 who have a positive view of the U.S.-Germany relationship from the chart. It should be 35% and 72%, rather than 31% and 40%.

Figure 23: Error example about *Perceptual Error*

**Perceptual Error: Case 2**

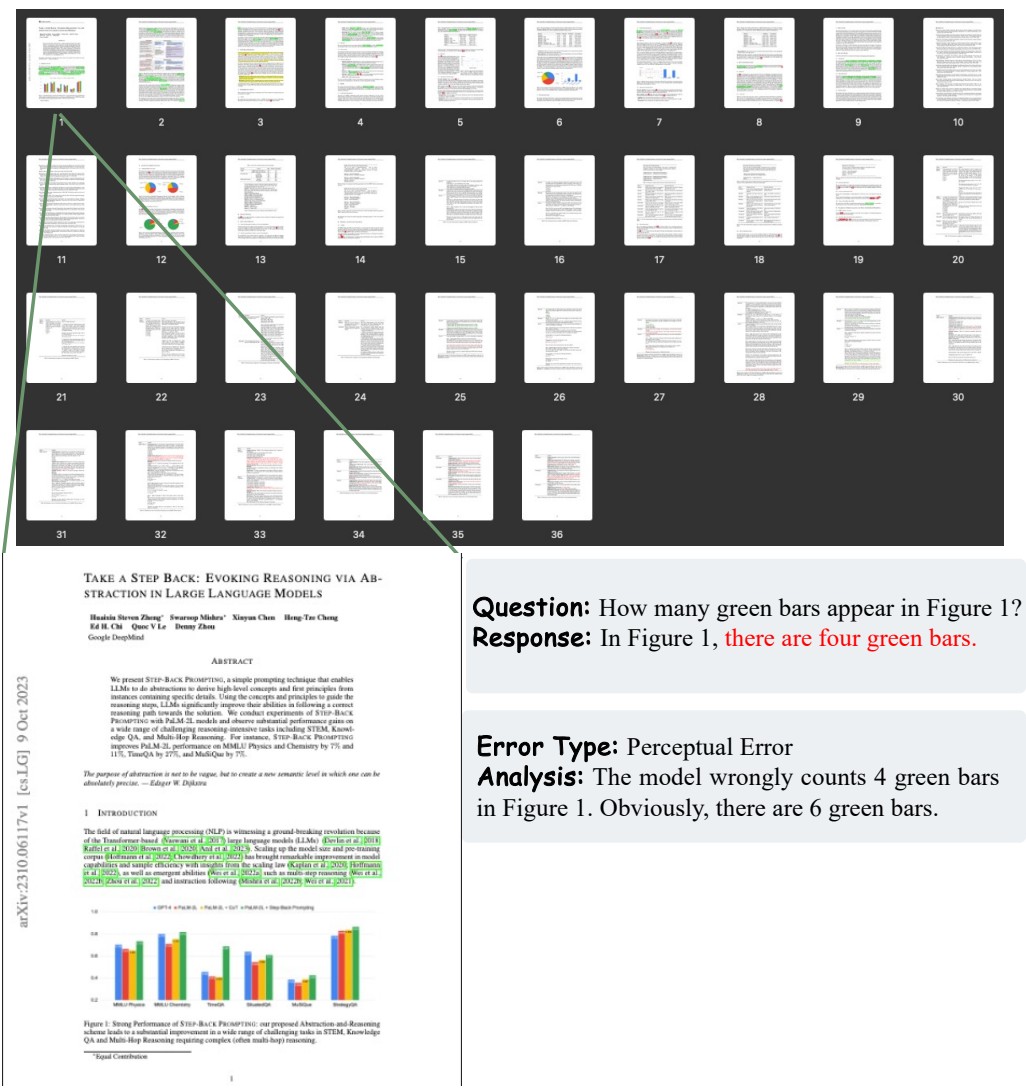

**Question:** How many green bars appear in Figure 1?
**Response:** In Figure 1, there are four green bars.

**Error Type:** Perceptual Error
**Analysis:** The model wrongly counts 4 green bars in Figure 1. Obviously, there are 6 green bars.

Figure 24: Error example about *Perceptual Error*

**Incomplete Evidence: Case 1**

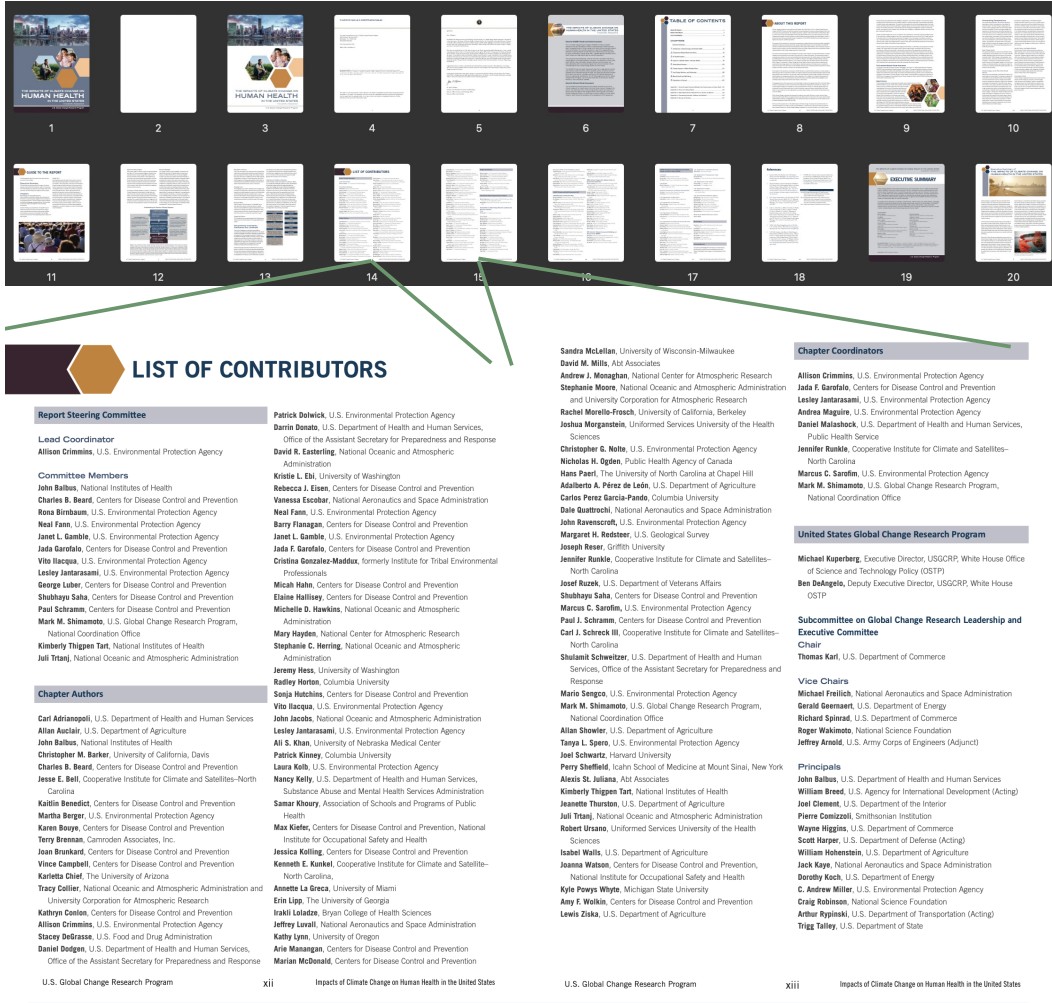

**Question:** How many chapter authors of this report come from Columbia University?

**Response:** The report lists the chapter authors on page xii. From Columbia University, the chapter authors are: 1. Radley Horton, 2. Patrick Kinney. Therefore, there are two chapter authors from Columbia University.

**Error Type:** Incomplete Evidence

**Analysis:** The model only finds out 2 of 3 chapter authors from Columbia University, leaving out Garcia-Pando shown in the top-left of Page 15.

Figure 25: Error example about *Incomplete Evidence*

**Incomplete Evidence: Case 2**

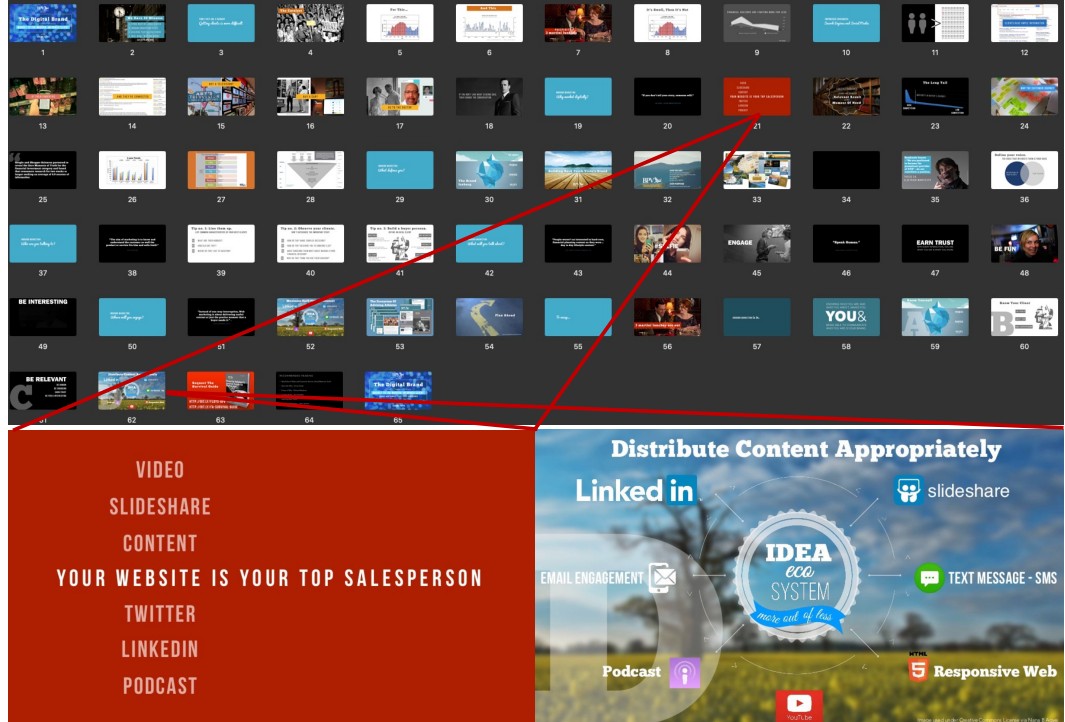

**Question:** What are the overlapped apps between page 21 and 62?
**Response:** The overlapped apps between page 21 and page 62 are:
- LinkedIn
- SlideShare

**Error Type:** Incomplete Evidence
**Analysis:** The model correct identifies the two shared apps (LinkedIn and SlideShare) between two pages. But it leaves out another one app: Podcast.

Figure 26: Error example about *Incomplete Evidence*

**Hallucinated Evidence: Case 1**

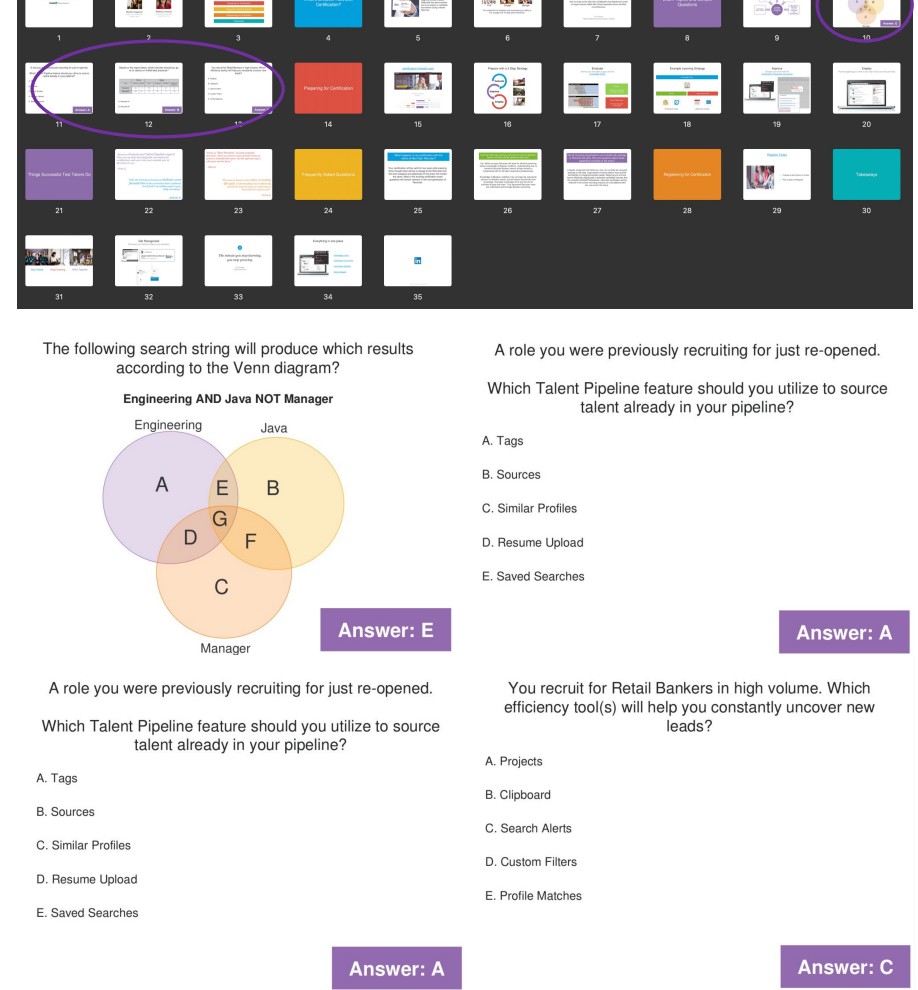

The following search string will produce which results according to the Venn diagram?

**Engineering AND Java NOT Manager**

Engineering · Java

A E B
G
D F
C

Manager

**Answer: E**

A role you were previously recruiting for just re-opened.

Which Talent Pipeline feature should you utilize to source talent already in your pipeline?

A. Tags

B. Sources

C. Similar Profiles

D. Resume Upload

E. Saved Searches

**Answer: A**

A role you were previously recruiting for just re-opened.

Which Talent Pipeline feature should you utilize to source talent already in your pipeline?

A. Tags

B. Sources

C. Similar Profiles

D. Resume Upload

E. Saved Searches

**Answer: A**

You recruit for Retail Bankers in high volume. Which efficiency tool(s) will help you constantly uncover new leads?

A. Projects

B. Clipboard

C. Search Alerts

D. Custom Filters

E. Profile Matches

**Answer: C**

**Question:** How many sampled questions illustrated in this slide?
**Response:** There are five sample questions illustrated in this slide deck. They are shown on the following slides:
1. Slide 10: "The following search string will produce which results according to the Venn diagram?"
2. Slide 11: "A role you were previously recruiting for just re-opened. Which Talent Pipeline feature should you utilize to source talent already in your pipeline?"
3. Slide 12: "Based on the report below, which recruiter should you go to for advice on InMail best practices?"
4. Slide 13: "You recruit for Retail Bankers in high volume. Which efficiency tool(s) will help you constantly uncover new leads?"
5. Slide 14: "What happens to my certification with the rollout of Next-Gen Recruiter?"

**Error Type:** Hallucinated Evidence
**Analysis:** The model totally fabricates the contents of Slide 14 (Page 14). This slide is not about sampled question.

Figure 27: Error example about *Hallucinated Evidence*

**Hallucinated Evidence: Case 2**

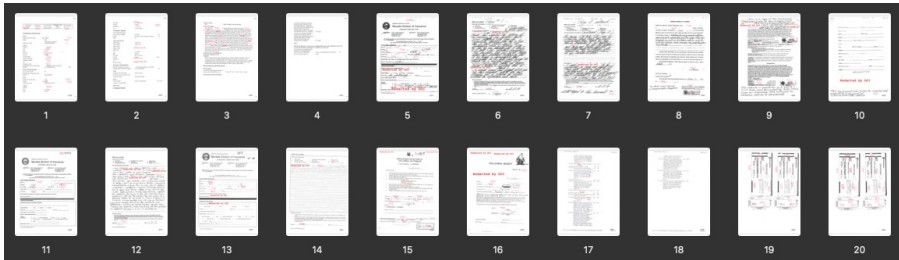

**Question:** Is there blue color handwritten words in page 30?
**Response:** Yes, there are blue handwritten words on page 30.

**Error Type:** Hallucinated Evidence
**Analysis:** This document has only 20 pages. There is no "so-called" page 30 in the response.

Figure 28: Error example about *Hallucinated Evidence*

**Knowledge Lacking: Case 1**

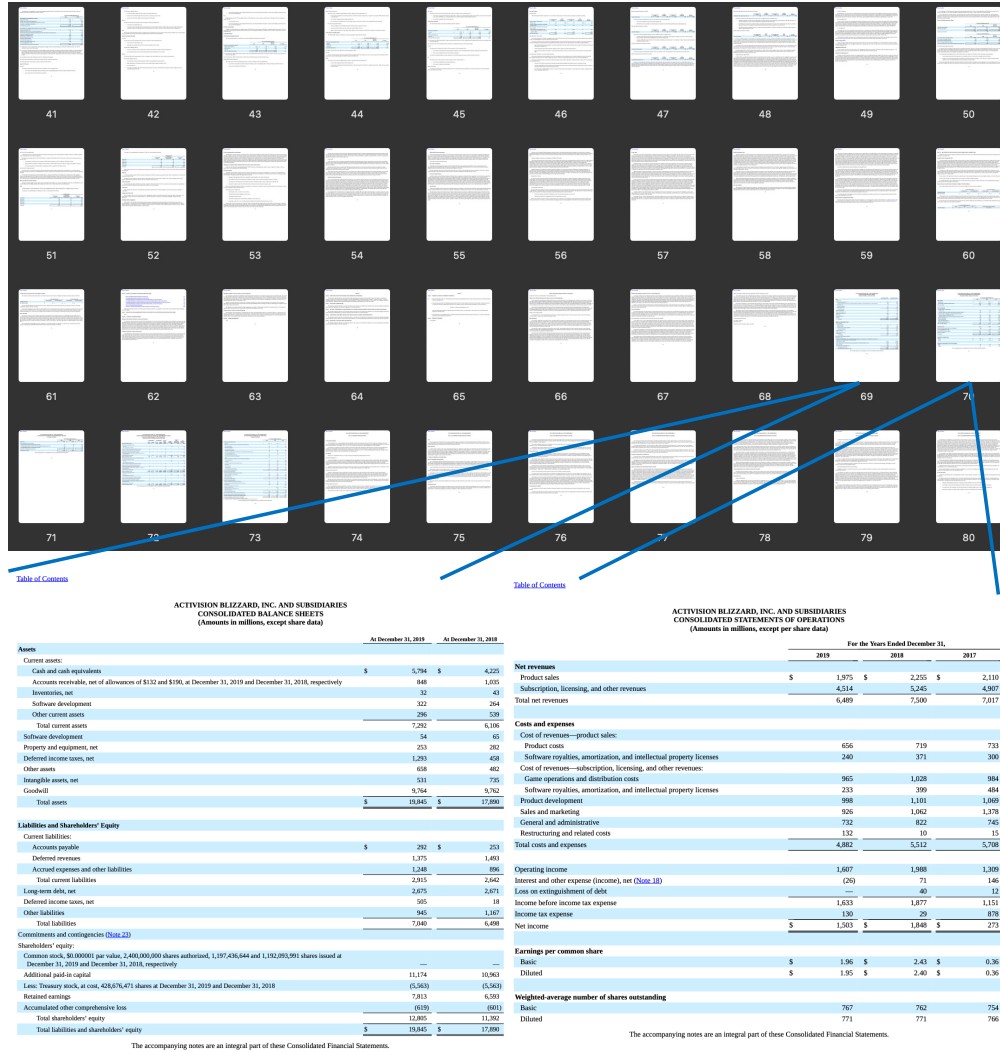

Question: What is the FY2019 fixed asset turnover ratio for Activision Blizzard?

Response: The fixed asset turnover ratio is calculated by dividing net sales by net fixed assets. The formula is: Fixed Asset Turnover Ratio = Net Sales / Net Fixed Assets
From the provided document, we can find the necessary values for the fiscal year 2019:
- Net Sales (Net Revenues) for 2019: $6,489 million
- Net Fixed Assets (Property and Equipment, net) at the end of 2019: $253 million
Using these values, the fixed asset turnover ratio is:
6,489 / 253 ≈ 25.65
Therefore, the fixed asset turnover ratio for Activision Blizzard for the fiscal year 2019 is approximately 25.65.

Error Type: Knowledge Lacking
Analysis: The model uses a wrong formula to calculate the financial metric. To calculate the Fixed Asset Turnover Ratio, the denominator should be the average value of net fixed assets in the past two years, instead of the value of a single year.

Figure 29: Error example about *Knowledge Lacking*

**Knowledge Lacking: Case 2**

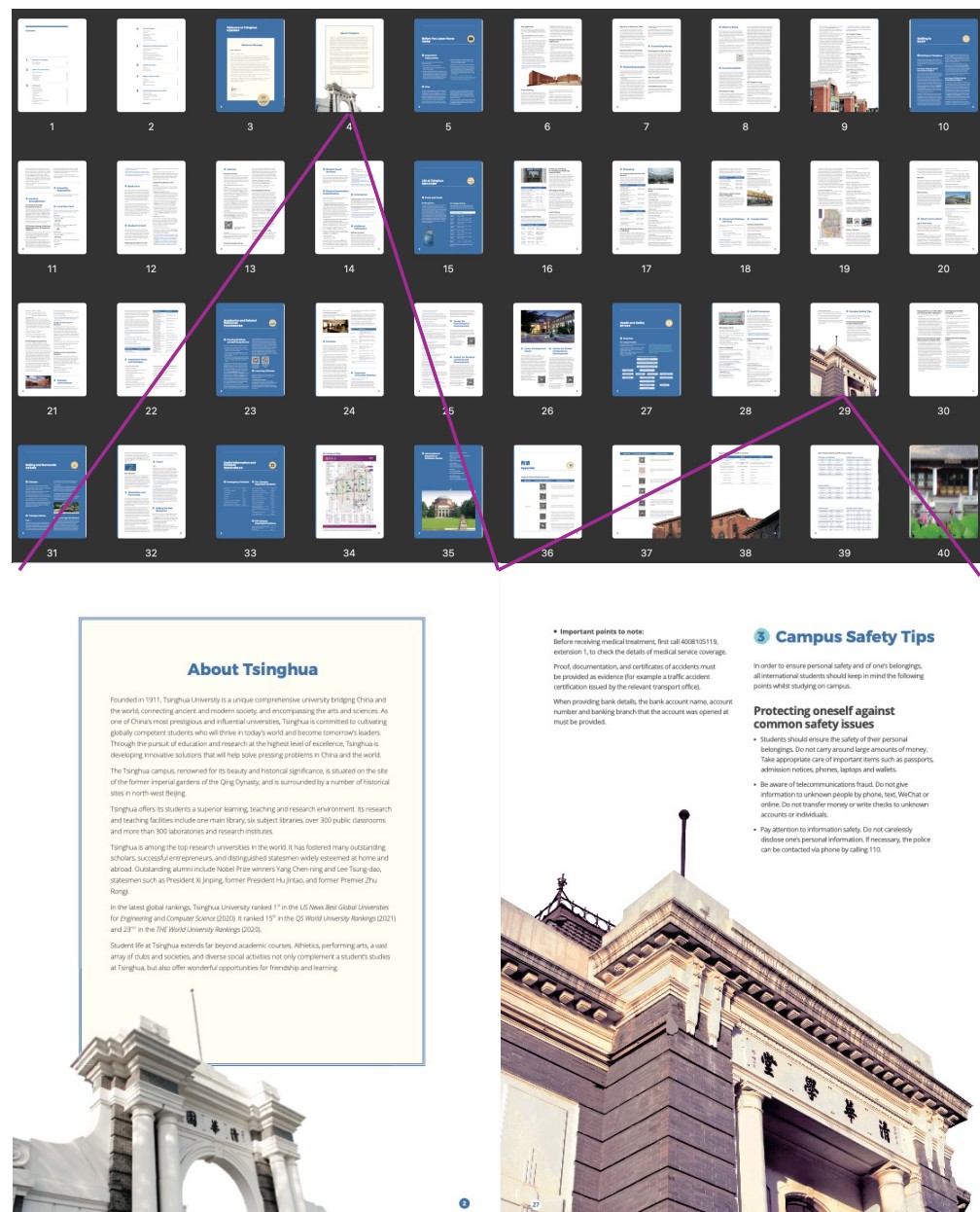

**Question:** How many figures in this document show the old gate of Tsinghua ("Er Xiao Men" in Chinese)?

**Response:** The document contains two figures showing the old gate of Tsinghua ("Er Xiao Men" in Chinese). The first one is on page 2, and the second one is on page 27

**Error Type:** Knowledge Lacking

**Analysis:** Though it has similar appearance, the building shown in Page 27 (i.e. the right figure) is not "Er Xiao Men" but "Qing Hua Xue Tang".

Figure 30: Error example about *Knowledge Lacking*

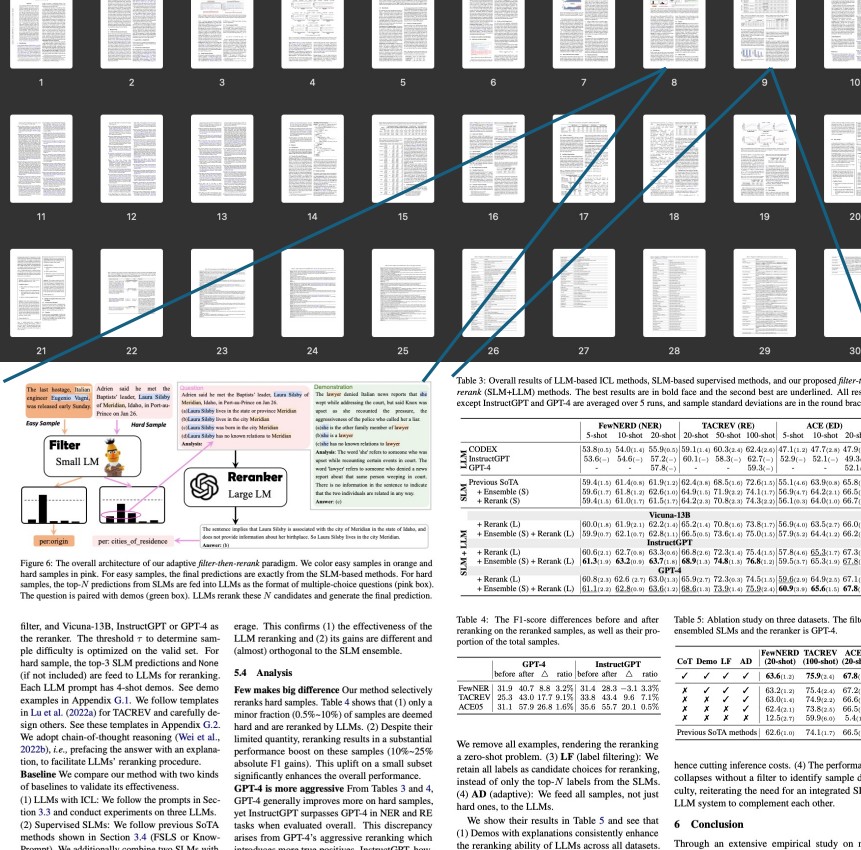

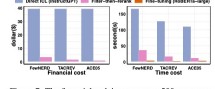

**Question:** If I drop the green module shown in Figure 6, what is the absolute percentage drop on 100-shot TACREV setting?

**Response:** The absolute percentage drop on the 100-shot TACREV setting when the green module is dropped can be found in Table 4. For the 100-shot TACREV setting, the F1 score before reranking is 72.0, and after reranking, it is 74.3. The absolute percentage drop is calculated as follows:

Absolute Percentage Drop = ((F1 score before reranking - F1 score after reranking) / F1 score after reranking) × 100

Absolute Percentage Drop = ((74.3 - 72.0) / 74.3) × 100

Absolute Percentage Drop = (2.3 / 74.3) × 100

Absolute Percentage Drop ≈ 3.1%

So, the absolute percentage drop on the 100-shot TACREV setting when the green module is dropped is approximately 3.1%.

**Error Type:** Reasoning Error

**Analysis:** The formula listed in model response is the relative percentage drop, instead of the absolute percentage drop.

Figure 31: Error example about *Reasoning Error*

**Reasoning Error: Case 2**

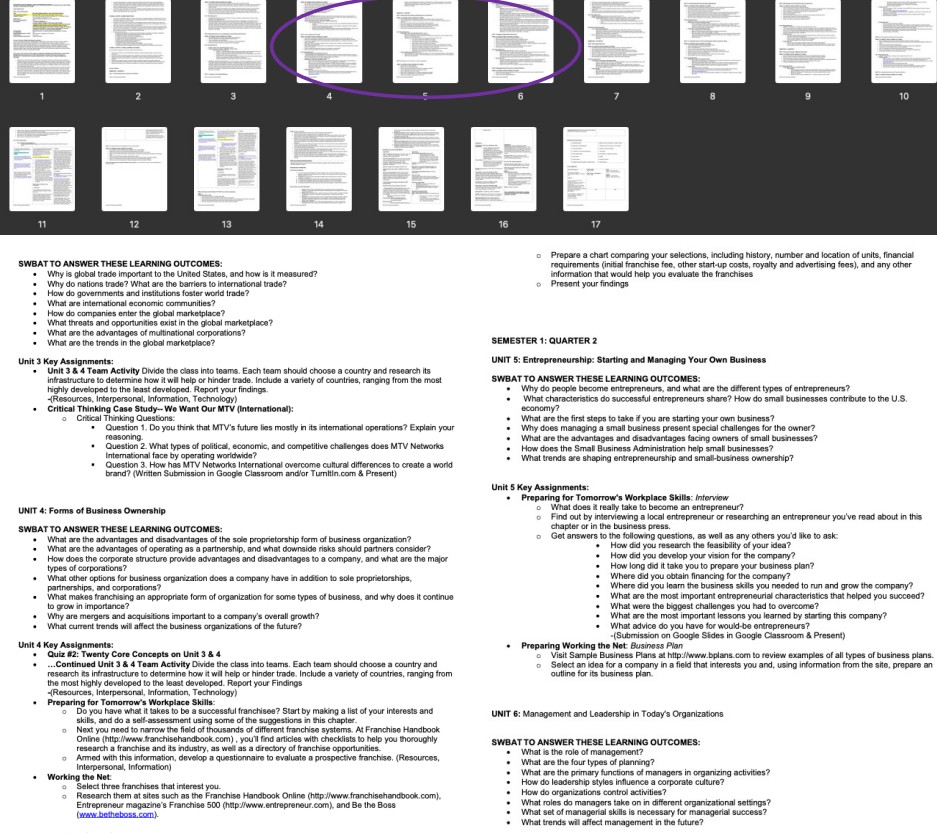

**SWBAT TO ANSWER THESE LEARNING OUTCOMES:**
- Why is global trade important to the United States, and how is it measured?
- Why do nations trade? What are the barriers to international trade?
- How do governments and institutions foster world trade?
- What are international economic communities?
- How do companies enter the global marketplace?
- What threats and opportunities exist in the global marketplace?
- What are the advantages of multinational corporations?
- What are the trends in the global marketplace?

**Unit 3 Key Assignments:**
- **Unit 3 & 4 Team Activity** Divide the class into teams. Each team should choose a country and research its infrastructure to determine how it will help or hinder trade. Include a variety of countries, ranging from the most highly developed to the least developed. Report your findings.
  -(Resources, Interpersonal, Information, Technology)
- **Critical Thinking Case Study-- We Want Our MTV (International):**
  - Critical Thinking Questions:
    - Question 1. Do you think that MTV's future lies mostly in its international operations? Explain your reasoning.
    - Question 2. What types of political, economic, and competitive challenges does MTV Networks International face by operating worldwide?
    - Question 3. How has MTV Networks International overcome cultural differences to create a world brand? (Written Submission in Google Classroom and/or TurnItIn.com & Present)

**UNIT 4: Forms of Business Ownership**

**SWBAT TO ANSWER THESE LEARNING OUTCOMES:**
- What are the advantages and disadvantages of the sole proprietorship form of business organization?
- What are the advantages of operating as a partnership, and what downside risks should partners consider?
- How does the corporate structure provide advantages and disadvantages to a company, and what are the major types of corporations?
- What other options for business organization does a company have in addition to sole proprietorships, partnerships, and corporations?
- What makes franchising an appropriate form of organization for some types of business, and why does it continue to grow in importance?
- Why are mergers and acquisitions important to a company's overall growth?
- What current trends will affect the business organizations of the future?

**Unit 4 Key Assignments:**
- **Quiz #2: Twenty Core Concepts on Unit 3 & 4**
- **...Continued Unit 3 & 4 Team Activity** Divide the class into teams. Each team should choose a country and research its infrastructure to determine how it will help or hinder trade. Include a variety of countries, ranging from the most highly developed to the least developed. Report your Findings
  -(Resources, Interpersonal, Information, Technology)
- **Preparing for Tomorrow's Workplace Skills:**
  - Do you have what it takes to be a successful franchisee? Start by making a list of your interests and skills, and do a self-assessment using some of the suggestions in this chapter.
  - Next you need to narrow the field of thousands of different franchise systems. At Franchise Handbook Online (http://www.franchisehandbook.com) , you'll find articles with checklists to help you thoroughly research a franchise and its industry, as well as a directory of franchise opportunities.
  - Armed with this information, develop a questionnaire to evaluate a prospective franchise. (Resources, Interpersonal, Information)
- **Working the Net:**
  - Select three franchises that interest you.
  - Research them at sites such as the Franchise Handbook Online (http://www.franchisehandbook.com), Entrepreneur magazine's Franchise 500 (http://www.entrepreneur.com), and Be the Boss (www.betheboss.com).

Writer-ER Kackery/Cajon/SBCUSD

- Prepare a chart comparing your selections, including history, number and location of units, financial requirements (initial franchise fee, other start-up costs, royalty and advertising fees), and any other information that would help you evaluate the franchises
- Present your findings

**SEMESTER 1: QUARTER 2**

**UNIT 5: Entrepreneurship: Starting and Managing Your Own Business**

**SWBAT TO ANSWER THESE LEARNING OUTCOMES:**
- Why do people become entrepreneurs, and what are the different types of entrepreneurs?
- What characteristics do successful entrepreneurs share? How do small businesses contribute to the U.S. economy?
- What are the first steps to take if you are starting your own business?
- Why does managing a small business present special challenges for the owner?
- What are the advantages and disadvantages facing owners of small businesses?
- How does the Small Business Administration help small businesses?
- What trends are shaping entrepreneurship and small-business ownership?

**Unit 5 Key Assignments:**
- **Preparing for Tomorrow's Workplace Skills:** *Interview*
  - What does it really take to become an entrepreneur?
  - Find out by interviewing a local entrepreneur or researching an entrepreneur you've read about in this chapter or in the business press.
  - Get answers to the following questions, as well as any others you'd like to ask:
    - How did you research the feasibility of your idea?
    - How did you develop your vision for the company?
    - How long did it take you to prepare your business plan?
    - Where did you obtain financing for the company?
    - Where did you learn the business skills you needed to run and grow the company?
    - What are the most important entrepreneurial characteristics that helped you succeed?
    - What were the biggest challenges you had to overcome?
    - What are the most important lessons you learned by starting this company?
    - What advice do you have for would-be entrepreneurs?
    -(Submission on Google Slides in Google Classroom & Present)
- **Preparing Working the Net:** *Business Plan*
  - Visit Sample Business Plans at http://www.bplans.com to review examples of all types of business plans.
  - Select an idea for a company in a field that interests you and, using information from the site, prepare an outline for its business plan.

**UNIT 6:** Management and Leadership in Today's Organizations

**SWBAT TO ANSWER THESE LEARNING OUTCOMES:**
- What is the role of management?
- What are the four types of planning?
- What are the primary functions of managers in organizing activities?
- How do leadership styles influence a corporate culture?
- How do organizations control activities?
- What roles do managers take on in different organizational settings?
- What set of managerial skills is necessary for managerial success?
- What trends will affect management in the future?

Writer-ER Kackery/Cajon/SBCUSD

**Unit 6 Key Assignments:**
- **Quiz #3-** Comprehension of Twenty Core Concepts from Unit 5 & 6
- **Preparing for Tomorrow's Workplace Skills:** Self Reflection (1 paragraph per question submitted into Google Classroom)
  - **Question 1:** Would you be a good manager? Do a self-assessment that includes your current technical, human relations, and conceptual skills. What skills do you presently possess, and which do you need to add? Where do your strengths lie? Based on this exercise, develop a description of an effective manager. (Resources, Information)
  - **Question 2:** Successful managers map out what they want to do with their time (planning), determine the activities and tasks they need to accomplish in that time frame (organizing), and make sure they stay on track (controlling). How well do you manage your time? Do you think ahead, or do you tend to procrastinate? Examine how you use your time, and identify at least three areas where you can improve your time management skills. (Resources)
  - **Question 3:** Often researchers cast leadership in an inspirational role in a company and management in more of an administrative role. That tendency seems to put leadership and management in a hierarchy. Do you think one is more important than the other? Do you think a company can succeed if it has bad managers and good leaders? What about if it has good managers and bad leaders? Are managers and leaders actually the same? (Systems)
  - **Question 4:** Today's managers must be comfortable using all kinds of technology. Do an inventory of your computer skills, and identify any gaps. After listing your areas of weakness, make a plan to increase your computer competency by enrolling in a computer class on or off campus. You may want to practice using common business applications such as Microsoft Excel by building a spreadsheet to track your budget, Microsoft PowerPoint by creating slides for your next class project, and Microsoft Outlook by uploading your semester schedule. (Information, Technology)

**UNIT 7: Designing Organizational Structures**

**SWBAT TO ANSWER THESE LEARNING OUTCOMES:**
- What are the traditional forms of organizational structure?
- What contemporary organizational structures are companies using?
- Why are companies using team-based organizational structures?
- What tools do companies use to establish relationships within their organizations?
- How can the degree of centralization/decentralization be altered to make an organization more successful?
- How do mechanistic and organic organizations differ?
- How does the informal organization affect the performance of the company?
- What trends are influencing the way businesses organize?

**Unit 7 Key Assignments:**
- **Unit 7 Ethics Activity-Ethics Report**; Training IT Replacement
  - Using a web search tool, locate articles about this topic and then write responses to the following questions. Be sure to support your arguments and cite your sources.
  - Ethical Dilemma: Are UCSF and other companies justified in outsourcing technology jobs to India? Do they have any obligation to find other jobs or provide training for displaced workers? Should organizations ask employees who are being laid off to train their replacements?
- **Working the Net**
  - Using a search engine, look for the term "company organizational charts," and find at least three examples of organizational charts for corporations, not-for-profits, or government agencies.
  - Analyze each entity's organizational structure. Is it organized by function, product/service, process, customer type, or geographic location?
  - Submit your answers onto the google docs graphic organizer provided.

**UNIT 8: Managing Human Resources and Labor Relations**

**SWBAT TO ANSWER THESE LEARNING OUTCOMES:**
- What is the human resource management process, and how are human resource needs determined?
- How do firms recruit applicants?
- How do firms select qualified applicants?

Writer-ER Kackery/Cajon/SBCUSD

---

**Question:** How many quizzes are there in units 4, 5, and 6 combined?
**Response:** There are three quizzes in units 4, 5, and 6 combined:
- Unit 4: Quiz 2: Twenty Core Concepts on Unit 3 & 4
- Unit 5: No quiz listed
- Unit 6: Quiz #3: Comprehension of Twenty Core Concepts from Unit 5 & 6
So, the total number of quizzes in units 4, 5, and 6 is 3.

**Error Type:** Reasoning Error
**Analysis:** The response analyze correctly the quizzes in each unit, but count their number wrongly at the last step.

Figure 32: Error example about *Reasoning Error*

**Irrelevant Answer: Case 1**

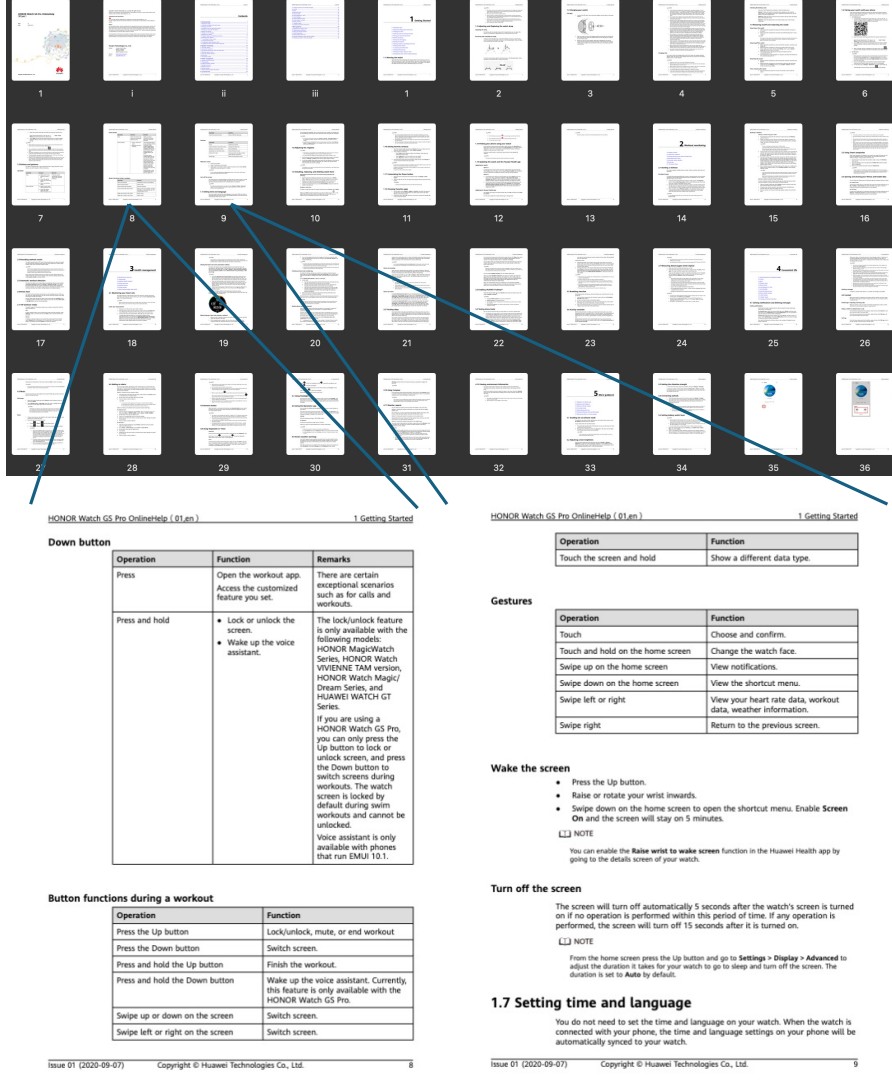

**Question:** How many distinct button functions are there during a workout?
**Response:** During a workout, the HONOR Watch GS Pro has the following distinct button functions:
1. Press the Up button: Lock, mute, or end workout.
2. Press the Down button: Switch screen.
3. Press and hold the Up button: Finish the workout.
4. Press and hold the Down button: Wake up the voice assistant.
5. Swipe up or down on the screen: Switch screen.
6. Swipe left or right on the screen: Switch screen.
So, there are 6 distinct button functions during a workout.

**Error Type:** Irrelevant Answer
**Analysis:** The question asks the button functions. The response, however, answers the button operations and outputs an irrelevant answer.

Figure 33: Error example about *Irrelevant Answer*

**Irrelevant Answer: Case 2**

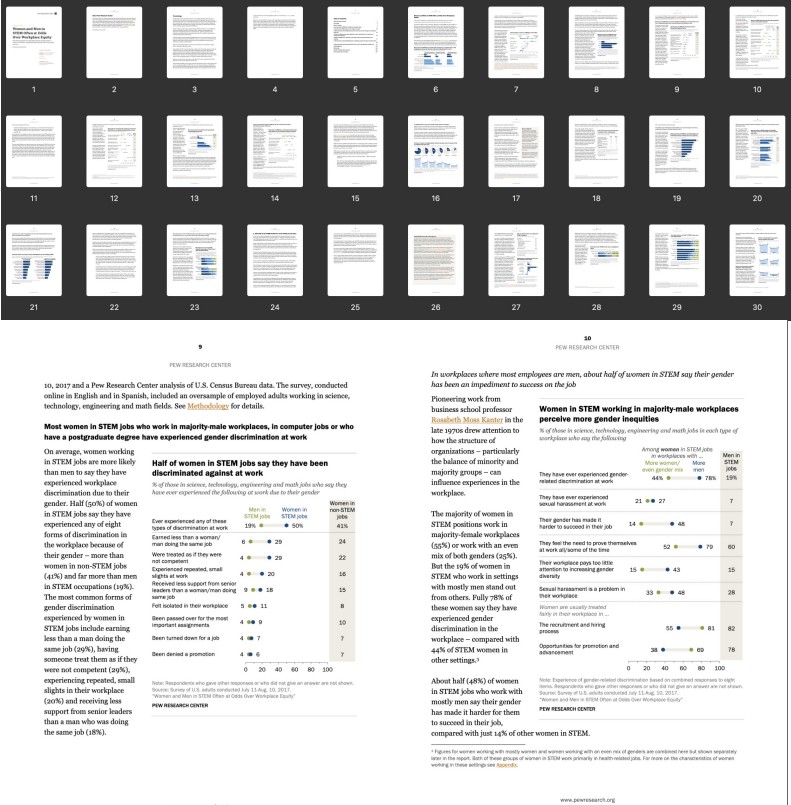

**Question:** What types of discrimination are most commonly reported by women in STEM jobs compared to men in the survey of Pew Research Center, January 2018.

**Response:** In the Pew Research Center survey titled "Women and Men in STEM Often at Odds Over Workplace Equity," published in January 2018, several types of discrimination were reported by women in STEM jobs compared to men, and by women in workplaces with a higher proportion of women. Here are the key findings:

1. **Gender Discrimination**:
   - **Overall Experience**: 50% of women in STEM jobs reported experiencing gender discrimination at work, compared to 19% of men in STEM jobs.
   - **Specific Types of Discrimination**: Women in STEM jobs reported various forms of gender discrimination, including earning less than a man doing the same job (29%), being treated as if they were not competent (29%), experiencing repeated small slights at work (20%), and receiving less support from senior leaders (18%).
......
5. **Advanced Degrees**:
   - **Higher Discrimination**: 62% of women in STEM jobs with a postgraduate degree reported experiencing gender discrimination, compared to 41% of women with some college or less education.
   - **Impact on Fair Treatment**: Women with advanced degrees were less likely to believe that women are treated fairly in opportunities for promotion and advancement.

These findings highlight the significant gender disparities and challenges that women in STEM fields face, particularly in male-dominated environments and specific job roles like computer-related positions.

**Error Type:** Irrelevant Answer
**Analysis:** The question asks about the MOST discrimination types, while response summarizes various types of discriminations towards women in STEM jobs.

Figure 34: Error example about *Irrelevant Answer*

## C.2 Case Study

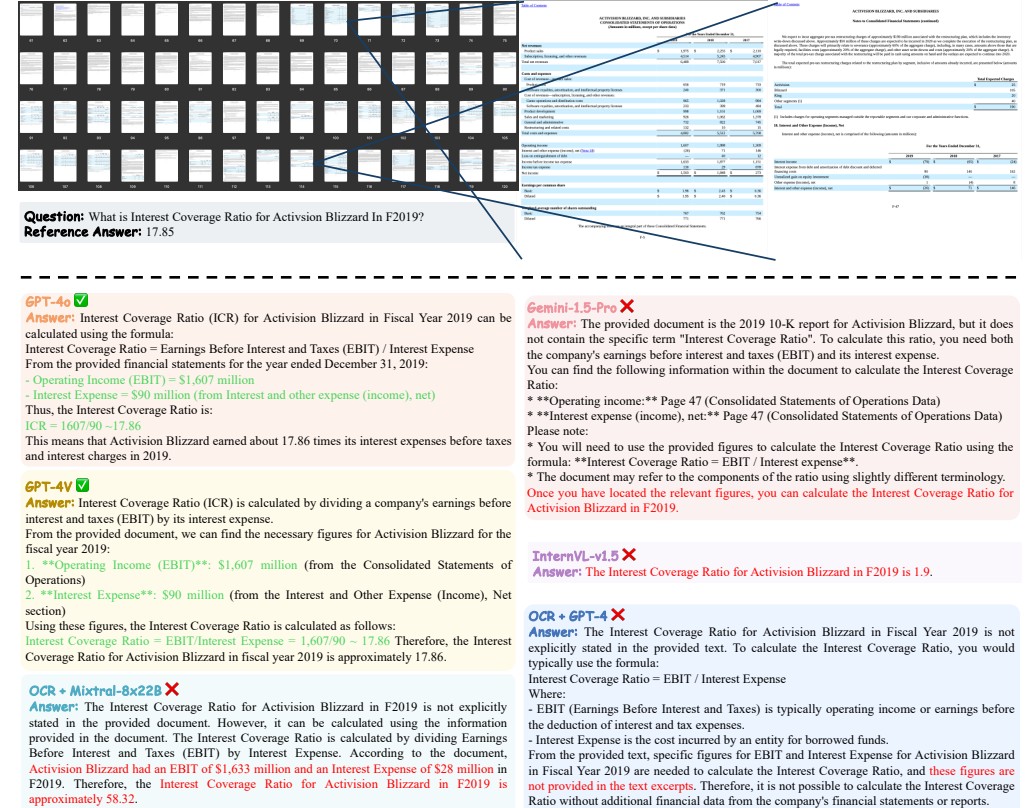

**Question:** What is Interest Coverage Ratio for Activsion Blizzard In F2019?
**Reference Answer:** 17.85

**GPT-4o** ✅
**Answer:** Interest Coverage Ratio (ICR) for Activision Blizzard in Fiscal Year 2019 can be calculated using the formula:
Interest Coverage Ratio = Earnings Before Interest and Taxes (EBIT) / Interest Expense
From the provided financial statements for the year ended December 31, 2019:
- Operating Income (EBIT) = $1,607 million
- Interest Expense = $90 million (from Interest and other expense (income), net)
Thus, the Interest Coverage Ratio is:
ICR = 1607/90 ~17.86
This means that Activision Blizzard earned about 17.86 times its interest expenses before taxes and interest charges in 2019.

**GPT-4V** ✅
**Answer:** Interest Coverage Ratio (ICR) is calculated by dividing a company's earnings before interest and taxes (EBIT) by its interest expense.
From the provided document, we can find the necessary figures for Activision Blizzard for the fiscal year 2019:
1. **Operating Income (EBIT)**: $1,607 million (from the Consolidated Statements of Operations)
2. **Interest Expense**: $90 million (from the Interest and Other Expense (Income), Net section)
Using these figures, the Interest Coverage Ratio is calculated as follows:
Interest Coverage Ratio = EBIT/Interest Expense = 1,607/90 ~ 17.86 Therefore, the Interest Coverage Ratio for Activision Blizzard in fiscal year 2019 is approximately 17.86.

**OCR + Mixtral-8x22B** ❌
**Answer:** The Interest Coverage Ratio for Activision Blizzard in F2019 is not explicitly stated in the provided document. However, it can be calculated using the information provided in the document. The Interest Coverage Ratio is calculated by dividing Earnings Before Interest and Taxes (EBIT) by Interest Expense. According to the document, Activision Blizzard had an EBIT of $1,633 million and an Interest Expense of $28 million in F2019. Therefore, the Interest Coverage Ratio for Activision Blizzard in F2019 is approximately 58.32.

**Gemini-1.5-Pro** ❌
**Answer:** The provided document is the 2019 10-K report for Activision Blizzard, but it does not contain the specific term "Interest Coverage Ratio". To calculate this ratio, you need both the company's earnings before interest and taxes (EBIT) and its interest expense.
You can find the following information within the document to calculate the Interest Coverage Ratio:
* **Operating income:** Page 47 (Consolidated Statements of Operations Data)
* **Interest expense (income), net:** Page 47 (Consolidated Statements of Operations Data)
Please note:
* You will need to use the provided figures to calculate the Interest Coverage Ratio using the formula: **Interest Coverage Ratio = EBIT / Interest expense**.
* The document may refer to the components of the ratio using slightly different terminology.
Once you have located the relevant figures, you can calculate the Interest Coverage Ratio for Activision Blizzard in F2019.

**InternVL-v1.5** ❌
**Answer:** The Interest Coverage Ratio for Activision Blizzard in F2019 is 1.9.

**OCR + GPT-4** ❌
**Answer:** The Interest Coverage Ratio for Activision Blizzard in Fiscal Year 2019 is not explicitly stated in the provided text. To calculate the Interest Coverage Ratio, you would typically use the formula:
Interest Coverage Ratio = EBIT / Interest Expense
Where:
- EBIT (Earnings Before Interest and Taxes) is typically operating income or earnings before the deduction of interest and tax expenses.
- Interest Expense is the cost incurred by an entity for borrowed funds.
From the provided text, specific figures for EBIT and Interest Expense for Activision Blizzard in Fiscal Year 2019 are needed to calculate the Interest Coverage Ratio, and these figures are not provided in the text excerpts. Therefore, it is not possible to calculate the Interest Coverage Ratio without additional financial data from the company's financial statements or reports.

Figure 35: Case Study. Evidence source: table. The evidence pages are zoomed in. The correct extracted information and reasoning are colored in green, and the wrong ones are colored in red.

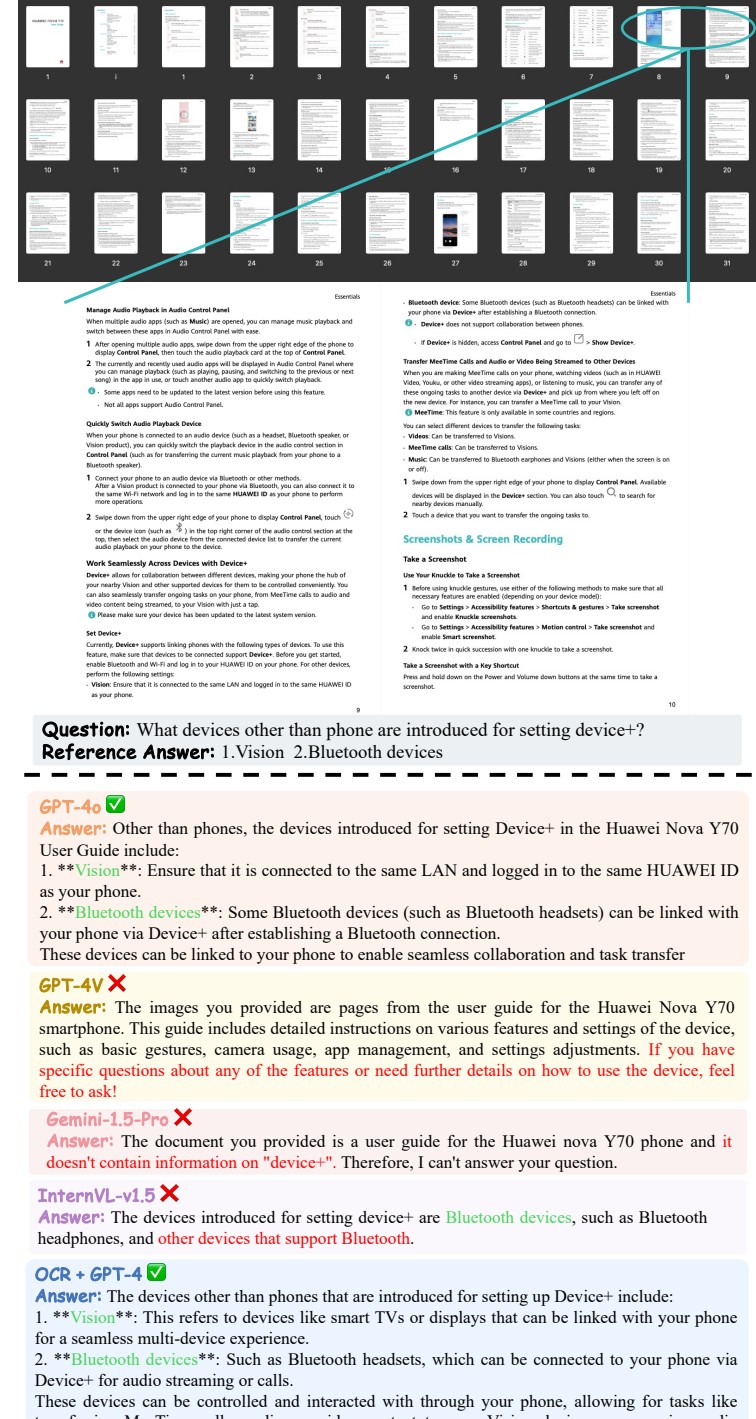

Question: What devices other than phone are introduced for setting device+?
Reference Answer: 1.Vision  2.Bluetooth devices

GPT-4o ✅
Answer: Other than phones, the devices introduced for setting Device+ in the Huawei Nova Y70 User Guide include:
1. **Vision**: Ensure that it is connected to the same LAN and logged in to the same HUAWEI ID as your phone.
2. **Bluetooth devices**: Some Bluetooth devices (such as Bluetooth headsets) can be linked with your phone via Device+ after establishing a Bluetooth connection.
These devices can be linked to your phone to enable seamless collaboration and task transfer

GPT-4V ❌
Answer: The images you provided are pages from the user guide for the Huawei Nova Y70 smartphone. This guide includes detailed instructions on various features and settings of the device, such as basic gestures, camera usage, app management, and settings adjustments. If you have specific questions about any of the features or need further details on how to use the device, feel free to ask!

Gemini-1.5-Pro ❌
Answer: The document you provided is a user guide for the Huawei nova Y70 phone and it doesn't contain information on "device+". Therefore, I can't answer your question.

InternVL-v1.5 ❌
Answer: The devices introduced for setting device+ are Bluetooth devices, such as Bluetooth headphones, and other devices that support Bluetooth.

OCR + GPT-4 ✅
Answer: The devices other than phones that are introduced for setting up Device+ include:
1. **Vision**: This refers to devices like smart TVs or displays that can be linked with your phone for a seamless multi-device experience.
2. **Bluetooth devices**: Such as Bluetooth headsets, which can be connected to your phone via Device+ for audio streaming or calls.
These devices can be controlled and interacted with through your phone, allowing for tasks like transferring MeeTime calls, audio, or video content to your Vision device, or managing audio playback through connected Bluetooth devices.

OCR + Mixtral-8x22B ✅
Answer: For setting Device+, ensure that devices to be connected support Device+. Before you get started, enable Bluetooth and Wi-Fi and log in to your HUAWEI ID on your phone. For other devices, perform the following settings:
+ Vision: Ensure that it is connected to the same LAN and logged in to the same HUAWEI ID as your phone.
+ Bluetooth device: Some Bluetooth devices (such as Bluetooth headsets) can be linked with your phone via Device+ after establishing a Bluetooth connection.
@ Device+ does not support collaboration between phones.
If Device+ is hidden, access Control Panel and go to > Show Device+.

Figure 36: Case Study. Evidence source: text. The evidence pages are zoomed in. The correct extracted information and reasoning are colored in green, and the wrong ones are colored in red.

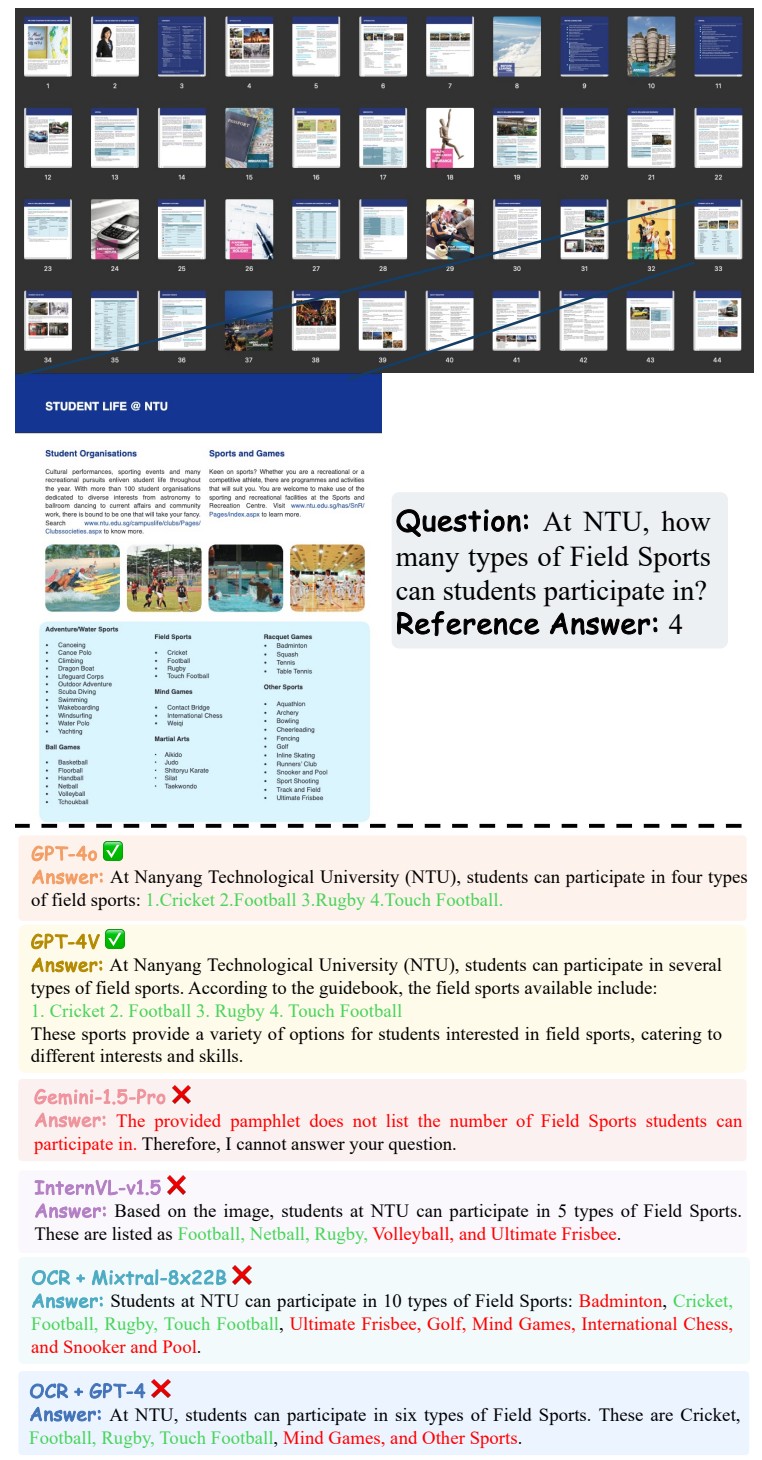

Figure 37: Case Study. Evidence source: layout. The evidence page is zoomed in. The correct extracted information and reasoning are colored in green, and the wrong ones are colored in red.

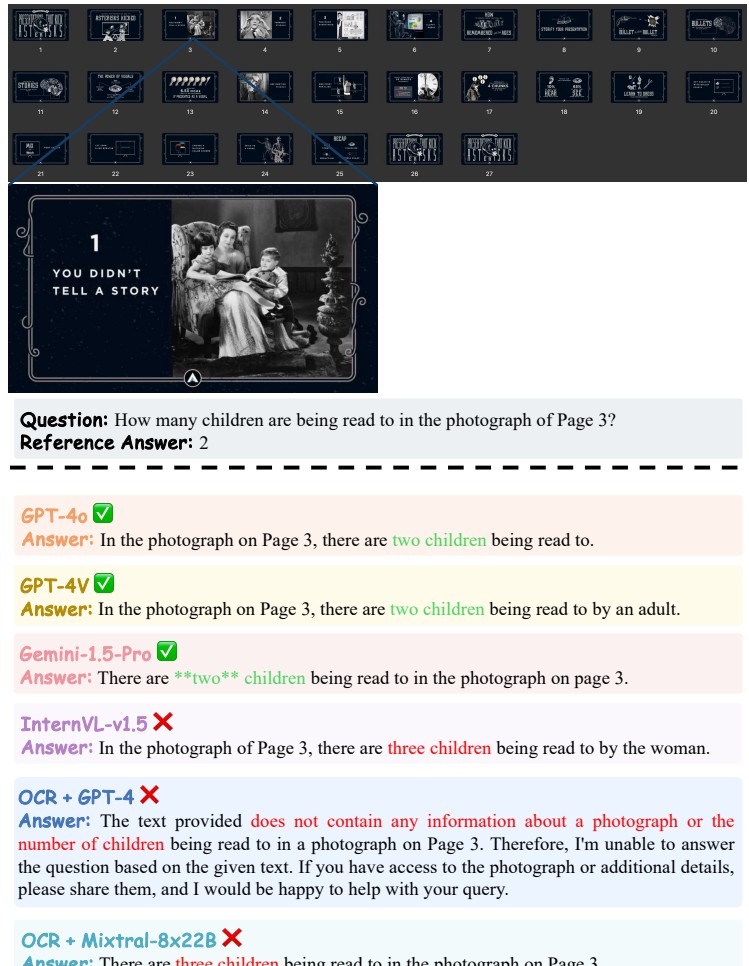

**Question:** How many children are being read to in the photograph of Page 3?
**Reference Answer:** 2

**GPT-4o** ✅
**Answer:** In the photograph on Page 3, there are two children being read to.

**GPT-4V** ✅
**Answer:** In the photograph on Page 3, there are two children being read to by an adult.

**Gemini-1.5-Pro** ✅
**Answer:** There are **two** children being read to in the photograph on page 3.

**InternVL-v1.5** ❌
**Answer:** In the photograph of Page 3, there are three children being read to by the woman.

**OCR + GPT-4** ❌
**Answer:** The text provided does not contain any information about a photograph or the number of children being read to in a photograph on Page 3. Therefore, I'm unable to answer the question based on the given text. If you have access to the photograph or additional details, please share them, and I would be happy to help with your query.

**OCR + Mixtral-8x22B** ❌
**Answer:** There are three children being read to in the photograph on Page 3.

Figure 38: Case Study. Evidence source: image. The evidence page is zoomed in. The correct extracted information and reasoning are colored in green, and the wrong ones are colored in red.

# D  Limitations

MMLONGBENCH-DOC is the first comprehensive benchmark designed to evaluate the long-context document understanding capabilities of LVLMs. While our benchmark addresses significant gaps in the previous datasets, we acknowledge several limitations.

One primary limitation is the scale of the benchmark. Currently, our benchmark includes a test set comprising 135 documents and 1,082 questions. It is much smaller compared to previous datasets. The complexity and difficulty of annotations limit the scale of our benchmark. As a long-context benchmark, our documents average about 50 pages and 20,000 tokens. And most questions require either complicated reasoning or cross-page comprehension. It takes more than one hour for an expert-level annotator to read through a single document, and then edit existing instances and create new instances on this document. Given the purpose of MMLONGBENCH-DOC as an evaluation benchmark, we prioritize annotation quality over quantity. Moreover, the results presented in Sections 3.3 and 3.4 confirm that the scale of our benchmark is sufficient for fine-grained evaluations across different document types, evidence sources, evidence pages, *etc.*. Additionally, we plan to expand our benchmark by adding more documents and questions in future iterations.

We roughly categorize these questions into three types, *i.e.,* single-page, cross-page, and unanswerable questions, based on whether evidence can be found in the documents and the number of evidence pages. However, unlike MMBench [41] or MathVista [56], we provide no further taxonomy to classify some (*e.g.,* 7 or 20) fine-grained, evaluated reasoning or perception capabilities out of two main reasons: (1) Prior (*i.e.,* pre-annotation) taxonomy limits the diversity of the questions. Therefore we provide no predefined classifications in our guideline and encourage the expert-level annotators to freely write questions without constraints. (2) The intrinsic complexity of document understanding presents significant challenges for establishing a posterior (*i.e.,* post-annotation) taxonomy.

While there exist limitations in our benchmark, MMLONGBENCH-DOC surely represents a significant step forward in this field. We would iteratively maintain and refine this benchmark and hope it could push forward the development of long-context document understanding.

# E  Social Impacts

The development and use of MMLONGBENCH-DOC may have potential societal implications. For instance, biased or inaccurate outputs from benchmarked models could perpetuate harmful stereotypes or reinforce existing social inequalities. Additionally, the ability to process and analyze long documents could potentially be used to surveil or monitor individuals' personal information. Developers and users of MMLONGBENCH-DOC benchmark must be aware of these potential consequences and take steps to ensure responsible development and deployment of AI models.

# F  Author Statement

The authors state that all of the previous datasets that we collected are licensed under the Creative Commons license (CC-BY) or other open-source licenses. Using this dataset should abide by the policy of OpenAI. Regarding the newly collected documents, we manually check them to ensure their availability for academic use. Should any authors request the removal of their documents, we will promptly comply.

