# Datasheet

**Yubo Ma**[1]**, Yuhang Zang**[2*]**, Liangyu Chen**[1]**, Meiqi Chen**[3]**, Yizhu Jiao**[4]
**Xinze Li**[1]**, Xinyuan Lu**[5]**, Ziyu Liu**[6]**, Yan Ma**[7]**, Xiaoyi Dong**[2]**, Pan Zhang**[2]
**Liangming Pan**[8]**, Yu-Gang Jiang**[9]**, Jiaqi Wang**[2]**, Yixin Cao**[9*]**, Aixin Sun**[1]

[1] S-Lab, Nanyang Technological University, [2] Shanghai AI Laboratory, [3] Peking University
[4] University of Illinois Urbana-Champaign, [5] National University of Singapore, [6] Wuhan University
[7] Singapore Management University, [8] University of Arizona, [9] Fudan University

## A    Motivation

*Q1: For what purpose was the dataset created? Was there a specific task in mind? Was there a specific gap that needed to be filled?*

**A1:** We construct MMLONGBENCH-DOC to evaluate the understanding capabilities of Large Vision-Lanaguage Models (LVLMs) on long-context, multi-modality documents. This benchmark targets document understanding (DU), which is a long-standing task in urgent and practical needs. As stated in Section 1, most previous datasets on DU focus on single-page DU. There lacks a unified, high-quality benchmark that (1) includes diverse lengthy documents and questions with detailed meta-data annotations and (2) evaluates how LVLMs perform on lengthy documents with multi-modality components (*e.g.,* plain text, table, chart, image). Our benchmark is constructed to bridge such a gap.

*Q2: Who created the dataset (e.g., which team, research group) and on behalf of which entity (e.g., company, institution, organization)?*

**A2:** MMLONGBENCH-DOC is created by a joint team from multiple academic institutions: Yubo Ma (S-Lab, Nanyang Technological University), Yuhang Zang (Shanghai AI Laboratory), Liangyu Chen (S-Lab, Nanyang Technological University), Meiqi Chen (Peking University), Yizhu Jiao (the University of Illinois Urbana-Champaign), Xinze Li (S-Lab, Nanyang Technological University), Xinyuan Lu (National University of Singapore), Ziyu Liu (Wuhan University), Yan Ma (Singapore Management University), Xiaoyi Dong (Shanghai AI Laboratory), Pan Zhang (Shanghai AI Laboratory), Liangming Pan (University of Arizona), Yu-Gang Jiang (Fudan University), Jiaqi Wang (Shanghai AI Laboratory), Yixin Cao (Fudan University), Aixin Sun (S-Lab, Nanyang Technological University).

*Q3: What support was needed to make this dataset?*

**A3:** This work was jointly funded by S-Lab, Nanyang Technological University and Shanghai AI Laboratory.

## B    Composition

*Q1: What do the instances that comprise the dataset represent (e.g., documents, photos, people, countries)? Are there multiple types of instances (e.g., movies, users, and ratings; people and interactions between them; nodes and edges)? How many instances are there in total (of each type, if appropriate)? What data does each instance consist of?*

**A1:** MMLONGBENCH-DOC incorporate a total of 1082 annotated questions from 135 collected lengthy documents across 7 types. Each instance includes a question, a document upon which the

---

*Corresponding Authors.

question is created, and a reference answer. Any meta-data including evidence pages, evidence sources, and answer formats is also provided.

***Q2****: Does the dataset contain all possible instances or is it a sample (not necessarily random) of instances from a larger set? If the dataset is a sample, then what is the larger set? Is the sample representative of the larger set (e.g., geographic coverage)? If so, please describe how this representativeness was validated/verified. If it is not representative of the larger set, please describe why not (e.g., to cover a more diverse range of instances, because instances were withheld or unavailable).*

**A2:** As a benchmark, MMLONGBENCH-DOC incorporates 76 (out of 135) documents and 184 (out of) 1082 questions from four previous datasets: DUDE [1], SlideVQA [2], ChartQA [3] and FinanceBench [4]. These existing documents and questions are not representative of the larger set because our benchmark stands as long-context, challenging document understanding. However, almost all documents in previous datasets are not enough long. We merely pick out some lengthy documents (taking a very small portion in their original datasets) and remove their unqualified questions from previous datasets.

***Q3****: Are relationships between individual instances made explicit (e.g., users' movie ratings, social network links)? If so, please describe how these relationships are made explicit.*

**A3:** N/A.

***Q4****: Are there recommended data splits (e.g., training, development/validation, testing)? If so, please provide a description of these splits, explaining the rationale behind them.*

**A4:** Our benchmark only has test instances.

***Q5****: Are there any errors, sources of noise, or redundancies in the dataset? If so, please provide a description.*

**A5:** We conduct a three-round quality control procedure, as described in Section 2.3 to reduce the annotation errors as much as possible. We believe that most annotation errors have been corrected. If new errors are found in the future, we will provide an erratum.

***Q6****: Is the dataset self-contained, or does it link to or otherwise rely on external resources (e.g., websites, tweets, other datasets)? If it links to or relies on external resources, a) are there guarantees that they will exist, and remain constant, over time; b) are there official archival versions of the complete dataset (i.e., including the external resources as they existed at the time the dataset was created); c) are there any restrictions (e.g., licenses, fees) associated with any of the external resources that might apply to a dataset consumer? Please provide descriptions of all external resources and any restrictions associated with them, as well as links or other access points, as appropriate.*

**A6:** Yes, this dataset is self-contained and licensed under CC BY-NC-SA 4.0.

***Q7****: Does the dataset contain data that might be considered confidential (e.g., data that is protected by legal privilege or by doctor-patient confidentiality, data that includes the content of individuals' non-public communications)? If so, please provide a description.*

**A7:** No.

***Q8****: Does the dataset contain data that, if viewed directly, might be offensive, insulting, threatening, or might otherwise cause anxiety? If so, please describe why.*

**A8:** No.

## C   Collection Process

***Q1****: How was the data associated with each instance acquired? Was the data directly observable (e.g., raw text, movie ratings), reported by subjects (e.g., survey responses), or indirectly inferred/derived from other data (e.g., part-of-speech tags, model-based guesses for age or language)? If the data was reported by subjects or indirectly inferred/derived from other data, was the data validated/verified? If so, please describe how.*

**A1:** We collect documents from both existing datasets and websites. Similarly, the questions are either from existing datasets or newly annotated. Both the documents (in PDF format) and the questions are directly observable.

*Q2: What mechanisms or procedures were used to collect the data (e.g., hardware apparatuses or sensors, manual human curation, software programs, software APIs)? How were these mechanisms or procedures validated?*

**A2:** Both the documents and the questions are manually collected or created. As stated in Section 2.3, we conduct quality control to validate the quality of our benchmark.

*Q3: If the dataset is a sample from a larger set, what was the sampling strategy (e.g., deterministic, probabilistic with specific sampling probabilities)?*

**A3:** To ensure the length of documents, we set a threshold (15 pages) to pick out documents from previous datasets. Regarding their original datasets, we also manually check their qualities and revise/remove unqualified ones as detailed in Section 2.2.

*Q4: Who was involved in the data collection process (e.g., students, crowdworkers, contractors) and how were they compensated (e.g., how much were crowdworkers paid)?*

**A4:** This benchmark is annotated by the authors of this paper. The data collection does not need compensation.

*Q5: Over what timeframe was the data collected? Does this timeframe match the creation timeframe of the data associated with the instances (e.g., recent crawl of old news articles)? If not, please describe the timeframe in which the data associated with the instances was created.*

**A5:** Our benchmark was created from April 2024 to June 2024.

*Q6: Were any ethical review processes conducted (e.g., by an institutional review board)? If so, please provide a description of these review processes, including the outcomes, as well as a link or other access point to any supporting documentation.*

**A6:** There is no need for ethical review.

## D Preprocessing/Cleaning/Labeling

*Q1: Was any preprocessing/cleaning/labeling of the data done (e.g., discretization or buck- eting, tokenization, part-of-speech tagging, SIFT feature extraction, removal of instances, processing of missing values)? If so, please provide a description. If not, you may skip the remaining questions in this section.*

**A1:** We manually select documents and their related questions as a part of our benchmark, as detailed in Section 2.1 and Section 2.2. Additionally, we add meta-labels for each instance, including the evidence pages, evidence sources, and answer formats.

*Q2: Was the "raw" data saved in addition to the preprocessed/cleaned/labeled data (e.g., to support unanticipated future uses)? If so, please provide a link or other access point to the "raw" data.*

**A2:** No.

*Q3: Is the software that was used to preprocess/clean/label the data available? If so, please provide a link or other access point.*

**A3:** No. Because we pre-process and label the data in a manual approach.

## E Uses

*Q1: Has the dataset been used for any tasks already? If so, please provide a description.*

**A1:** No.

*Q2: Is there a repository that links to any or all papers or systems that use the dataset? If so, please provide a link or other access point.*

**A2:** No, it is a new dataset.

*Q3: What (other) tasks could the dataset be used for?*

**A3:** This dataset could be used in the visual document understanding task. Multi-modality information retrieval might be another potential applied task since we annotate the evidence pages for each question.

*Q4: Is there anything about the composition of the dataset or the way it was collected and preprocessed/cleaned/labeled that might impact future uses? For example, is there anything that a dataset consumer might need to know to avoid uses that could result in unfair treatment of individuals or groups (e.g., stereotyping, quality of service issues) or other risks or harms (e.g., legal risks, financial harms)? If so, please provide a description. Is there anything a dataset consumer could do to mitigate these risks or harms?*

**A4:** N/A.

*Q5: Are there tasks for which the dataset should not be used? If so, please provide a description.*

**A5:** N/A.

## F    Distribution

*Q1: Will the dataset be distributed to third parties outside of the entity (e.g., company, institution, organization) on behalf of which the dataset was created? If so, please provide a description.*

**A1:** No.

*Q2: How will the dataset be distributed (e.g., tarball on the website, API, GitHub)? Does the dataset have a digital object identifier (DOI)?*

**A2:** This dataset will be distributed on website `https://mayubo2333.github.io/MMLongBench-Doc`. It doesn't have a DOI.

*Q3: Will the dataset be distributed under a copyright or other intellectual property (IP) license, and/or under applicable terms of use (ToU)? If so, please describe this license and/or ToU, and provide a link or other access point to, or otherwise reproduce, any relevant licensing terms or ToU, as well as any fees associated with these restrictions.*

**A3:** This dataset will be distributed under CC BY-NC-SA 4.0.

*Q4: Have any third parties imposed IP-based or other restrictions on the data associated with the instances? If so, please describe these restrictions, and provide a link or other access point to, or otherwise reproduce, any relevant licensing terms, as well as any fees associated with these restrictions.*

**A4:** No.

*Q5: Do any export controls or other regulatory restrictions apply to the dataset or to individual instances? If so, please describe these restrictions, and provide a link or other access point to, or otherwise reproduce, any supporting documentation.*

**A5:** No.

## G    Maintenance

*Q1: Who will be supporting/hosting/maintaining the dataset?*

**A1:** This dataset will be maintained by this paper's authors, mainly by Yubo Ma and Yuhang Zang.

*Q2: How can the owner/curator/manager of the dataset be contacted (e.g., email address)?*

**A2:** The manager of the dataset can be contacted via yubo001@e.ntu.edu.sg.

*Q3: Is there an erratum? If so, please provide a link or other access point.*

**A3:** No. If there exists an error, we will upload it on our website.

***Q4:*** *Will the dataset be updated (e.g., to correct labeling errors, add new instances, delete instances)? If so, please describe how often, by whom, and how updates will be communicated to dataset consumers (e.g., mailing list, GitHub)?*

**A4:** This benchmark may be updated to incorporate more documents and questions. If so, updates will be announced on the website `https://mayubo2333.github.io/MMLongBench-Doc`.

***Q5:*** *If the dataset relates to people, are there applicable limits on the retention of the data associated with the instances (e.g., were the individuals in question told that their data would be retained for a fixed period of time and then deleted)? If so, please describe these limits and explain how they will be enforced.*

**A5:** This dataset is not related to people.

***Q6:*** *Will older versions of the dataset continue to be supported/hosted/maintained? If so, please describe how. If not, please describe how its obsolescence will be communicated to dataset consumers.*

**A6:** If this benchmark were updated, the old version would serve as a subset of the whole dataset.

***Q7:*** *If others want to extend/augment/build on/contribute to the dataset, is there a mechanism for them to do so? If so, please provide a description. Will these contributions be validated/verified? If so, please describe how. If not, why not? Is there a process for communicating/distributing these contributions to dataset consumers? If so, please provide a description.*

**A7:** Feel free to use email or Github to contact us.