# OpenReview forum: "MMLONGBENCH-DOC: Benchmarking Long-context Document Understanding with Visualizations"
_NeurIPS.cc/2024/Datasets_and_Benchmarks_Track — NeurIPS 2024 Track Datasets and Benchmarks Spotlight_

### Official Review · Reviewer_yYQY · 2024-07-23
**Carefully built new long document understnading benchmark.**

**Rating:** 7
**Confidence:** 3
**Correctness:** yes
**Clarity:** very well written

**Review:**

The paper is very well-written providing motivation and relation to previous works. It also provides clear description of how the benchmark is constructed as well as how the authors ensured the quality of the dataset. However, their seems to be no description of how the benchmark will possibly be shared to the community.

**Strengths:**

- Clear description of how the benchmark was made.
- Clear description of how the quality of the dataset was checked.
- Solid evaluation using various commercial and open-source VLMs and LLMs.
- The analaysis providing good information.

**Additional Feedback:**

.

**Documentation:**

There should be the description of how the annotators were paid and how much annotation times were required.

**Ethics:**

there is no.

**Limitations:**

.

**Opportunities For Improvement:**

1. Although the paper is well-written, it provides a benchmark and as one of the major goal of the benchmark is to provide common ground, it would be great to be shared to the community. Thus, there should be a clear description of whether the benchmark will be shared or not.

2. The paper does not provide how the annotators were paid and how much man months were required to build the dataset.

**Relation To Prior Work:**

yes

**Summary And Contributions:**

The paper presents a new benchmark for assessing the long-document understanding ability of vision-language models (VLM). The dataset consists of 1k questions from 130 documents that consits of 50 pages on average.
The documents from various soures (Brochure, Finantial Report, Research Report, etc) were used and annotated by 10 Ph. D candidates or people with doctorial degrees. The resulting datasets are evaluated with 14 VLMs and LLMs showing the current best VLMs (GPT-4o) achieves ~40% F1 and found many models show lower performance compared to OCR + LLMs combinations.

---

> ### Author Rebuttal · Authors · 2024-08-16
>
> Thanks a lot for your reviews! Your professional reviews offer us great advice towards writing a more comprehensive and competitive paper! And, we appreciate your positive comments on the clear description about our benchmark and the insights of our analysis. Below are our responses to your questions.
>
> ---
>
> > Q1. Although the paper is well-written, it provides a benchmark and as one of the major goal of the benchmark is to provide common ground, it would be great to be shared to the community. Thus, there should be a clear description of whether the benchmark will be shared or not.
>
> * Thanks for your interest in our benchmark. Following the submission instruction, we provide the links of our dataset and project page in the attached supplementary material. The dataset has been shared with the community and used by some very recent work like [1]. We will add these links to the very beginning of our main paper (maybe in the abstract).
>
> Dataset link: https://huggingface.co/datasets/yubo2333/MMLongBench-Doc
>
> Project page: https://mayubo2333.github.io/MMLongBench-Doc
>
>
> > Q2. The paper does not provide how the annotators were paid and how much man months were required to build the dataset.
>
> * **Annotation Payment**: we provide related information in the attached supplementary material (Section H3, Q4-Q5; lines 466-469). Specifically, this benchmark is annotated by the authors of this paper. Therefore, the data collection does not need compensation.
>
> * **Annotation workload**: We count the time cost of our benchmark as below.
>     1. Pre-annotation (about 45h): the development of annotation interface (10h), the writing of annotation guideline (5h), training session (10h), preliminary annotation and personalized feedback (20h)
>     2. Annotation (about 150h): It takes about 60-90 minutes for the annotation of each document. And all of the 130 documents take about 150 hours.
>     3. Post-annotation (about 45h): quality checking (30h), data processing and release preparation (15h)
>
>     In summary, **our benchmark annotation approximately takes a total of 45+150+45=240 hours (1.36 man months)**. We will add this information to the Appendix of our paper.
>
> ---
>
> Thanks again for your comments and reading our responses. We hope the explanations above can address your concerns about our work. Shall you have any further question, free free to tell us and we will try our best to solve it.
>
> [1] Borchmann et al. Arctic-TILT. Business Document Understanding at Sub-Billion Scale. arXiv 2024.

---

> > ### Comment · Reviewer_yYQY · 2024-08-23
> >
> > Thank you for the clarification. I have adjusted the score accordingly.

---

### Official Review · Reviewer_u61o · 2024-07-23
**Review of "MMLONGBENCH-DOC: Benchmarking Long-context Document Understanding with Visualizations"**

**Rating:** 9
**Confidence:** 3
**Clarity:** Yes.

**Review:**

Quality
The quality of this work is high. The paper is well-structured, providing a clear and comprehensive overview of the problem it addresses. The authors have conducted extensive experiments and provided a detailed analysis of their results. The benchmark they propose, MMLONGBENCH-DOC, is well-designed, incorporating a diverse set of lengthy documents and a variety of question types to thoroughly evaluate the capabilities of Large Vision-Language Models (LVLMs) in long-context document understanding.

Clarity
The paper is clearly written and easy to follow. The introduction effectively sets the stage for the work, explaining the importance of document understanding and the challenges posed by long-context documents. The methodology section provides a detailed account of how the benchmark was constructed, including the selection and annotation of documents and questions. The evaluation protocol is well-explained, and the results are presented in a clear and organized manner, with helpful tables and figures to illustrate key points.

Originality
The originality of this work lies in its focus on long-context document understanding, a relatively unexplored area in the field of document understanding. While previous datasets have primarily focused on single-page documents, MMLONGBENCH-DOC addresses the unique challenges posed by documents that span tens or even hundreds of pages and contain multi-modal information. The authors' approach to benchmarking LVLMs on such documents is novel and provides valuable insights into the current limitations and future directions for research in this area.

Significance
The significance of this work is considerable. By highlighting the limitations of current LVLMs in handling long-context documents, the paper underscores the need for further research and development in this area. The benchmark dataset provided by the authors is a valuable resource that can be used by the research community to develop and evaluate new models and techniques for long-context document understanding. The detailed analysis of the experimental results also provides important insights that can guide future work in this field.

Pros and Cons
Pros:
Comprehensive Benchmark: MMLONGBENCH-DOC is a well-designed benchmark that includes a diverse set of lengthy documents and a variety of question types, providing a thorough evaluation of LVLMs' capabilities.
Detailed Analysis: The authors provide a detailed analysis of the experimental results, identifying key challenges and areas for improvement in long-context document understanding.
Clear Presentation: The paper is well-written and clearly presented, making it easy to follow and understand the authors' methodology and findings.
Novel Focus: The focus on long-context document understanding is novel and addresses an important gap in the current literature.
Significant Findings: The paper provides valuable insights into the limitations of current LVLMs and highlights the need for further research in this area.
Cons:
Evaluation Metrics: The evaluation metrics used are appropriate, but additional metrics or more detailed breakdowns of existing metrics could provide further insights into model performance.
Potential for Improvement: The authors acknowledge the need for further research, but the paper could benefit from more specific suggestions for future work and potential improvements to the benchmark and evaluation protocols.
Overall, this paper makes a significant contribution to the field of document understanding by addressing the challenges posed by long-context documents and providing a valuable benchmark for evaluating LVLMs in this context.

**Strengths:**

### Strengths of the Submission

#### Significance of the Contribution
1. **Addressing Long-Context Document Understanding**: The submission tackles the challenging and relatively unexplored area of long-context document understanding. This is significant as it extends beyond the capabilities of current models that primarily focus on single-page documents.
2. **Comprehensive Benchmark**: The introduction of MMLONGBENCH-DOC provides a significant resource for evaluating and advancing the capabilities of Large Vision-Language Models (LVLMs) in understanding lengthy, multi-modal documents. This benchmark sets a new standard for future research in this area.

#### Relevance to the Broader Research Community
1. **Diverse Dataset**: The benchmark includes a diverse set of documents from various domains, making it relevant to a wide range of applications, including academic research, financial reporting, and administrative documentation.
2. **Evaluation of State-of-the-Art Models**: The paper evaluates several state-of-the-art models, both proprietary and open-source, providing valuable insights into their strengths and limitations. This is highly relevant for researchers and practitioners looking to understand the current landscape and identify areas for improvement.

#### Quality of the Research
1. **Rigorous Methodology**: The authors employ a rigorous methodology in constructing the benchmark, including detailed document and question selection, expert annotation, and multi-round quality control. This ensures the reliability and validity of the benchmark.
2. **Extensive Experiments**: The research includes extensive experiments and a thorough analysis of results, providing a clear picture of the current capabilities of LVLMs in long-context document understanding. The detailed error analysis further enhances the quality of the research by identifying specific areas where models struggle.

#### Ethical and Social Implications
1. **High-Quality Annotations**: The use of expert annotators and a comprehensive quality control process ensures high-quality annotations, which is crucial for the ethical use of the benchmark in developing and evaluating models.
2. **Addressing Hallucinations and Errors**: By including unanswerable questions and analyzing model hallucinations, the research highlights the importance of addressing ethical issues related to the accuracy and reliability of LVLMs. This focus is essential for developing trustworthy AI systems that can be used in real-world applications without causing harm due to incorrect information.
3. **Transparency and Reproducibility**: The authors provide detailed descriptions of their methods and include supplementary material, ensuring transparency and reproducibility of their work. This is a key ethical consideration in research, promoting openness and enabling other researchers to build on their work.

Overall, the submission demonstrates significant strengths in terms of its contribution to the field, relevance to the broader research community, high quality of research, and consideration of ethical and social implications.

**Additional Feedback:**

See previous sections.

**Correctness:**

### Evaluation of Claims and Benchmark Construction

#### Correctness of Claims
The claims made in the submission appear to be well-supported by the provided data and analysis. The authors claim that long-context document understanding is a challenging and relatively unexplored area, and their results demonstrating the limitations of current LVLMs in this domain support this assertion. They also claim that their benchmark, MMLONGBENCH-DOC, provides a comprehensive evaluation of LVLMs, which is corroborated by the detailed construction and diverse set of documents and questions included in the benchmark.

#### Sound Construction of the Dataset
The dataset is constructed in a sound and methodical way. The authors provide a clear and detailed description of their document collection process, which includes selecting documents from various domains and ensuring diversity in document types and lengths. They also describe their annotation process, which involves expert annotators and multiple rounds of quality control to ensure high-quality annotations. This meticulous approach ensures the reliability and validity of the benchmark dataset.

#### Appropriateness of Evaluation Methods and Experiment Design
The evaluation methods and experiment design are appropriate and performed correctly. The authors follow a three-step evaluation protocol involving response generation, answer extraction, and score calculation, which is suitable for assessing long-context document understanding. Their experiments are extensive and involve a wide range of both proprietary and open-source models, providing a comprehensive comparison of model performance.

#### Constructive Feedback for Improvement

1. **Detailed Justification of Claims**:
   - **Improvement Suggestion**: While the claims are generally supported by the data, the authors could provide more detailed justifications for certain assertions, such as the specific advantages of their benchmark over existing datasets. Including comparative analyses with other benchmarks would strengthen their claims.

2. **Dataset Annotation Details**:
   - **Improvement Suggestion**: The authors provide a good overview of their annotation process, but additional details on the training and calibration of annotators, as well as inter-annotator agreement statistics, would enhance the transparency and credibility of the dataset construction.

3. **Error Analysis Expansion**:
   - **Improvement Suggestion**: The error analysis is valuable, but it could be expanded to include more specific examples and a deeper exploration of the types of errors encountered. This would provide clearer insights into the specific challenges faced by LVLMs and guide future improvements.

4. **Evaluation Metrics**:
   - **Improvement Suggestion**: While the evaluation metrics used are appropriate, introducing additional metrics or providing a more granular breakdown of existing metrics could offer further insights into model performance. Metrics such as recall, precision, and more detailed breakdowns by question type and document type would be useful.

### Conclusion
Overall, the claims made in the submission are correct, and the dataset is constructed in a sound manner. The evaluation methods and experiment design are appropriate and performed correctly. By incorporating the suggested improvements, the authors can further enhance the robustness and impact of their work.

**Documentation:**

Yes.

**Ethics:**

No.

**Limitations:**

### Addressing Limitations and Potential Negative Societal Impact

#### Adequacy of Addressing Limitations
The authors have made a commendable effort to address some of the limitations and potential negative societal impacts of their work. They provide a thorough discussion on the challenges posed by long-context document understanding and the limitations of current LVLMs. However, there are areas where the discussion could be expanded and made more comprehensive.

#### Constructive Suggestions for Improvement

1. **Detailed Error Analysis**:
   - **Current State**: The paper includes an error analysis that categorizes errors but lacks in-depth explanations for each type.
   - **Improvement Suggestion**: Expand the error analysis to provide detailed case studies and specific examples of common errors. This could include visual examples and a step-by-step breakdown of why certain errors occur, which would be highly beneficial for researchers looking to improve model performance.

2. **Bias and Fairness**:
   - **Current State**: The paper does not address potential biases in the dataset or the models.
   - **Improvement Suggestion**: Include a section dedicated to bias and fairness. This could involve analyzing the dataset for potential biases (e.g., in document types, question distribution) and discussing how these biases might affect model performance. Propose strategies for mitigating these biases, such as ensuring diverse representation in the dataset and using fairness metrics to evaluate models.

3. **Scalability and Accessibility**:
   - **Current State**: The benchmark’s complexity and resource requirements may limit its accessibility.
   - **Improvement Suggestion**: Provide guidelines or best practices for implementing the benchmark on different scales of resources. This could include recommendations for lower-resource settings and potential collaborations with cloud service providers to offer access to necessary computational resources.

4. **Ethical and Social Implications**:
   - **Current State**: The paper mentions the need to address hallucinations and errors but does not delve deeply into the broader ethical and social implications.
   - **Improvement Suggestion**: Expand the discussion on the potential long-term impacts of deploying advanced document understanding systems. Address issues such as the implications for privacy, security, and the potential for misuse. Discuss safeguards and ethical guidelines for the responsible use of these technologies in various applications.

5. **Future Directions**:
   - **Current State**: The paper acknowledges the need for further research but lacks specific suggestions.
   - **Improvement Suggestion**: Provide concrete recommendations for future research directions. This could include exploring new model architectures, developing specialized datasets for particular document types, and investigating cross-disciplinary approaches that combine document understanding with other fields such as natural language processing and computer vision.

#### Rewarding Upfront Acknowledgment
The authors should be commended for their upfront acknowledgment of the limitations and potential societal impacts of their work. This transparency is essential for fostering a responsible and ethical research community. By addressing these critical points and providing constructive feedback, the authors can further enhance the impact and relevance of their research.

### Summary
Overall, while the authors have made significant strides in addressing some limitations and societal impacts, there is room for a more comprehensive discussion and more detailed recommendations. Expanding on these areas will not only strengthen the paper but also provide valuable guidance for the research community moving forward.

**Opportunities For Improvement:**

### Limitations of the Work

#### Significance of the Contribution
1. **Focus on Existing Models**: While the benchmark is designed to address long-context document understanding, the paper primarily evaluates existing LVLMs. The contribution would be more significant if it included innovative methods or new models specifically designed to tackle the identified challenges.

#### Relevance to the Broader Research Community
1. **Implementation Complexity**: The complexity of the benchmark and the evaluation protocols may pose challenges for broader adoption within the research community. Researchers may find it difficult to replicate the results or integrate the benchmark into their own work without significant effort.
2. **Limited Scope of Models**: The study evaluates a relatively small set of models. Including a broader range of models, particularly newer or more diverse architectures, would enhance the relevance and applicability of the findings across different research areas.

#### Quality of the Research
1. **Limited Error Analysis**: While the error analysis provides valuable insights, it could be more detailed. A deeper analysis of specific types of errors and their root causes would offer more actionable insights for improving model performance.
2. **Evaluation Metrics**: The evaluation relies on generalized accuracy and F1 scores. Additional metrics or a more granular breakdown of existing metrics could provide further insights into model performance, especially regarding different document types and question categories.

#### Ethical and Social Implications
1. **Bias and Fairness**: The paper does not address potential biases in the benchmark dataset or the evaluated models. Understanding and mitigating biases is crucial for developing fair and ethical AI systems, particularly in document understanding tasks that may impact diverse user groups.
2. **Scalability and Accessibility**: The benchmark's complexity and the resources required for its implementation may limit its accessibility to researchers with fewer computational resources. This could hinder the democratization of research in this area and limit contributions from a more diverse set of researchers.
3. **Long-term Impact**: While the paper emphasizes the need for better LVLMs, it does not fully explore the long-term ethical and social implications of deploying such models in real-world applications. A discussion on the potential risks and benefits of widespread use of advanced document understanding systems would provide a more balanced perspective.

Overall, while the paper makes a significant contribution to the field, there are limitations in terms of its scope, implementation complexity, depth of analysis, and consideration of ethical and social factors. Addressing these limitations in future work would further enhance the impact and relevance of this research.

**Relation To Prior Work:**

Yes.

**Summary And Contributions:**

This paper addresses the challenge of understanding documents with rich layouts and multi-modal components, a long-standing task in document understanding (DU). Recent advancements in Large Vision-Language Models (LVLMs) have shown promise in single-page DU tasks. However, the ability of these models to handle long-context DU remains underexplored due to the lack of appropriate benchmarks.

To fill this gap, the authors introduce MMLONGBENCH-DOC, a comprehensive benchmark for long-context, multi-modality document understanding. The benchmark is based on 130 lengthy documents, each averaging 49.4 pages and 20,971 tokens. It includes 1,062 expert-annotated questions designed to evaluate LVLMs' capabilities across three key aspects:

Information Identification: 44.0% single-page questions.
Cross-Page Comprehension: 33.2% cross-page questions.
Hallucination Severity: 22.8% unanswerable questions.
These questions span diverse evidence sources, including text, images, charts, tables, and layout structures.

Experiments conducted on 14 LVLMs reveal significant challenges in long-context DU, with the best-performing model, GPT-4o, achieving an F1 score of 42.7%, and GPT-4V scoring 31.4%. Additionally, most LVLMs underperform compared to single-modality Language Models (LLMs) using OCR-parsed documents.

These findings highlight the necessity for further research to improve long-context LVLMs, emphasizing the importance of the MMLONGBENCH-DOC benchmark in advancing this field.

---

> ### Author Rebuttal · Authors · 2024-08-16
>
> Thanks a lot for your reviews! Your professional reviews offer us great advice towards writing a more comprehensive and competitive paper! And, we appreciate your positive comments on the significance and originality of our benchmark. Below are our responses to your questions.
>
> ---
>
> > Q1. Focus on Existing Models: The contribution would be more significant if it included innovative methods or new models specifically designed to tackle the identified challenges.
>
> * The development of innovative models heavily rely on the existence of benchmark. To our best knowledge, MMLongBench-Doc is the first benchmark to evaluate the long-context document understanding abilities of current LVLMs. And we believe that an innovative, appropriate benchmark and extensive experiments have been sufficient contributions for a paper in the dataset track.
>
> * We appreciate your constructive suggestion for developing an innovative, open-source LVLMs towards this task. Accordingly, we will add a section in the appendix to discuss the potential directions of innovative methods, and leave its development as our near-future work.
>
>
> > Q2. Implementation Complexity: The complexity of the benchmark and the evaluation protocols may pose challenges for broader adoption within the research community.
>
> We fully agree with you about the importance of usability for a benchmark. Therefore, we have released the code (including evaluation part) in our [official codebase](https://github.com/mayubo2333/MMLongBench-Doc). Moreover, we have integrated our benchmark into [VLMEvalKit](https://github.com/open-compass/VLMEvalKit)[1] for the convenience of unified and fair evaluation.
>
> > Q3. Limited Scope of Models: The study evaluates a relatively small set of models. Including a broader range of models, particularly newer or more diverse architectures, would enhance the relevance and applicability of the findings across different research areas.
>
> We evaluate the long-context document understanding abilities on 14 LVLMs and 10 LLMs. From our understanding, the set of models is representative and broad. Please feel free to tell us the models which you believe are necessary for our evaluation, and we are glad to add them during the discussion phase.
>
> > Q4. Limited Error Analysis: While the error analysis provides valuable insights, it could be more detailed. A deeper analysis of specific types of errors and their root causes would offer more actionable insights for improving model performance.
>
> We agree with you about the importance of visualized cases to illustrate the challenges of current LVLMs. We have a more detailed error analysis in the attached supplementary material (Section D.1), including descriptions and cases for each error reason. Specifically, the cases are well-visualized and contain step-by-step breakdowns about why certain errors occur. We will provide a clearer link about these detailed analysis in Section 4.2.
>
> > Q5. Evaluation Metrics: Additional metrics or a more granular breakdown of existing metrics could provide further insights into model performance, especially regarding different document types and question categories.
>
> * **Additional metrics**: The reference answers in our benchmark are short-form and deterministic. From our understanding, accuracy and F1 score are sufficient to evaluate such kinds of questions.
> * **Metrics categorized by different document and question types**: We do report document- and question-categorized evaluation results in Table 3, and discuss the performance difference across different documents/questions in Section 3.4.
>
> > Q6. Ethical and Social Implications: The paper mentions the need to address hallucinations and errors but does not delve deeply into the broader ethical and social implications.
>
> We truly thank your suggestions for a deeper and more broad discussion about the ethical and social implications of our benchmark. We have discussed the **Biases or Fairness** in the attached Supplementary Material (Section F). For further improvement, we have revised the discussion to add new parts about the (1) **Scalability and Accessibility** and (2) **Long-term Impact**, and will update this discussion in the final paper.
>
> > Q7. Dataset Annotation Details: The authors provide a good overview of their annotation process, but additional details on the training and calibration of annotators, as well as inter-annotator agreement statistics, would enhance the transparency and credibility of the dataset construction.
>
> * **The training and calibration of annotators**: Thanks for your suggestion and we will add our annotation instructions to the Appendix.
> * **Inter-annotator agreement statistics**: We calculate the Cohen’s kappa value as 0.42 (17.5% annotation inconsistency) during the quality control procedure (Section 2.3; lines 148-152), showing a moderate agreement. Given the high challenges of long-context document understanding, we believe this annotation agreement is acceptable.
>
>
> ---
>
> Thanks again for your comments and reading our responses. We hope the explanations above can address your concerns about our work. Shall you have any further question, free free to tell us and we will try our best to solve it.
>
> [1] Duan et al. VLMEvalKit: An Open-Source Toolkit for Evaluating Large Multi-Modality Models. arXiv 2024.
>
> [2] DeepSeek-AI. DeepSeek-V2: A Strong, Economical, and Efficient Mixture-of-Experts Language Model. arXiv 2024.
>
> [3] Gu et al. Mamba: Linear-Time Sequence Modeling with Selective State Spaces. arXiv 2024.

---

### Official Review · Reviewer_xHZL · 2024-07-25
**MMLONGBENCH-DOC**

**Rating:** 7
**Confidence:** 3

**Review:**

This paper presents a benchmark to assess L(V)LMs (large (vision-) language models) document understanding capabilities for question answering when long-range dependencies are present. The multimodal benchmark datasets consists of 1062 human-curated questions  to which the answer requires information spanning a single or multiple pages; unanswerable questions are included to catch hallucinations or shortcuts. The questions are sourced originally created or sourced selectively from other datasets spanning 130 documents with 6422 pages containing.

**Strengths:**

- **Thorough annotation pipeline**: Human annotators are highly trained (Ph.D. students) and appear trained and sufficiently incentivized to generate high-accuracy label. Moreover, reasonable measures are in place to ensure high quality (cross-checking).

- **document length** Annotating long-form documents is a balancing act between selecting documents that are too short (and therefore, trivial) or too long (failing to represent the distribution of documents of that category accurately due to an overly adverse selection). The document length averaging around 49 pages is fairly sizable but very reasonable IMO. For reference: scientific documents (when averaged over diverse disciplines though) span approximately 15 pages albeit at a higher information density and many disciplines with outlier page counts (e.g. medicine).

- **Granular evaluation** across document, question types, and modality of the LLM. Evaluation included closed and open sourced models that reflect the model landscape (at the time of the submission) very well. Including conventional LLMs through OCR is an important baseline that the authors included.

- **Challenging nature of the dataset**: The experiment reveals that the benchmark dataset is very challenging (e.g. generalized accuracy below 41%). On the one hand, this offers important insights into the capabilities of LVLMs (or the lack thereof for long-range DU). On the other hand, it speaks to the benchmark's potential longevity.

**Additional Feedback:**

n/a

**Clarity:**

The paper is well-written. It leverages textual explanations and plots very skillfully to describe the dataset (Figure 3), the results of the empirical investigation in a highly granular way (Table 3) and even failure modes of the LVLM responses (Figure 7). Still, the paper paper is not wordy but appears very crips.

**Correctness:**

Line 101: “over-simple”, “over-difficult” → “overly simple”, “overly difficult”.

Claims are overall well-backed by the empirical investigation conducted in the paper.

**Documentation:**

The authors provide extensive suppl. material on the data curation and empirical investigation.

**Limitations:**

The authors outline limitations of LVLMs clearly.

**Opportunities For Improvement:**

- **Limited number of documents per (one) category**: Since the majority of documents (76/130) is sourced from other datasets, namely, DUDE, SlideVQA, ChartQA, FinanceBench it is reasonable to augment categories that are insufficiently diverse. Academic papers in particular vary drastically among the scientific discipline they touch upon. However, they are underrepresented as compared to research reports. Understanding how LVLMs perform in DU on different domains (ideally from astrology to zoology) requires more questions that relate to these subdomains. Therefore, augmenting the data with questions relating to those categories would greatly increase the appeal and granularity of the empirical investigation.

- **Figure 1** is the opening visualization of the paper. Unfortunately, it is hard to make out what it tries to communicate even when zooming considerably. Streamlining the visualization would greatly increase the appeal and clarity.

**Relation To Prior Work:**

The paper clearly indicates its relation to previous benchmark datasets. Moreover, it actively leverages DUDE, SlideVQA, ChartQA, and FinanceBench for assembling `MMLONGBENCH-DOC`.

**Summary And Contributions:**

Contributions are two-fold:
(1.) Multimodal QA benchmark of 1062 human-curated questions that relate to 130 documents (6422 pages) to which the answer - if answerable - can span multiple document pages.
(2.) An empirical investigation of state-of-the-art LVLMs [large (vision-)language models] on the dataset to assess their performance in document understanding (DU) when the answer requires evidence from diverse sources and spans potentially multiple pages.

---

> ### Author Rebuttal · Authors · 2024-08-16
>
> Thanks a lot for your reviews! Your professional reviews offer us great advice towards writing a more comprehensive and competitive paper! And, we appreciate you finding our benchmark high-quality and challenging, and our evaluation extensive. Below are our responses to your questions.
>
> ---
>
> > Q1. Limited number of documents per (one) category:  Since the majority of documents (76/130) is sourced from other datasets, namely, DUDE, SlideVQA, ChartQA, FinanceBench it is reasonable to augment categories that are insufficiently diverse. Academic papers in particular vary drastically among the scientific discipline they touch upon. However, they are underrepresented as compared to research reports. Understanding how LVLMs perform in DU on different domains (ideally from astrology to zoology) requires more questions that relate to these sub-domains. Therefore, augmenting the data with questions relating to those categories would greatly increase the appeal and granularity of the empirical investigation
>
> * We agree about the necessity to expand the diversity of this benchmark. To this end, in addition to 76 documents from previous datasets, we newly collect 54 documents to include more academic papers, brochures and guidelines (as detailed in Section 2.1).
> * We agree that the category distribution of our collected documents can be further optimized, especially increasing the number and (sub-)category of scientific papers. We will update our benchmark to include more scientific papers, **with larger number and better diversity**, in the near future.
>
> > Q2. Figure 1 is the opening visualization of the paper. Unfortunately, it is hard to make out what it tries to communicate even when zooming considerably. Streamlining the visualization would greatly increase the appeal and clarity.
>
> * Figure 1 is composed of two subfigures: (1) a visulized example incorporating one document and three questions constructed from this document. In this example, the cross-modality document (top-left corner) has 40 pages. Regarding three questions, we highlight each of them in a grey block and additionally show their (zoomed-in) evidence pages, reference answers, etc.. (2) a performance comparison between OCR+LLM and LVLM to illustrate the current situation and challenges of LVLMs for document understanding.
>
> * Thanks for your constructive suggestion for further improving the appeal and clarity of this figure. We merge two sub-figures mentioned above into one opening figure due to page limitations and somewhat undermine its clarity and readability. In our next-version paper, we will split it into two separate figures (each with larger scale and more explanations) for convenience of visualization.
>
> ---
>
> We will also correct the typos you mentioned in Line 101. Thanks again for your comments and reading our responses. We hope the explanations above can address your concerns about our work. Shall you have any further question, free free to tell us and we will try our best to solve it.

---

> > ### Comment · Reviewer_xHZL · 2024-08-27
> > **Re:**
> >
> > Thank you for the rebuttal.
> >
> > > Q1
> >
> > A more diverse dataset of larger sample size is appreciated.
> >
> > > Q2
> >
> > In our next-version paper, we will split it into two separate figures.
> >
> > I believe these improvements will help the paper to get a greater audience.

---

### Official Review · Reviewer_ecPu · 2024-07-25
**New official review**

**Rating:** 7
**Confidence:** 4
**Correctness:** Yes, the metrics and baselines look c…
**Clarity:** Yes.

**Review:**

Please see my feedback under each section below.

**Strengths:**

1. Focuses on document understanding, which is an important use case of vision-language models. As far as I’m aware, this is one of the first works that studies the long-context capability of large multi-modal models.
2. The dataset is relatively large-scale, covering diverse domains a multiple types of questions. The annotations are mostly done by humans, with several quality control measurements.
3. The evaluation considers both open-source and proprietary models. In addition, this work also considers an important baseline, OCR + uni-modal (i.e., text-only) LLM, which I think makes a lot of sense and highlights the current situation and challenges of LVLMs for document understanding.

**Additional Feedback:**

Please see my reviews on the pros and cons above.

**Documentation:**

Yes.

**Ethics:**

No potential ethical concern found.

**Limitations:**

The data filtering and quality control process heavily relies on GPT-4o, which makes me wonder whether the benchmark can be biased toward GPT-4o’s answers. This might explain why GPT-4o is the best-performing model by a large margin.

**Opportunities For Improvement:**

1. In addition to the LLM and LVLM baselines, I think it’s also worth including a “human baseline”. Just to give a sense of how difficult these tasks are for humans.
2. This work categorizes question types by the number of evidence pages, and it shows some fine-grained results based on document domains. In addition to these categorizations, I think it’d also be helpful to categorize questions by the capabilities required to answer them. For example, some question requires grounding/localizing information from the documents, while others may also require some further reasoning abilities.

**Relation To Prior Work:**

Yes. The authors discussed related works in both the main paper and the appendix.

**Summary And Contributions:**

This work presents a benchmark for long-context document understanding of vision-language models. The benchmark, MMLongBench-Doc includes over 1000 human annotated questions filtered by some quality control procedures. The authors evaluated a wide range of existing multimodal models on the proposed benchmark and did some analysis of the data statistics.

---

> ### Author Rebuttal · Authors · 2024-08-16
>
> Thanks a lot for your reviews! Your professional reviews offer us great advice towards writing a more comprehensive and competitive paper! And, we appreciate you finding our benchmark practical and novel, and our evaluation extensive. Below are our responses to your questions.
>
> ---
>
> > Q1. In addition to the LLM and LVLM baselines, I think it’s also worth including a “human baseline”. Just to give a sense of how difficult these tasks are for humans.
>
> We appreciate your suggestions and agree with you that a human baseline is helpful to better understand the challenging extent of this dataset. We plan to add this human baseline in a two-phase way:
>
> **Stage 1**: We have randomly selected 29 documents and 238 questions to seven annotators. The human-evaluation results on this subset are expected to be updated during the discussion phase (before 31th, August). We request your kind patience and continual attention.
>
> **Stage 2**: We will conduct human evaluation on the remaining samples later and add their results to our paper.
>
> > Q2. I think it’d also be helpful to categorize questions by the capabilities required to answer them. For example, some question requires grounding/localizing information from the documents, while others may also require some further reasoning abilities.
>
> * Our categorization on evidence pages can be viewed as a **rough** mapping to the required abilities. To some extent, single-page questions focus more on information retrieval/grounding, while multi-page questions necessitate comprehensive reasonings across different pages.
> * We also realized the absence of an **accurate** taxonomy and discussed it in the Limitation section (Supplementart material section E; Lines 351-361). Briefly speaking, the intrinsic complexity of document understanding presents significant challenges for establishing a clear, fine-grained taxonomy on required abilities. We will explore it in the future.
>
> > Q3. The data filtering and quality control process heavily relies on GPT-4o, which makes me wonder whether the benchmark can be biased toward GPT-4o’s answers. This might explain why GPT-4o is the best-performing model by a large margin.
>
> * The short answer is: **our GPT4o-involved quality control process do bring slight bias (1.1% absolute difference at maximum) toward the predictions from GPT4o. However, such bias is NOT the main cause of GPT4o's significantly best performance. And all primary conclusions in our paper still hold**. We detail about it briefly as below:
>
> * We check the effect of GPT-4o's involvement in the quality control step-by-step. Specifically, we compare the performance of samples remained after each step across GPT4o and two other competitive models (GPT-4V and Gemini-1.5-Pro). We show their results in the table below.
>
> | | GPT-4o | GPT-4V | Gemini-1.5-Pro |
> |---|---|---|---|
> |No quality control|43.1|35.2 (-7.9%)|23.3 (-19.8%)|
> |+ document-relevance detection| 41.2 | 31.0 (-10.2%) | 20.5 (-20.7%) |
> | + document-relevance detection + self-reflection / cross-checking |42.7 | 31.4 (-11.3%) | 20.9 (-21.8%)|
>
> *The numbers in the brackets represent the performance gaps compared with GPT-4o.*
>
> 1. The potential bias in step 1 (document-relevance detection) actually reduce (rather than increase) the performance gap between GPT4o and other models. It is because that we filter out *all samples correctly answered by GPT4o* without the access to documents. Under this case, the more significant performance drop of GPT-4V and Gemini-1.5-Pro can only be attributed to their limited document understanding and over-reliance on their internal knowledge.
> 2. Regarding the step 2 and 3 (self-reflection and cross-checking), we provide inconsistent answers between human annotations and GPT4o's predictions to annotators and ask them to check and revise accordingly. The potential bias of this step does lead to a slight performance bias (1.1% absolute difference at maximum):
> ```
> Bias for GPT-4V: (42.7%-41.2%) - (31.4%-31.0%) = 1.1%
> Bias for Gemini-1.5-Pro: (42.7%-41.2%) - (20.9%-20.5%) = 1.1%
> ```
>
> * We believe that such bias is NOT the main cause of GPT4o's significantly best performance. Without the involvement of GPT-4o in the quality control process, GPT-4o still significantly outperforms GPT-4V by `43.1% - 35.2% = 7.9%` and Gemini-1.5-Pro by `43.1% - 23.3% = 19.8%`. Accordingly, all primary conclusions in our paper still hold.
>
> * Thanks for your insightful comment about the potential bias in quality control. We have added a discussion and clarification about this potential bias in Appendix and will update in the final paper. Additionally, we will use a set of models (instead GPT4o only) to ensembly involve the quality control and mitigate the potential bias in our future updated dataset.
>
> ---
>
> Thanks again for your comments and reading our responses. We hope the explanations above can address your concerns about our work. Shall you have any further question, free free to tell us and we will try our best to solve it.

---

> > ### Author Response · Authors · 2024-08-21
> > **Human Baseline**
> >
> > As mentioned above, we firstly conduct human evaluation on a subset of our benchmark (29 out of 130 documents). And the results are shown as below:
> >
> > ||Acc|F1|
> > |---|---|---|
> > |Gemini-Pro-1.5|28.8|20.9|
> > |GPT-4V|32.5|31.4|
> > |GPT-4o|40.8|42.7|
> > |**Human**|65.8|66.0|
> >
> > We observe a significant performance gap (exceeding 20% in absolute) between the current LVLMs and humans. This gap highlights the challenges of document understanding for LVLMs and the necessity of our benchmark.
> >
> > We will complete the human evaluation of remaining samples, and add their results in our updated paper. Thanks again for your constructive suggestion.

---

### Decision · Program_Chairs · 2024-09-26

**Decision:**

Accept (Spotlight)

**Comment:**

This paper addresses the challenge of understanding documents with rich layouts and multi-modal components. All reviewers, after rebuttal discussion, gave very positive feedback. I also believe the topic of the paper is important, the dataset quality is excellent, evaluation is thorough and careful. Hence, I recommend a definite accept. Also, it is highly suggested that the new information, e.g. human evaluation etc., should be added to the revised paper.